# Enhanced Notch dependent gliogenesis and delayed physiological maturation underlie neurodevelopmental defects in Lowe syndrome

Yojet Sharma [1,2], Priyanka Bhatia [1,3], Gagana Rangappa [1], Sankhanil Saha [1,4] & Padinjat Raghu [1✉]

## Abstract

**Coordination of cellular and physiological development by signaling is required for normal brain structure and function. Mutations in *OCRL*, a phosphatidylinositol 4,5 bisphosphate [PI(4,5)P$_2$], 5-phosphatase leads to Lowe Syndrome (LS). However, the mechanism by which mutations in *OCRL* leads to the neurodevelopmental phenotypes of LS is not understood. We find that on differentiation of LS patient iPSC, neural cultures show reduced excitability and enhanced GFAP levels. Multiomic single-nucleus RNA and ATACseq analysis of neural stem cells revealed enhanced numbers of cells with a gliogenic cell state. Analysis of snRNA seq revealed increased levels of *DLK1*, a Notch ligand in LS patient NSC associated increased levels of cleaved Notch and elevation of its transcriptional target *HES5*, indicating upregulated Notch signaling. Treatment of iPSC derived brain organoid with an inhibitor of PIP5K, the lipid kinase that synthesizes PI(4,5)P$_2$, was able to restore neuronal excitability and rescue Notch signaling defects in OCRL deficient organoids. Overall, our results demonstrate a role for PI(4,5)P$_2$ dependent regulation of Notch signaling, cell fate specification and neuronal excitability regulated by OCRL.**

**Keywords** Phosphoinositides; iPSC-derived Neurons; Brain Organoids; Lowe Syndrome; Disease Model
**Subject Categories** Genetics, Gene Therapy & Genetic Disease; Neuroscience

## Introduction

The human brain is a complex organ whose development involves the regulated activity of many genes. In many cases, an understanding of how the function of a gene during brain development has come from studying brain disorders in human patients that harbor mutations in specific genes. These conditions, referred to as neurodevelopmental disorders (NDD) are a heterogeneous group of

conditions in which individuals present with abnormalities of brain structure and/or function (Dyment et al, 2015; Parenti et al, 2020). NDD represent the single largest class of monogenic inherited brain disease (Raghu et al, 2024) listed in Online Catalog of Human Genes and Genetic Disorders (OMIM®) database (https://omim.org/). Therefore, the study of cellular and developmental mechanisms underpinning NDD can lead to an understanding of normal brain development. Conversely, an understanding of underlying cellular and molecular mechanisms in NDD can help devise better strategies for the clinical management of patients.

The discovery of a diverse set of genes underlying NDD has led to a better understanding of cellular and development processes underpinning human brain development. Key examples include the identification of doublecortin (*DCX*) as the gene underpinning abnormal cerebral cortex development and morphology in patients with X-linked lissencephaly (XLS); analysis of XLS has underscored the importance of normal neuronal migration during cerebral cortex development (Allen and Walsh 1999; Gleeson et al, 1999). Likewise, altered brain size and malformations in patients with mutations in the lipid phosphatase *PTEN* (Sansal and Sellers 2014) or the PI3K/AKT pathway (Poduri et al, 2012; Jansen et al, 2015) [reviewed in (Raghu et al, 2019)] has revealed the importance of lipid signaling in normal brain development. Mutations in *NOTCH2NL*, a human specific gene that regulates Notch signaling have underscored the importance of Notch-regulated neurogenesis in the development of normal size and structure in the human neocortex (Fiddes et al, 2018).

Lowe syndrome (LS) is a congenital X-linked recessive disorder in which three organs are affected, the eye, kidney and the brain (Bökenkamp and Ludwig 2016). Young boys with LS present with a cataract in early life, proximal renal tubular dysfunction and various degrees of neurodevelopmental deficit. The neurological features of LS include delayed cognitive milestones, ranging from mild cognitive impairment to severe intellectual impairment, susceptibility to febrile seizures, hypotonia and in later years psychiatric symptoms. Imaging of brain structure in LS patients through Magnetic Resonance Imaging (MRI) (Charnas et al, 1988; Allmendinger et al, 2014) reveals normal brain size with features of delayed myelination and focal, periventricular cystic lesions that develop with age; these findings not specific to but consistent with

[1]National Centre for Biological Sciences-TIFR, GKVK Campus, Bangalore 560065, India. [2]Centre for Doctoral Studies, Manipal Academy of Higher Education, Manipal 576104, India. [3]Present address: Berlin Institute for Medical Systems Biology (BIMSB), Max Delbrück Center for Molecular Medicine (MDC) in the Helmholtz Association, 10115 Berlin, Germany. [4]Present address: Department of Neuroscience, Tufts University School of Medicine, Boston, MA 02111, USA. ✉E-mail: praghu@ncbs.res.in

gliotic lesions. LS results from a mutation in the Oculo Cerebro Renal syndrome of Lowe *OCRL* gene (Attree et al, 1992). *OCRL* encodes for a protein that is part of a larger family of phosphoinositide 5 phosphatases (Ramos et al, 2019). The OCRL protein mainly shows phosphatase activity on phosphatidylinositol 4,5 bisphosphate ($PI(4,5)P_2$) generating phosphatidylinositol 4 phosphate (PI4P) (Zhang et al, 1995). OCRL localizes to a number of cellular organelles including the plasma membrane, Golgi, lysosomes and early endosomal compartments (Mehta et al, 2014) and multiple studies have implicated OCRL in the regulation of vesicular transport and the actin cytoskeleton (reviewed in Mehta, Pietka and Lowe, 2014; De Matteis et al, 2017). While many studies have attempted to address the function of *OCRL* in relation to the kidney phenotypes noted in human LS patients, much less is known about the cellular and developmental basis of the brain phenotypes in this disease. Some studies have sought to create models of LS with a view to understand brain phenotypes, including a zebrafish and an iPSC-derived neuronal model (Ramirez et al, 2012; Fuentealba et al, 2024; Liu et al, 2020). It has been noted that a humanized mouse model for LS does not show phenotypes that recapitulate the cognitive impairment noted in LS patients although a hypotonia secondary to the renal manifestations has been reported (Festa et al, 2019). This coupled with the inability to obtain biopsy samples from LS patient brains have severely impaired the understanding of the neurodevelopmental basis of LS syndrome.

In this study, we report the cellular, developmental and physiological phenotypes of 2D and 3D brain organoid cultures differentiated from LS patient derived induced pluripotent stem cell (iPSCs) lines as well as an isogenic OCRL knockout generated in wild-type iPSCs. We find that *OCRL* deficient iPSC derived neurons exhibit reduced neuronal excitability than controls. Loss of OCRL also led to an increase the expression of Glial Fibrillary Acidic Protein (GFAP) protein in 2D and 3D brain organoid models derived from LS patient as well as OCRL knockout cultures. Multiomic analysis of neural stem cells (NSC) derived from LS patients revealed an enhanced number of cells bearing molecular signatures of glioblasts and astrocytes accompanied by enhanced Notch signaling. These findings could be rescued by chemical inhibitors that reduce the synthesis of $PI(4,5)P_2$, the substrate of OCRL. Together, these findings suggest that during neural development, OCRL function is required to regulate $PI(4,5)P_2$-dependent processes that control cell fate specification and neuronal maturation in the developing brain.

## Results

### *OCRL* is expressed during human brain development

To understand the role of *OCRL* in neurodevelopmental defects observed in Lowe Syndrome (LS) patients, we utilized publicly available single cell and bulk transcriptomic datasets and analyzed OCRL expression patterns during neural development, as outlined in the schematic Fig. 1A. We analyzed the gene expression pattern of *OCRL* in the developing human brain using a single-cell multiomics dataset (Wang et al, 2025) that included 38 biological samples divided into 27 brain regions from across five major developmental stages, ranging from the first trimester to adolescence. This dataset allowed us to map *OCRL* expression at the transcriptomic level across various cell types in the developing brain. A UMAP projection of this dataset reveals distinct clusters representing major cell types of the brain, including radial glia, intermediate progenitor cells (IPC), neurons, astrocytes, oligodendrocytes and microglia (Fig. 1B). In addition, *OCRL* mRNA was expressed at varying levels depending on the developmental stage and cell type. The dot plot shows that *OCRL* expression was detected throughout development, with the highest expression detected during the 246-507 days post conception (Fig. 1C). Further, analysis of cell-type specific enrichment of *OCRL*, revealed that in this dataset, this gene was predominantly expressed in GABAergic and glutamatergic neurons compared to other cell-types (Fig. 1D). These observations suggest that *OCRL* plays a role during in utero brain development.

We also examined expression of the OCRL protein in iPSC-derived human brain organoids at different stages of in vitro development using a publicly available proteomics dataset (Sidhaye et al, 2023). The proteomics dataset revealed that at all stages of in vitro differentiation, hSyn+ mature neurons expressed higher levels of OCRL protein compared to Sox2+ NSC (Fig. 1E). Likewise, in the transcriptomic dataset, *OCRL* transcript was seen in both neurons and NSC (Fig. 1F). Importantly, OCRL expression increased progressively over time, indicating a potential role in neuronal maturation and further supporting the age-specific enrichment observed in fetal brain samples.

To validate these in silico findings, we utilized a previously established wild-type control induced pluripotent stem cell (iPSC) line (WT1) (Iyer et al, 2018) to generate dorsal forebrain cortical NSC using the dual-SMAD inhibition protocol (Fig. 1G) (Shi et al, 2012). The NSC stained positive for characteristic markers: Nestin, Pax6; intermediate progenitor marker TBR2 and also proliferative marker KI67 (Fig. 1H). Upon terminal differentiation into cortical neurons, cells expressed mature neuronal markers such as MAP2 (pan-neuronal), CTIP2 (deep-layer), and GABA (interneurons) (Fig. 1H). To further characterize the neuronal cultures, we first examined the expression of synapsin-1, a synaptic vesicle protein that marks synapses. Western blot analysis revealed a strong synapsin-1 band at 60 days in vitro (DIV), indicating the formation of mature synaptic connections in our cultures (Fig. 1I). We observed a decline in synapsin-1 levels at 90DIV compared to earlier time points. It's important to note that levels of synaptic proteins can fluctuate due to various factors, including culture conditions and the heterogeneity of neuronal populations present at different time points (Togo et al, 2021). Similarly, we determined OCRL expression during differentiation. Western blot analysis of OCRL expression at various time points (iPSC, NSC, 15, 30, 60, 90DIV) confirmed a gradual increase in OCRL protein levels as differentiation progressed, with higher levels in mature neurons compared to NSC or early-stage neurons (Fig. 1I,J). Overall, our findings support an age dependent increase in OCRL protein expression during neuronal differentiation.

### Loss of *OCRL* leads to reduced neuronal activity during development

We generated iPSCs derived from three individual patients (LSP) with LS from a single family (Pallikonda et al, 2021; Akhtar et al, 2022). These include two monozygotic twins (LSP2 and LSP3) and their

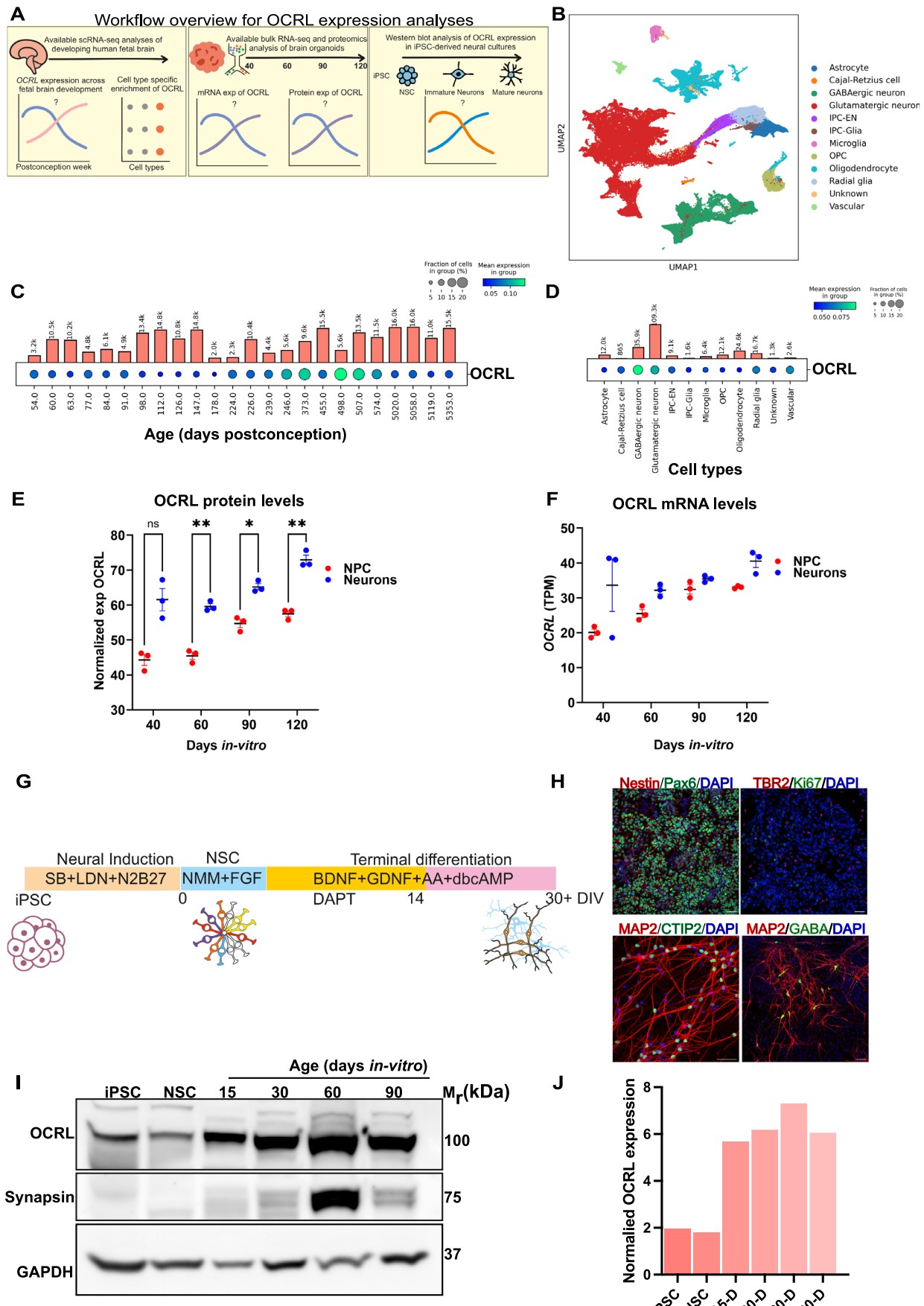

◄

Figure 1. OCRL expression during human brain development.

maternal cousin (LSP4). All three LSP carry the same mutation in exon 8 (c688C>T) leading to truncated protein (p.Arg230X) as previously reported (Pallikonda et al, 2021; Akhtar et al, 2022). These iPSCs were differentiated into NSC which expressed the characteristic protein markers Nestin, Pax6 and FoxG1 (Fig. EV1A). Further, terminal differentiation was initiated by seeding equal number of NSC (45,000–55,000 cells/cm²) and culturing them for 30DIV. The cortical neurons so generated were confirmed by the expression of mature neuronal markers MAP2 and a neuronal subtype marker CTIP2, indicative of cortical layer 6–5 (Fig. 2A).

To measure neuronal excitability, we selected field of views that had uniform regions of cells for all genotypes (Movies EV1–4) and performed intracellular calcium ($[Ca^{2+}]_i$) imaging on WT1 and LSP neurons (LSP2, LSP3, LSP4). Representative $[Ca^{2+}]_i$ traces from each genotype are shown in Fig. 2B. Quantification revealed a significant reduction in the frequency of $[Ca^{2+}]_i$ transients in all three LSP neurons compared to WT1 controls (Fig. 2C). To test the dependence of this phenotype noted in LSP derived neurons on OCRL function, we generated a knockout iPSC line (OCRL[KO]) using CRISPR-Cas9 genome engineering and a stop codon was introduced in exon 8 thus recapitulating the OCRL mutation observed in the LSP used in this study (Fig. EV1B,C). As an isogenic control, we used the parent line (NCRM5 or WT2) that was used to generate OCRL[KO]. The WT2 and OCRL[KO] iPSC were positive for pluripotency markers SSEA4, Sox2, Tra-160 and Oct4 (Fig. EV1D) and OCRL[KO] iPSC also had normal karyotype (Fig. EV1E). We generated NSC from WT2 and OCRL[KO] (Fig. EV1F) and differentiated these into neurons (Fig. 2D). Western blot analysis confirmed the absence of OCRL protein in OCRL[KO] neurons compared to WT2 controls (Fig. 2E). To test the biochemical consequence of OCRL depletion in OCRL[KO], we measured the levels of PIP₂ from 30 DIV neurons using liquid chromatography, based mass spectrometry. As expected, PIP₂ levels were significantly elevated in OCRL[KO] neurons relative to WT2 controls (Fig. 2F), recapitulating the findings previously reported by us for LSP lines (Akhtar et al, 2022). $[Ca^{2+}]_i$ imaging on OCRL[KO] neurons (Fig. 2G) revealed a statistically significant reduction in the frequency of $[Ca^{2+}]_i$ transients in OCRL[KO] compared to WT2 (Fig. 2G,H) recapitulating our findings in LSP-derived neurons (Fig. 2B,C).

To corroborate our finding of reduced neuronal excitability in OCRL-deficient neurons, we performed whole-cell patch-clamp recordings on 30 and 60DIV neurons (Fig. 2I). Resting membrane potential in 30 DIV OCRL[KO] neurons showed no significant difference, while 60DIV OCRL[KO] neurons revealed a significant reduction compared to WT2 (Fig. EV2A). Capacitance measurements were no different between WT2 and OCRL[KO] 60DIV neurons (Fig. EV2B). Moreover, inward currents were reduced in OCRL[KO] neurons compared to WT2 (Fig. 2J) and outward currents also showed a similar trend of being reduced in OCRL[KO] (Fig. 2K). Similar patterns were observed in recordings from LSP2 and LSP3 patient-derived neurons compared to WT1 at 60 DIV, particularly for inward currents but not for outward currents (Fig. EV2C,D). However, it is important to note that fewer patient and OCRL[KO] derived neurons were available for recording at this time point, limiting our sample size and thus leading to a higher standard error of mean (Fig. EV2C,D). Collectively, these results demonstrate that loss of OCRL leads to reduced neuronal excitability as evidenced by reduction in calcium transients and reduced inward and outward currents. This phenotype is likely to be mediated by the observed increase in PI(4,5)P₂ levels. Thus, potentially highlighting a link between OCRL deficiency and altered neuronal function during neurodevelopment in vitro.

## OCRL deficiency leads to altered cell type composition in iPSC-derived neural cultures

During this study we stained iPSC-derived cultures from control and OCRL-deficient lines for markers of neurons and astrocytes. We checked if the neuronal subtype specification was affected in LSP neural cultures. It is well established that mammalian cortical lamination follows an inside-out model where deep layer neurons (CTIP2+/TBR1+) are generated first followed by upper layer (BRN2+/SATB2+) neurons (Molyneaux et al, 2007; Leone et al, 2008) (Fig. 3A). Immunocytochemistry revealed that both upper layer neurons (BRN2+, green) and deep layer neurons (TBR1+, green) are present in similar distributions across wild-type (WT1) and all three patient lines (LSP2, LSP3, LSP4) (Fig. 3B).

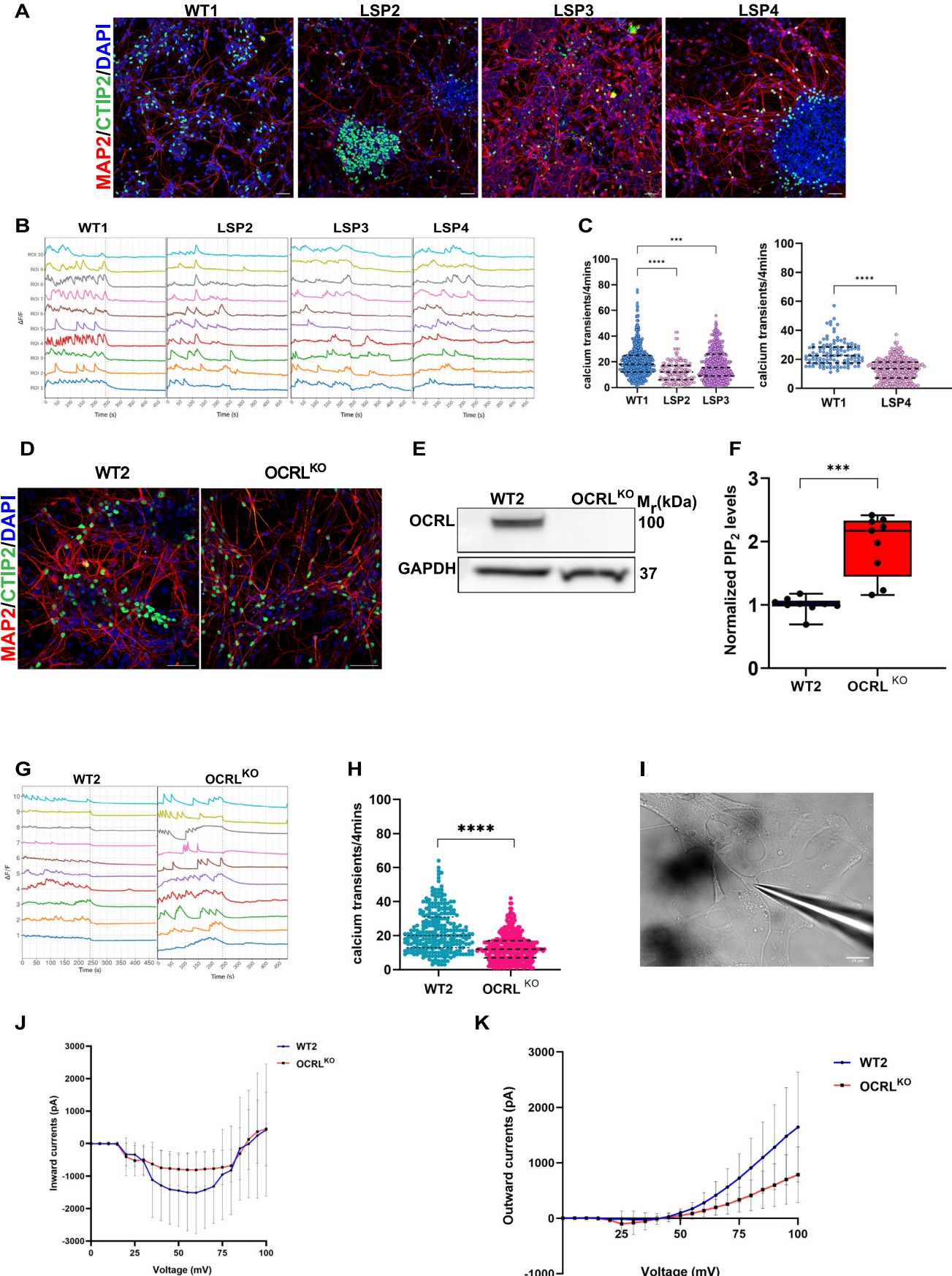

**Figure 2.  Altered physiological activity in OCRL deficient developing neurons.**

(A) Immunofluorescence images of 30-day in vitro (DIV) neurons from WT1, LSP2, LSP3, LSP4, stained for mature forebrain cortical neuron markers MAP2 (red) and CTIP2 (green). Nuclei are stained with DAPI (blue). Scale bar = 50 µm. (B) Representative calcium ([$Ca^{2+}$]$_i$) transient traces from 30 DIV neurons of WT1, LSP2, LSP3, and LSP4, normalized to the first frame. Each trace (of a different color, represents [$Ca^{2+}$]$_i$ transients from the soma of a single neuron (ROI). A dotted line marks the addition of TTX (10 µM), which abolishes calcium transients. (C) Scatter plots showing the frequency of calcium transients per 4 min for WT1, LSP2, LSP3 and LSP4. Each data point represents calcium transient activity from a single neuron, collected from three independent ($n = 3$) neuronal differentiations. Statistical test: one-way ANOVA Dunnett's multiple comparison test with $p = 0.0004^{***}$, $p = <0.0001^{****}$. (D) Immunofluorescence images of 30 DIV neurons from WT2 and OCRL$^{KO}$ stained for mature forebrain cortical neuron markers MAP2 (red) and CTIP2 (green). Nuclei are stained with DAPI (blue). Scale bar = 50 µm. (E) Western blot showing OCRL protein expression in WT2 and OCRL$^{KO}$ neurons, with GAPDH as the loading control. (F) Liquid chromatography-mass spectrometry (LC-MS) analysis measuring total PIP$_2$ levels in 30 DIV WT2 and OCRL$^{KO}$ neurons. Data are shown as individual points from two independent biological replicates ($n = 2$) for each genotype. The data is plotted as box and whiskers plot with min and max to include all data points, indicating the median, the interquartile range (box limits indicate 25th and 75th percentiles), whiskers extend to the minimum and maximum values. Statistical test: Unpaired t-test with Welch's correction, $p = <0.0003^{***}$. (G) Representative calcium transient traces for 30 DIV WT2 and OCRL$^{KO}$ neurons. (H) Frequency of [$Ca^{2+}$]$_i$ transients in neurons from WT2 and OCRL$^{KO}$. Each data point represents a single neuron, collected from 2 independent differentiations ($n = 2$). Statistical test: Unpaired t-test with Welch's correction, $p = <0.0001^{****}$. (I) Showing a representative image of patched 60 DIV neuron. Whole-cell patch clamp recording of WT2 and OCRL$^{KO}$ neurons in voltage-clamp mode exhibits. Scale bar = 20 µm. (J) inward currents and (K) outward currents, respectively. Y-axis -current in pA; X-axis voltage in mV. For both inward and outward currents multiple neurons were recorded from $n = 2$ independent differentiations. Error bars in all the data represent Mean $+/-$ SEM.

MAP2 staining (red) showed comparable staining pattern between control and patient cultures, with inset zoomed-in images confirming no obvious morphological differences (Fig. 3B). Cultures stained for the mature astrocytic marker GFAP and immature marker S100β, showed an increase in staining in each of the three LS patient lines compared to WT1 at 30 DIV (Fig. 3C). This increase in GFAP staining was confirmed by Western blot analysis, which showed elevated GFAP protein expression in LSP neural cultures relative to WT1 (Fig. 3D), with quantification revealing an increased fold change in each of the LSP lines compared to WT1 (Fig. 3E). To test if loss of OCRL was sufficient to produce this phenotype, we compared cultures differentiated from WT2 with OCRL$^{KO}$ at 30 DIV. Immunofluorescence images from 30 DIV OCRL$^{KO}$ neural cultures also displayed an increased signals of GFAP+ cells compared to their isogenic WT2 controls (Fig. 3F) and western blot analysis confirmed significantly higher GFAP expression observed in OCRL$^{KO}$ neural cultures compared to WT2 (Fig. 3G,H). Together, these findings imply that loss of OCRL leads to increased gliogenesis in LSP neural cultures but no effects were observed in neuronal subtype specification.

## Altered cellular neurodevelopmental state in Lowe syndrome patient NSC

Since, we noted an increased expression of the glial marker GFAP in LSP and OCRL$^{KO}$ cultures on terminal differentiation of NSC into neural cultures, we wondered if absence of OCRL in LSP NSC had altered cell states compared to controls. To test this, we carried out single nuclei analysis of the NSC population derived from iPSC of WT1 and the LSP lines LSP2 and LSP3. For our experiments, we employed single-nuclei multiomics (scMultiome) analysis, which integrates both transcriptomic (RNA-seq) and chromatin accessibility (ATAC-seq) and allows the simultaneous assessment of gene expression and chromatin accessibility at single-nuclei resolution (Ma et al, 2020; Zhu et al, 2023).

The workflow for our scMultiome analysis is outlined (Fig. EV3A). Nuclei were isolated from LS patient-derived NSCs (LSP2, LSP3) and wild-type controls (WT1), followed by transcriptome (RNA-seq) and chromatin accessibility (ATAC-seq) data generation using the 10X Chromium Single Nuclei Multiome ATAC + Gene Expression kit. After pre-processing and quality control, we retained 19780 nuclei for

WT1, 19854 nuclei for LSP2 and 19953 nuclei for LSP3 for analysis. The UMAP generated from the single nuclei RNA-seq data showed three distinct clusters corresponding to WT1, LSP2 and LSP3 (Fig. 4A). Using the Leiden graph-based clustering method, we obtained 14 clusters (Fig. EV3B). Each cluster was defined by a set of highly expressed transcripts (Fig. EV3C). To ensure robust cluster annotation, we employed two levels of annotation: a first level was performed using snapseed (He et al, 2024) and a second level of cluster annotation was carried out using scVI-scanVI framework (Gayoso et al, 2022) to refine cluster identities into more specific cell-states. To this end, we leveraged already available snRNA/snMultiome reference of developing human brain datasets (Braun et al, 2023; Wang et al, 2025) for accurate mapping of cell states using the scVI-scanVI framework (Figs. 1B and EV3D). First, using snapseed, we obtained preliminary marker-based hierarchical cell type annotation. Second, we projected our NSC dataset to the latent-space of the developing human reference datasets using scVI and scanVI, and generated confusion matrices (Fig. EV3E,F). Cell type labels were assigned based on the highest confidence scores, represented by the yellow and green/yellow bands in the confusion matrices (Fig. EV3E,F). These high confidence matches in the transfer learning approach were used to establish the final cell type annotations, resulting in the labeled UMAP visualization (Fig. 4B). Notably, LSP2 NSCs showed 0.5% (112 nuclei) corresponding to astrocytes, 86% (17096 nuclei) to glioblast. On the other hand, LSP3 NSCs had 28% nuclei (5730) for astrocytes and 57.53% towards glioblast cells. WT1 NSCs had 54.91% (10862) radial-glia cells, but only 0.16% (33) for glioblast and none for astrocytes (Fig. 4C). To verify whether this gliogenic bias could be observed in NSC, we carried out RT-PCR for glial lineage markers NFIA and OLIG2. We observed that the expression level of NFIA and OLIG2 was increased in LSP NSC compared to WT1 NSC (Fig. EV3G,H). This marked increase in glioblast and astrocyte cell states in LSP compared WT1 at NSC stage explains the increased GFAP expression observed in LS patient-derived neural cultures after terminal differentiation (Fig. 3A).

To independently study the cell composition of LSP NSC, we used SnapATAC2 for clustering cells based on chromatin accessibility profiles. UMAP plots of the ATAC-seq data showed cells from WT1, LSP2 and LSP3 clustering separately (Fig. EV4A) with LSP2 and LSP3 clustering closer to each other and away from WT1. Further clustering led to the identification of clusters of cells corresponding to the 14 types (Figs. 4D and EV4B) also identified

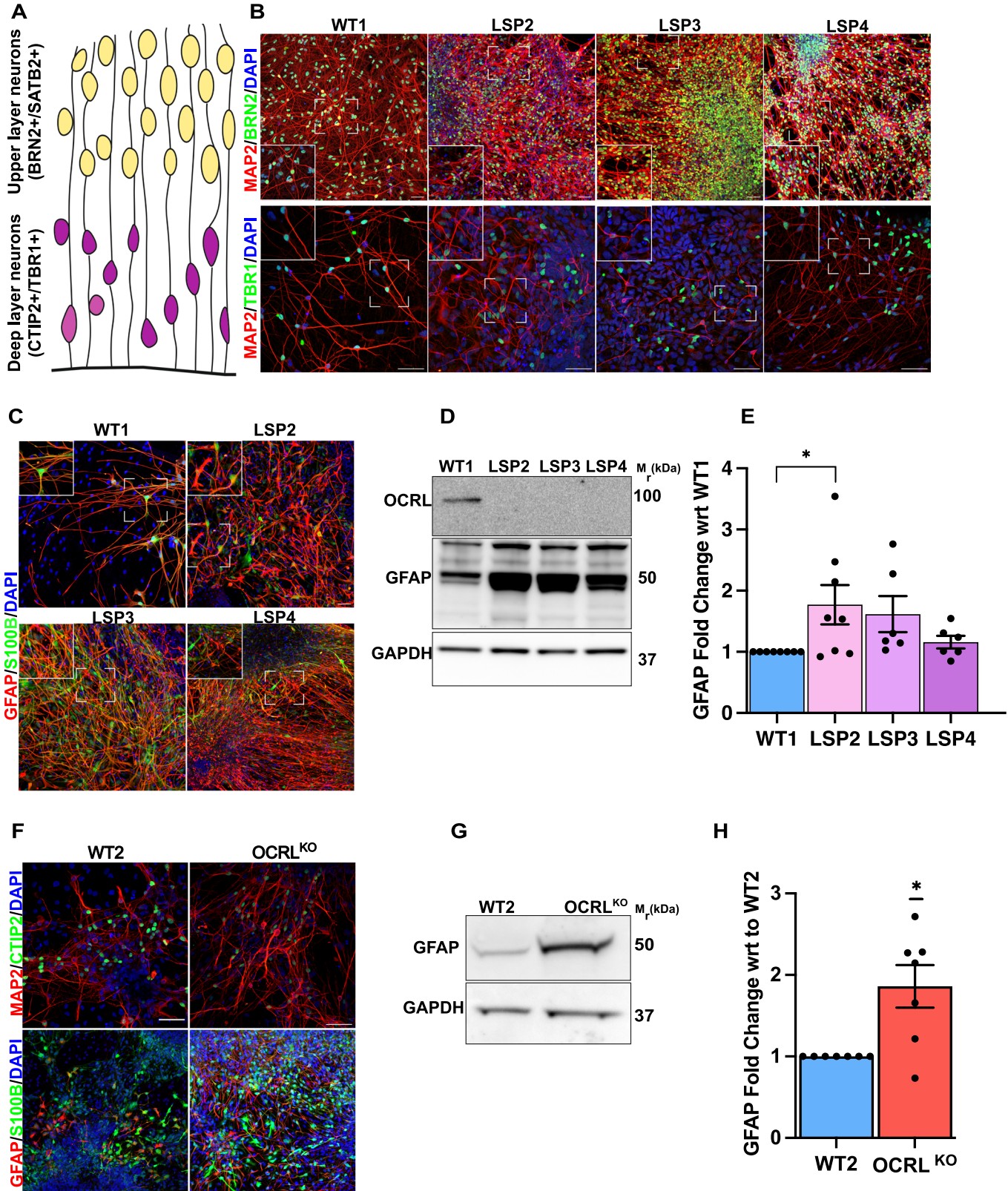

Figure 3. Altered cellular composition on OCRL depletion.

(A) Schematic representation of cortical neuron laminar arrangement in developing cortex. Upper layer neurons (yellow) are identified by BRN2 and SATB2 expression, while deep layer neurons (purple) are marked by CTIP2 and TBR1. (B) 30 DIV neural cultures from WT1, LSP2, LSP3, LSP4 stained with upper layer neuronal marker BRN2 (green, top panel) and deep layer neuronal marker TBR1 (green, bottom panel). MAP2 (red) was used as pan-neuronal marker and DAPI (blue) was used to nuclei. Insets highlight regions of interest. Scale bar = 20 μm, 50 μm. (C) 30 DIV neural cultures from WT1, LSP2, LSP3, LSP4 stained with mature astrocytic marker GFAP (red) and immature astrocytic marker S100β (green), respectively. Insets highlight regions of interest. Scale bar = 20 μm, 50 μm. (D) Western blot with protein lysates from 30 DIV iPSC derived cultures of WT1, LSP2, LSP3, LSP4 cell lysates; immunoblotting for OCRL (100 kDa) and GFAP (50 kDa) proteins are shown; GAPDH (37 kDa) was used as a loading control. (E) GFAP fold-change in LSP derived 30 DIV neural cultures analyzed and plotted w.r.t. the control WT1. Each point in the western blot analyses represents data from independent neuronal differentiation (n = 6). Y-axis shows the fold change in GFAP levels in LSP relative to WT1. Statistical test: Unpaired t-test with Welch's correction was used to calculate statistical significance p = 0.0479*. (F) 30 DIV WT2 and OCRL^KO neurons were also stained with MAP2 (red, top panel) and CTIP2 (green, top panel); and GFAP (red, bottom panel) and S100β (green, bottom panel). Scale bar = 50 μm. (G) Western blot of GFAP for WT2 and OCRL^KO showing levels of GFAP protein in 30 DIV neural cultures. (H) GFAP fold-change in GFAP levels analyzed and plotted w.r.t. WT2 control across independent neural differentiations (n = 7). Statistical significance was calculated using the one-sample t-test, p = 0.0168*. Error bars represent Mean +/− SEM.

in the single nuclei RNA-seq data (Fig. 4B). We found that an integration of transcriptome (RNA-seq) and chromatin accessibility (ATAC-seq) data showed strong concordance with each other, as evidenced by the diagonal pattern of high correlation (dark red) in the confusion matrix (Fig. 4E).

To identify enhancer driven gene regulatory interactions, we applied SCENIC+ (Bravo González-Blas et al, 2023), a tool that identifies putative enhancer regions in the genome and their associated transcription factors, while simultaneously mapping these regulatory elements to their potential downstream target. Motif analysis revealed enrichment of transcription factors (Fig. EV4C) associated with astrocyte lineage commitment, including nuclear factor one (NFI) family of transcription factors NFIA, NFIB, NFIC and NFIX, and SOX2, POU2F1 (OCT1) in glioblasts (Namihira et al, 2009; Piper et al, 2010; Tchieu et al, 2019). Additionally, radial glia-specific motifs such as LHX9 and EMX1 were identified in radial glia populations. Using, SCENIC+ we analyzed the multiome data in relation to cell type clusters asking which transcription factors, both activators and repressors were active in each cell type. We analyzed both the degree of enrichment and specificity of transcription factors as well as how well the binding sites of transcription factors are enriched in regulatory regions specific to each cell type. This analysis revealed a signature of transcription factor activity for each cell type cluster (Fig. 4F). For example, transcription factors and sites of active chromatin in neurons showed presence of neuron-specific transcription factors such as ONECUT1/3 that were not as enriched in glioblasts and astrocytes (Fig. 4F) (Rhee et al, 2016; van der Raadt et al, 2019). Simultaneously, we also generated the heatmap-dot plot of regulons sample-wise to understand the state of transcription factors of cells in each sample WT1, LSP2 and LSP3 (Fig. 4G). This analysis revealed that WT1 was enriched for binding sites for transcription factors overlapping with those enriched in radial glial cells, intermediate progenitors and neurons (Fig. 4G). By contrast, transcription factors enriched in LSP2 and LSP3 overlapped considerably with those identified for glioblasts and astrocytes such as ETV4 (Liu et al, 2023), SOX9 and NFIA (Fig. 4H). Overall, our findings demonstrate that OCRL deficiency alters the transcriptional landscape of NSC, priming the LSP NSC towards accelerated gliogenesis.

## Loss of OCRL leads to altered Notch signaling

During pseudobulk analysis of our single nucleus RNAseq dataset we noted that levels of the atypical Notch ligand, DLK1 (Bray et al, 2008; Ferrón et al, 2011), were elevated in LSP2 and LSP3 NSC compared to WT1, along with the long non-coding mRNA MEG3 and MEG8 that are found in the DLK1-DiO3 locus (Aronson et al, 2021) (Fig. 5A). Based on these transcriptomic changes, we hypothesized that absence of OCRL disrupts PI(4,5)P$_2$ metabolism, leading to altered Notch signaling that in turn leads to increased levels of DLK1 in neural stem cells (Fig. 5B). We tested the status of Notch signaling in OCRL deficient cultures. Given the paracrine nature of Notch signaling, that depends on cell–cell interactions we validated these observations by differentiating iPSC into three-dimensional (3D) brain organoid cultures or neural spheroids.

To validate our previous findings of elevated GFAP expression in a 3D neural tissue context, WT1, LSP, WT2 and OCRL^KO iPSC were differentiated into dorsal forebrain cortical organoids (Sloan et al, 2018) (Fig. EV5A). Brightfield microscopy revealed that both patient-derived lines (LSP2, LSP3, LSP4) and OCRL^KO line formed organoids with comparable morphology to their respective controls (WT1, WT2) at days 6 and 15 of differentiation (Fig. EV5B,C). Immunofluorescence analysis at 60 DIV demonstrated robust formation of neural rosettes across all lines, as evidenced by SOX2+ neural progenitor zones with proper apical-basal polarity marked by PKCζ (Fig. EV5D). RT-PCR analysis revealed MAP2 neuronal marker expression showed no significant difference in 90DIV brain organoids (Fig. EV5E). While, varying expression of NFIA was observed: slight reduction in LSP2, and moderate increase in LSP3 and LSP4 relative to WT1 90DIV brain organoids (Fig. EV5F), but a significant elevation was observed in GFAP expression in LSP2 and LSP3 organoids compared to WT1 (Fig. EV5G). We also noted a variable but an increased expression of GFAP in LSP4 brain organoids relative to WT1 (Fig. EV5G). This transcriptional shift toward astrocytic identity was confirmed at the protein level, where western blot analysis demonstrated substantially increased GFAP protein in all 90DIV patient-derived organoids (LSP2, LSP3, LSP4) relative to WT1, with concurrent absence of OCRL protein observed in all patient lines (Fig. EV5H). Dual immunofluorescence staining for GFAP and S100β revealed dramatically expanded astrocytic domains in patient-derived organoids compared to controls (Fig. EV5I). While WT1 organoids showed minimal and dispersed astrocytic staining, LSP2, LSP3, and LSP4 organoids displayed robust, co-localized GFAP and S100β expression throughout large regions of the tissue, indicating widespread emergence of astrocytes (Fig. EV5I). These results demonstrate that the astrocytic bias observed in two-dimensional neural cultures is robustly recapitulated in three-dimensional organoid systems, confirming that OCRL depletion fundamentally

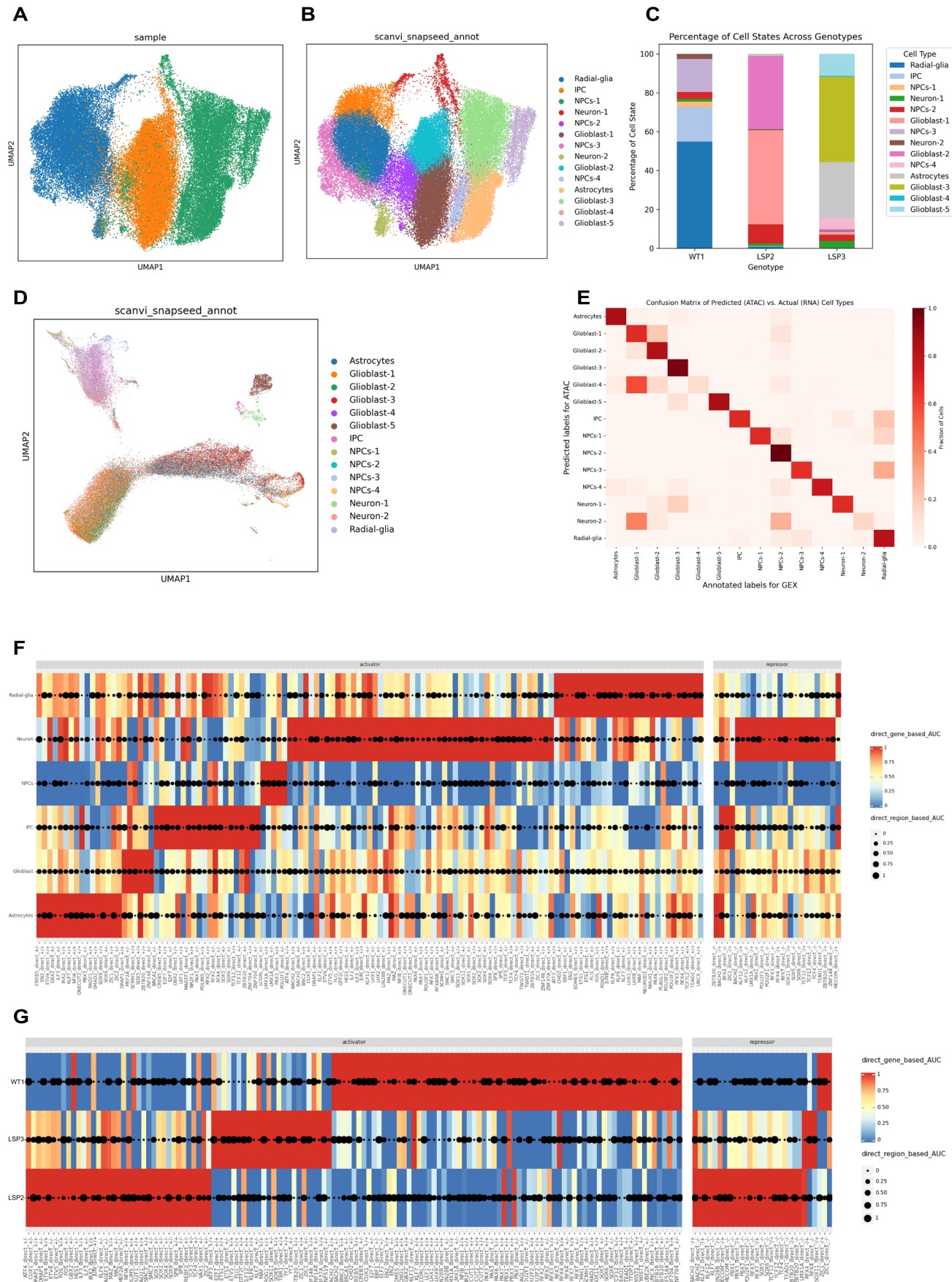

◄

**Figure 4. Single-cell analysis of OCRL deficient NSC.**

(A) UMAP projection of single-nuclei RNA-seq (snRNA-seq) data from WT1, LSP2, and LSP3 samples, showing the distribution of cells across the three genotypes. (B) UMAP projection of snRNA-seq data with cell types annotated using scVI-scANVI mapping. Major cell types include astrocytes, glioblasts, intermediate progenitor cells (IPCs), neuronal progenitor cells (NSCs 1-4), neurons (1-2), and radial glia are shown. (C) Stacked bar plot showing the percentage of each cell type in WT1, LSP2, and LSP3 samples. X-Axis shows the samples identity; Y-axis shows the % of cells of each cluster per sample. (D) UMAP projection of ATAC-seq data from the cells used in the multiome experiment, colored by cell types defined in (B) analyzed using snapATAC2. (E) Confusion matrix comparing cell-type annotations between ATAC-seq and gene expression (GEX) data. X-axis shows annotated labels for GEX and Y-axis shows predicted labels for ATAC-seq. (F) Heatmap and dot plot representing transcriptional regulatory relationships across different cell clusters. The y-axis shows various cell clusters observed: radial-glia, neuron, NSCs, IPC, glioblast and astrocytes. The x-axis shows transcription factors (lower axis) and divided into two categories: activators and repressors (upper axis). The symbol between brackets indicates whether the TF activates (+) or represses (−) its target genes. The color intensity represents direct gene-based Area Under the Curve (AUC) values (ranging from 0 (blue) to 1 (red), indicating the degree of enrichment or specificity of transcription factors in each cell type. The size of black dots represents direct region-based AUC values (ranging from 0 to 1), indicating how well a transcription factor's binding sites are enriched in regulatory regions specific in each cell type. Larger dots indicate stronger enrichment of the transcription factor at regulatory regions characteristic of a cell type. (G) Heatmap and dot plot representing transcriptional regulatory relationships across different samples. The y-axis shows sample clusters: WT1, LSP2, LSP3. The x-axis shows transcription factors (below heat map) divided into two categories: activators and repressors (above heat map). The symbol between brackets indicates whether the TF activates (+) or represses (−) its target genes. The color intensity represents direct gene-based Area Under the Curve (AUC) values (ranging from 0 to 1, blue to red), indicating the degree of enrichment or specificity of transcription factors in each cell type. The size of black dots represents direct region-based AUC values (ranging from 0 to 1), indicating how well a transcription factor's binding sites are enriched in regulatory regions specific to each sample type. Larger dots indicate stronger enrichment of the transcription factor at regulatory regions characteristic of that sample type.

alters neural cell fate decisions toward enhanced gliogenesis across multiple experimental paradigms.

To test whether DLK1 levels were indeed impacted at NSC stage, we carried out western blot analysis from 25 DIV cultures. We chose 25DIV, as at this timepoint, the organoids consists of mainly NSC. The western blot revealed significantly elevated levels of DLK1 protein in lysates from LSP2, LSP3 and LSP4 organoids compared to WT1 (Fig. 5C,D). To test for the status Notch signaling, we also measured the levels of the cleaved Notch receptor (cNotch) by Western blotting. This revealed elevated levels of cNotch in LSP compared to WT1 (Fig. 5C,E) organoids. Lastly, we measured the levels of the Notch signaling dependent effector transcript *HES5* (Bansod et al, 2017) and found it to be significantly elevated in LSP2 and LSP3 relative to WT1 and a modest increase in LSP4 compared to WT1 (Fig. 5F). These observations of elevated DLK1 protein levels (Fig. 5G,H), cNotch (Fig. 5G,I) and *HES5* transcript (Fig. 5J) were also seen in OCRL^KO compared to WT2.

DLK1, a transmembrane protein, undergoes conserved proteolytic cleavage to release its extracellular domain, a process documented in neural development, oligodendrocyte differentiation and mylination (Falix et al, 2012). To test whether the levels of secreted DLK1 were higher in OCRL^KO due to increased Notch signaling, we assessed the levels of secreted levels of DLK1 in WT2 and OCRL^KO conditioned media. We found that mean intensity of DLK1 was indeed higher in OCRL^KO than WT2 conditioned media (Fig. 5K). Overall, these data demonstrate that OCRL deficiency leads to upregulation of Notch-DLK1 signaling components, consistent with our proposed mechanism linking PI(4,5)P$_2$ dysregulation to altered neural stem cell fate decisions.

## Elevated PI(4,5)P$_2$ levels underpin the altered cell fate composition and neuronal excitability in OCRL deficient cultures

As an enzyme, OCRL dephosphorylates PI(4,5)P$_2$ to generate PI4P and OCRL deficient cells show elevation of PI(4,5)P$_2$ levels (Fig. 2E). To test if this elevation of PI(4,5)P$_2$ levels is important for the observed phenotypes, we pharmacologically inhibited the enzyme, phosphatidylinositol 4 phosphate 5-Kinase (PIP5K) that is responsible for

synthesizing the majority of PI(4,5)P$_2$. The human genome PIP5K is encoded by three genes PIP5KA, PIP5KB and PIP5KC (van den Bout and Divecha 2009). We differentiated WT2 and OCRL^KO NSC into neurons and treated the cultures for 7 days with an inhibitor for PIP5KC (UNC3230) (Wright et al, 2015) (hereafter referred to as PIP5K1C$_i$), and compared to neurons which were treated with DMSO as vehicle controls (Fig. 6A). Compared to DMSO treated WT2 neurons, DMSO treated OCRL^KO neurons consistently showed a significant reduction in the frequency of calcium transients (Fig. 6B). However, following treatment with PIP5K1C$_i$, the frequency of [Ca$^{2+}$]$_i$ transients in OCRL^KO neurons was elevated to levels seen in WT2 derived cultures (Fig. 6B). Thus, highlighting that depleting PI(4,5)P$_2$ levels in OCRL^KO neurons restores neuronal excitability to that of WT2 neurons.

Using brain organoids, we tested the effect of treatment with PIP5K1C$_i$ on the protein levels of cNotch and DLK1 (Fig. 6C). We observed no significant changes in cNotch levels in WT2 organoids treated with the PIP5K1C$_i$ compared to WT2 vehicle control, while OCRL^KO organoids show a reduction in DLK1 levels upon inhibitor treatment compared to OCRL^KO vehicle control organoids (Fig. 6D,E). Similarly, the elevated levels of DLK1 in OCRL^KO organoids were reduced upon treatment with PIP5K1C$_i$, while no significant differences were observed among WT2 and WT2 organoids treated with PIP5K1C$_i$ (Fig. 6D,F). Lastly, a similar trend was observed with respect to *HES5* transcript. In the OCRL^KO organoids treated with PIP5K1C$_i$ the levels of *HES5* were downregulated relative to OCRL^KO, whereas no difference was observed between WT2 and WT2 organoids treated with PIP5K1C$_i$ (Fig. 6G). Together, these findings suggest that OCRL deficiency, leading to elevated PI(4,5)P$_2$ levels, selectively perturbs Notch pathway activity in LS brain organoids, leading to hyperactive Notch signaling and subsequent upregulation of its transcriptional effector *HES5* and the non-canonical Notch ligand DLK1.

## Discussion

During brain development, differentiating neurons progressively develop activity that can be monitored in experimental system as [Ca$^{2+}$]$_i$ transients; these transients have been shown to be critical for a

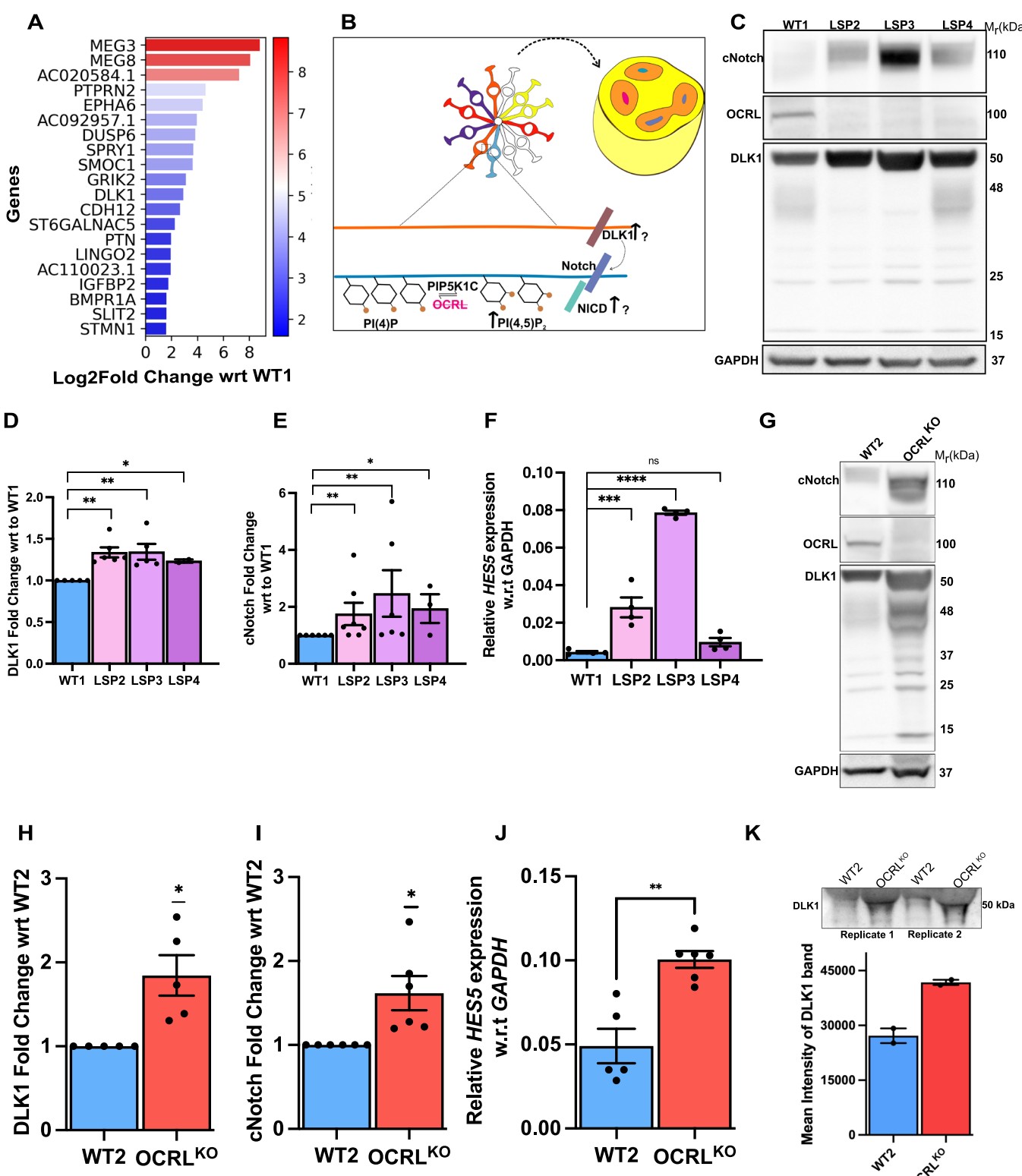

number of neurodevelopmental events such neurite outgrowth and neurotransmitter specification (reviewed in Rosenberg and Spitzer, 2011). In the case of LS, the cellular and physiological changes in the developing brain that manifest with clinical features of altered higher mental function are not known. In this study, using several independent LS iPSC derived disease-in-a-dish models, we found that neuronal cultures derived from LS patients showed reduced numbers of $[Ca^{2+}]_i$ transients and inward currents underscoring the reduced neuronal activity. Thus, it is likely that as the brain of LS patients develops in utero, neuronal activity is also reduced.

**Figure 5. Altered Notch signaling in OCRL deficient organoid cultures.**

(A) Bar plot showing the top differentially expressed genes (DEGs) identified through pseudobulk analysis of single nuclei RNA-seq data between aggregated LSP (LSP2 and LSP3) and WT1 samples. The x-axis displays the Log2 Fold Change values in LSP w.r.t. WT1 samples, while the y-axis lists the top 20 enriched genes in LSP samples that satisfied a stringent significance criterion of log2 fold change >1.5 and adjusted *p*-value < 0.01. A color gradient from blue (lower fold changes) to red (higher fold changes) indicates the magnitude of differential gene expression. (B) Diagrammatic representation of proposed model summarizing the hypothesized molecular and cellular mechanisms involved in OCRL deficient neural models. (C) Western blot with WT1, LSP2, LSP3 and LSP4 lysates obtained from 25 DIV brain organoids: OCRL (100 kDa), cleaved notch (cNotch, 110 kDa), DLK1 (50 kDa, 48 kDa) is shown; and GAPDH (37 kDa) was used as a loading control. (D) Quantification of DLK1 (50 kDa and 48 kDa) levels in 25DIV LSP brain organoids wrt to WT1. LSP2 wrt to WT1 $p = 0.0043$**; LSP3 wrt to WT1 $p = 0.0079$**; LSP4 wrt WT1 $p = 0.0467$*. (E) Quantification of cNotch levels in 25DIV LSP brain organoids relative to WT1. LSP2 wrt to WT1 $p = 0.0012$**; LSP3 wrt to WT1 $p = 0.0022$**; LSP4 wrt WT1 $p = 0.0119$*. (D, E) Each point represents 10–15 pooled organoids, across ($n = 5$) independent differentiations, statistical test: Mann-Whitney t-test. (F) RT-PCR analysis for 25 DIV brain organoids from WT1, LSP2, LSP3, LSP4 for the cNotch downstream effector *HES5* normalized to the housekeeping gene *GAPDH*. Each dot represents 15–20 brain organoids generated from independent batches of differentiation ($n = 5$). Statistical test: one-way ANOVA followed by Dunnett's multiple comparisons test. LSP2 wrt WT1 $p = 0.0002$***; LSP3 wrt WT1 $p = <0.0001$****. (G) Western blot from WT2 and OCRL^KO 25 DIV brain organoids for DLK1 and cNotch. GAPDH was used as a loading control. (H) Quantification of DLK1 bands is shown for WT2 and OCRL^KO brain organoids. Each point represents 15–20 brain organoids generated from multiple batches of differentiations ($n = 5$). Statistical test: one-sample t-test $p = 0.0248$*. (I) Quantification of cNotch bands for WT2 and OCRL^KO brain organoids ($n = 5$). Statistical test: one-sample t-test $p = 0.0287$*. (J) *HES5* transcripts relative to the housekeeping gene GAPDH quantified and plotted for WT2 and OCRL^KO brain organoids ($n = 5$). Statistical test: Unpaired Mann-Whitney t-test, $p = 0.0043$**. Error bars in all the dataset presented here represent Mean $+/-$ SEM. (K) Western blot for secreted DLK1 (sDLK1, 50 kDa) obtained from medium of WT2 and OCRL^KO brain organoid cultures maintained till 25 DIV. Equal amount (40 μg) of protein was loaded per sample. Quantification shown below (K) represents mean intensity of DLK1 band for two independent biological replicates ($n = 2$).

To establish causality independent of patient-specific genetic variations, we introduced a patient-mimetic truncating mutation in *OCRL* exon 8 into wild-type iPSCs from a Caucasian background (NCRM5/WT2 iPSC) using CRISPR-Cas9 editing, resulting in complete absence of the protein. This genetic manipulation recapitulated the reduced neuronal activity observed in LSP-derived cultures, confirming OCRL deficiency as the direct driver of neuronal dysfunction across distinct populations. A critical consideration in this study was the potential impact of genetic background on phenotypic outcomes. It is well established that genetic background significantly influences iPSC-differentiated cell phenotypes (Volpato and Webber, 2020; Anderson et al, 2021; Brunner et al, 2023). To overcome this confounding factor, we implemented a dual-control strategy: (1) ethnically matched controls (WT1) for patient-derived lines (LSP1-4), and (2) isogenic controls (WT2) for the CRISPR-generated OCRL^KO line. When directly compared, WT1 and WT2 showed no significant differences in frequency of calcium transients (Fig. EV2E). Furthermore, pooled analysis of all control lines (WT1 + WT2) versus disease lines (LSP1-4 + OCRL^KO) confirmed persistent phenotypic deviations in OCRL-deficient cells (Fig. EV2F). This consistency across distinct genetic backgrounds underscores that observed neurodevelopmental defects are indeed driven by loss of OCRL rather than confounding genetic variation. These findings underscore the critical role of OCRL in the development of neuronal activity in the brain.

Mechanistically, OCRL is a 5'-lipid phosphatase, and loss of OCRL elevates PI(4,5)P$_2$ levels, which is consistent with findings in LS patient cells (Wenk et al, 2001; Akhtar et al, 2022) and in our OCRL^KO model (Fig. 2, this study). Importantly, the reduced neuronal activity could be rescued by treating OCRL^KO cells with an inhibitor of PIP5KC enzyme that synthesizes PI(4,5)P$_2$. Taken together, these findings strongly suggest that the elevated PI(4,5)P$_2$ levels lead to reduced neuronal activity during brain development. PI(4,5)P$_2$ regulates several key molecules involved in regulating neuronal excitability including ion channels and transporters (Hille et al, 2015; Dickson and Hille, 2019) and also synaptic vesicle cycle proteins (Koch and Holt, 2012). Altered function of these molecules could underlie the reduced excitability observed in developing OCRL deficient neurons.

Brain development can be conceptualized as a series of cell specification events followed by differentiation into neural cell types forming circuits and develop activity and function. At which stage of this process is OCRL required for normal activity development? During our experiments, we noted that neural cultures derived from OCRL deficient iPSC expressed higher levels of GFAP that marks mature astrocytes. Increased numbers of astrocytes have previously been noted in other neurodevelopmental disorders including autism (Trudler et al, 2024), Tuberous Sclerosis Complex (Li et al, 2025) and Down's syndrome (Chen et al, 2014). While there are no studies on the histology of the developing brain from LS patients to directly corroborate our finding in human patients, a previous analysis of LS using the zebrafish model have revealed features suggestive of increased focal gliosis when OCRL is deficient (Ramirez et al, 2012). This finding of increased expression of astroglia markers in LS patient derived neural cultures raises the possibility these astrocytes may be responsible for the reduced neuronal activity noted in these cultures. A number of mechanisms by which astrocytes can modulate neuronal activity have been described including modulating calcium signaling, neurotransmitter re-uptake, gliotransmitter release, synapse formation and pruning (Allen and Eroglu 2017; Verkhratsky et al, 2021; Khakh 2025). In principle, these could be altered in OCRL deficient cultures leading to a non-cell autonomous impact on neuronal activity. Future experiments will be needed to address such mechanisms.

Why do LS patient derived cultures show an enhanced expression levels of GFAP during development in vitro? Through our single nuclei multiome analysis of NSC, we found that compared to controls, NSC derived from LS patients contained a larger proportion of cells with transcripts and signatures of astrocytes and glioblasts than neurons when compared to control WT1 (Fig. 4C). We observed variability within the LSP2 and LSP3 NSCs in their snRNA seq clustering profiles (Fig. 4B,C). This variability could potentially reflect the technical variability well reported in the field (Stegle et al, 2015; Baysoy et al, 2023), or biological diversity where cells transition along a continuum rather than distinct cell types (Heumos et al, 2023). Notably, through RT-PCR analysis of all LSP NSCs generated across multiple neural inductions, we observed that loss of OCRL led to an elevation in transcript levels of glial lineage markers *NFIA* and *OLGI2*

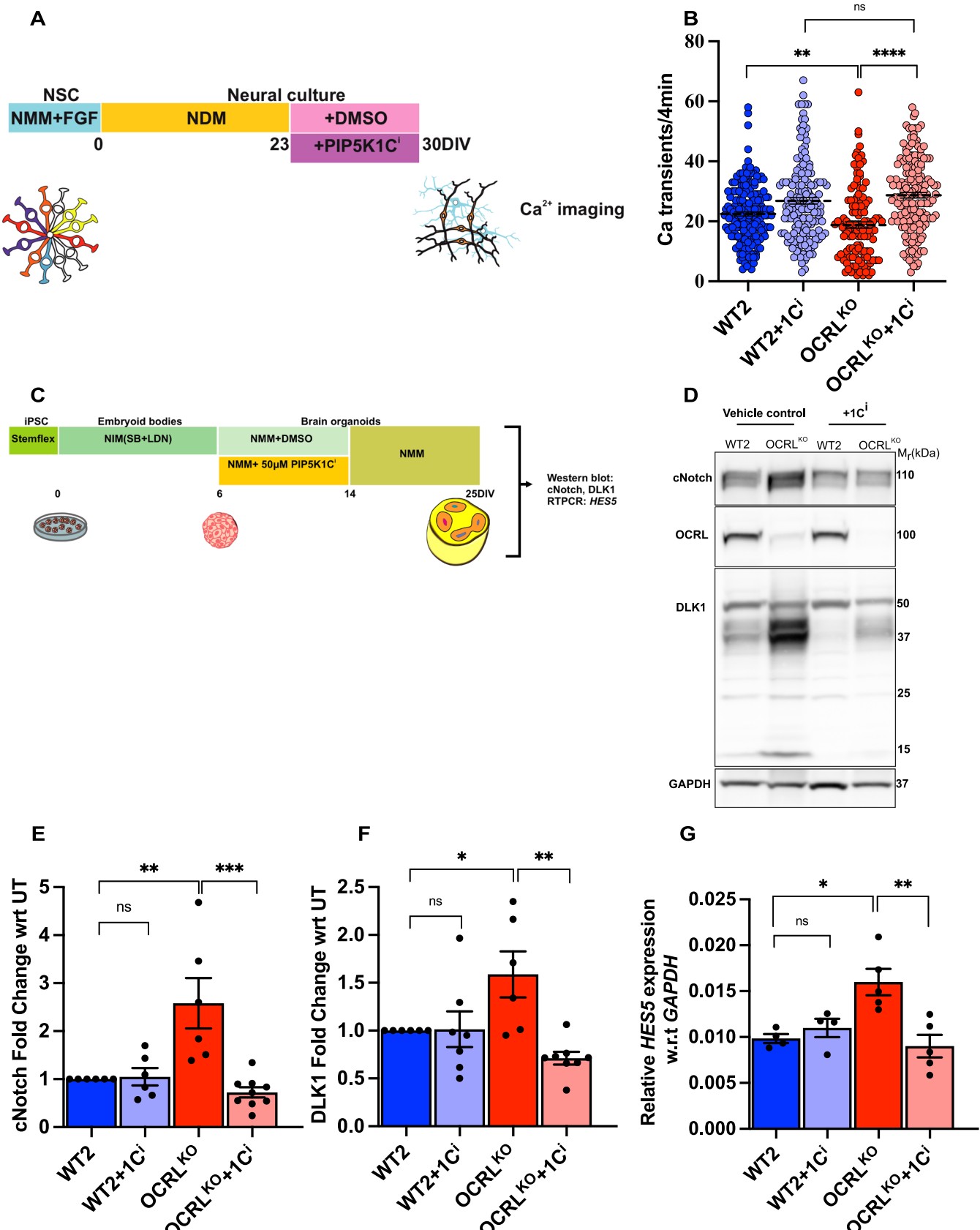

**Figure 6. Rescue of phenotypes in OCRL deficient cultures by inhibiting PI(4,5)P$_2$ synthesis.**

(A) Schematic of experimental design for testing the effect of UNC-3230, a PIP5KC inhibitor (PIP5KC$_i$) on [Ca$^{2+}$]$_i$ transient frequency in 30DIV WT2 and OCRL$^{KO}$ neurons. (B) Quantification of [Ca$^{2+}$]$_i$ transients in 30 DIV WT2, OCRL$^{KO}$ neurons treated with DMSO (vehicle control) and WT2 and OCRL$^{KO}$ treated with PIP5K1C$_i$ (50 μM) for a duration of 7 days in vitro. Y-axis represents the frequency of [Ca$^{2+}$]$_i$ transients. Each point represents [Ca$^{2+}$]$_i$ transients obtained from a single neuron, from independent differentiations ($n = 2$). Statistical test: Unpaired t-test Welch's correction, WT2 vs OCRL$^{KO}$ (DMSO treated) $p = 0.0042^{**}$; OCRL$^{KO}$ (DMSO treated) vs OCRL$^{KO} + 1$C$^i$ $p = <0.0001^{****}$. (C) Schematic of experimental design for testing the effect of PIP5KC$_i$ on Notch signaling in brain organoids. (D) Western blot showing presence of OCRL in WT2 and levels of cNotch and DLK1 in 25DIV organoids from WT2, OCRL$^{KO}$ treated with DMSO (vehicle control), and WT2, OCRL$^{KO}$ treated with PIP5K1C$_i$ (50 μM). GAPDH is used as a loading control. (E) Quantification of cNotch fold change w.r.t. to vehicle control is shown ($n = 3$). (F) DLK1 protein levels presented as fold-change w.r.t. to vehicle control condition, respectively ($n = 3$). Statistical test for (E, F): Mann–Whitney test. For cNotch, OCRL$^{KO}$ w.r.t. WT2 (DMSO treated) $p = 0.0022^{**}$; OCRL$^{KO}$ w.r.t OCRL$^{KO} + 1$ C$^i$ $p = 0.0004^{***}$. For DLK1, OCRL$^{KO}$ w.r.t. WT2 (DMSO treated) $p = 0.0476^*$; OCRL$^{KO}$ wrt OCRL$^{KO} + 1$ C$^i$ $p = 0.0027^{**}$. (G) Transcript levels of *HES5* measured using RTPCR for WT2, OCRL$^{KO}$ 25DIV organoids treated with PIP5K1C$_i$ presented relative to *GAPDH* transcripts. Each point shown is a single RNA sample from pooled 15-20 organoids ($n = 5$). Statistical test: Mann–Whitney test; OCRL$^{KO}$ w.r.t. WT2 (DMSO treated) $p = 0.0159^*$; OCRL$^{KO}$ wrt OCRL$^{KO} + 1$C$^i$ $p = 0.0079^{**}$. Error bars in all the dataset presented in this figure represent Mean $+/-$ SEM.

(Fig. EV3G,H) irrespective of the LSP lines. This highlights that the gliogenic bias is present not only at single cell level but also at a global level. Furthermore, our 3D brain organoid culture system also recapitulated the gliogenic phenotypes (Fig. EV5G–I), thus supporting the robustness and generalizability of this phenotypes. Collectively, these findings suggests that in LS cultures loss of OCRL function may lead to an early developmental switch that results in an increased number of glial precursor cells which in turn leads to increased levels of GFAP noted when these are differentiated. Notably, while OCRL expression peaks in mature neurons (Fig. 1I,J), loss of OCRL at NSC stage exerts a disproportionate impact on neurodevelopment. Our data reveals that even low OCRL levels in NSC crucially regulate cell-fate decisions through PI(4,5)P$_2$-dependent Notch hyperactivation. This triggers early gliogenic bias via *HES5* upregulation and propagates defects downstream, yielding increased expression of GFAP. This disturbs the temporal neurodevelopmental progression in LSP neural cultures and thereby leading to a defective neuronal excitability. In contrast, while control iPSCs express OCRL (Fig. 1I), loss of OCRL at the pluripotent stage, as seen in LSP iPSCs does not result in overt phenotypic abnormalities. We attribute this stage-specific vulnerability to compensatory upregulation of other 5-phosphatases at the pluripotent stage, as demonstrated by elevated PI(4,5)P$_2$ levels and phosphatase redundancy in our prior work (Akhtar et al, 2022). Thus, NSC-stage perturbations, though occurring at lower OCRL expression establish aberrant developmental trajectories that potentially results in the neuronal excitability defects observed in 30 DIV neural cultures (Fig. 2C,H,J,K).

The ratio of neurons to glia cells in any brain is precisely set in animal species (Herculano-Houzel, 2014). Both neurons and glia are specified during development through signaling mechanisms including the evolutionarily conserved Notch signaling pathway (Hori et al, 2013; Zhang et al, 2018). In this study, we found that DLK1 levels were elevated in LSP NSC along with evidence of elevated Notch signaling in *OCRL* deficient brain organoid cultures. These observations suggest that OCRL via PI(4,5)P$_2$ regulates Notch signaling, thus altering cell-fate specification during brain development. How might this occur? Critically, pharmacological inhibition of PIP5K1C (UNC-3230) to reduce PI(4,5)P$_2$ levels rescues the Notch signaling defects in OCRL$^{KO}$ implying a role for elevated PI(4,5)P$_2$ levels in this phenotype. However, the precise mechanistic link between increased PI(4,5)P$_2$ and hyperactive Notch signaling remains to be resolved. Notably, it has been

reported that altered PI(4,5)P$_2$ levels may indirectly modulate Notch cleavage via defective endocytosis, as demonstrated in *Drosophila* photoreceptors and wings, where phospholipid composition regulates Notch trafficking and activation (Weber et al, 2003; Skwarek and Boulianne, 2009). Previous studies have shown that the cleavage of Notch by the γ-secretase enzyme is a process influenced by PI(4,5)P$_2$ (Osawa et al, 2008; Osenkowski et al, 2008; Yan et al, 2017), thus providing a potential molecular link between the elevated cNotch levels and the enhanced PI(4,5)P$_2$ in OCRL deficient cells. Further, in mouse models, loss of DLK1 shows a dose dependent effect on hippocampal neurogenesis and these mice show defects in cognitive behavior (Ferrón et al, 2011; Montalbán-Loro et al, 2021). Together these findings suggest that during the developmental transition from pluripotent stem cells to NSC, *OCRL* depletion leads to enhanced Notch signaling. Therefore, this may be the reason why NSC derived from OCRL deficient stem cells show enhanced levels of gliogenic precursors.

Overall, our study provides insights into the cellular and developmental mechanism by which OCRL loss in LS patients may lead to cognitive deficits evident at birth and during early childhood. Our description of a cellular phenotype in a "disease-in-a-dish" model along with the finding the pharmacological inhibition of PIP5KC can reverse this phenotype opens avenues for screening for compounds, especially those with prior FDA approvals for those that can reverse the cellular phenotypes. Such an approach can rapidly accelerate the development of suitable therapeutics for clinical management of LS patients.

## Methods

**Reagents and tools table**

| Reagent/Resource | Reference or Source | Identifier or Catalog Number |
| --- | --- | --- |
| **Experimental models** | | |
| WT1 iPS cell line | Iyer et al, 2018; generated in-house | RRID:CVCL_VN36 |
| Lowe Syndrome Patient 2 (LSP2) iPS cell line | Akhtar et al, 2022; generated in-house | |
| Lowe Syndrome Patient 3 (LSP3) iPS cell line | Akhtar et al, 2022; generated in-house | |
| Lowe Syndrome Patient 4 (LSP4) iPS cell line | Akhtar et al, 2022; generated in-house | |
| WT2/NCRM5 iPS cell line | NINDS Human Cell and Data Repository | RRID: CVCL_1E75 |

| Reagent/Resource | Reference or Source | Identifier or Catalog Number |
|---|---|---|
| OCRL[KO] iPS cell line | This study | |
| **Recombinant DNA** | | |
| pSpCas9(BB)-2A-GFP (PX458) | Addgene | 48138 |
| pCXLE-EGFP | Addgene | 27082 |
| **Antibodies** | | |
| **Primary antibodies** | | |
| Mouse anti-Nestin (1:100) | Abcam | ab22035 |
| Rabbit anti-Pax6 (1:200) | BioLegend | 901301 |
| Rabbit anti-Foxg1 (1:200) | Abcam | ab18259 |
| Rabbit anti-GABA (1:200) | Sigma-Aldrich | A2052 |
| Rat anti-CTIP2 (1:800) | Abcam | ab18465 |
| Rabbit anti-BRN2 (1:200) | Cell Signalling Technologies | 12137 |
| Rabbit anti-TBR1 (1:200) | Abcam | ab31940 |
| Chicken anti-GFAP (1:1000, IF; 1:2000 WB) | Novus Biologicals | NBP1-05198 |
| Rabbit anti-S100β (1:1000) | Abcam | ab52642 |
| Rabbit anti-Cleaved-Notch (1:500) | Cell Signalling Technologies | CST- 4147S |
| Rabbit anti-DLK1 (1:500) | Abcam | ab21682 |
| Rabbit anti-GAPDH (1:1000) | Novus Biologicals | NB100-56875 |
| Mouse anti-OCRL (1:300) | Santacruz | sc393577 |
| Rabbit anti-TBR2 (1:200) | Abcam | ab23345 |
| Rabbit anti-Synapsin-1 (1:500) | Abcam | ab64581 |
| Rabbit anti-Ki-67 (1:100) | Abcam | CST-9449S |
| Chicken anti-MAP2 (1:1000) | Novus Biologicals | NB300-213 |
| **Secondary antibodies** | | |
| Alexa Fluor™ 488 (1:300) | Invitrogen | A11034 |
| Alexa Fluor™ 568 (1:300) | Invitrogen | A11041 |
| Alexa Fluor™ 488 (1:300) | Invitrogen | A11001 |
| Alexa Fluor™ 488 (1:300) | Invitrogen | A11006 |
| Alexa Fluor™ 568 (1:300) | Invitrogen | A10042 |
| Alexa Fluor™ 568 (1:300) | Invitrogen | A11004 |
| **Cell culture reagents** | | |
| hESC qualified Matrigel | Corning | 354277 |
| Poly-L-Ornithine | Merck | A-004-C |
| Laminin | Gibco | 23017015 |
| StemPro Accutase | Gibco | A11105-01 |
| Revita Cell Supplement | Gibco | A26445-01 |
| Neurobasal | Gibco | 21103049 |
| DMEM/F12+Glutamax | Gibco | 10565018 |
| Insulin | Sigma-Aldrich | 11061-68-0 |
| B27 with vitamin A | Gibco | 17504044 |

| Reagent/Resource | Reference or Source | Identifier or Catalog Number |
|---|---|---|
| N2 supplement | Gibco | 17502048 |
| Betamercaptoethanol | Gibco | 21985-023 |
| Sodium pyruvate | Gibco | 11360-070 |
| Non-Essential Amino Acids | Gibco | 11140-050 |
| Penstrep | Gibco | 15140-122 |
| Glutamax | Gibco | 35050-061 |
| SB-43142 | Sigma-Aldrich | 301836-41-9 |
| LDN | Sigma-Aldrich | SML0559 |
| FGF2 | Gibco | PHG0369 |
| EGF | Gibco | PHG0311 |
| BDNF | Gibco | PHC7074 |
| BDNF | Peprotech | 450-02 |
| GDNF | Gibco | PHC7041 |
| NT3 | Peprotech | 450-03 |
| Ascorbic acid | Sigma-Aldrich | A4544 |
| dbcAMP | Sigma-Aldrich | D0627 |
| DAPT | Sigma-Aldrich | D5942 |
| AggreWell 800 plate | STEMCELL Technologies | 34811 |
| Neurobasal-A | Gibco | 10888022 |
| B-27 (vitamin A-free) | Gibco | 12587010 |
| B27 plus supplement | Gibco | A3582801 |
| **Oligonucleotides and other sequence-based reagents** | | |
| OCRL sgRNA (G1) | Sigma-Aldrich | 5′-AAACAATACCCAATC TGGGCAGC-3′ |
| OCRL sgRNA (G2) | Sigma-Aldrich | 5′-CACCGGGTCTCA TCAAACATATCC-3′ |
| Gibson forward primer | Sigma-Aldrich | 5′-ctgcagacaaatggctctaga GAGGGCCTATTTCCCATG-3′ |
| Gibson reverse primer | Sigma-Aldrich | 5′-agttatgtaacgggtacc GCCATTTGTCTGCAGAATTG-3′ |
| OCRL amplifying forward primer | Bioserve | 5′-TCAAAGCCCT GTTACCCTGG-3′ |
| OCRL amplifying reverse primer | Bioserve | 5′-GACAGGAGCTT GAAACAGGCA-3′ |
| **RT-PCR primers** (5′ to 3′) | | |
| *GAPDH* forward primer | | TGCACCACCAACTGCTTAGC |
| *GAPDH* reverse primer | | GGCATGGACTGTGGTCATGAG |
| *GFAP* forward primer | | AGAACCGGATCACCATTCCC |
| *GFAP* reverse primer | | CACGGTCTTCACCACGATGT |
| *HES5* forward primer | | CATCAACAGCAGCATCGAGC |
| *HES5* reverse primer | | TGCTTCAGGTAGCTGACAGC |
| *MAP2* forward primer | | GCGCCAATGGATTCCCATAC |
| *MAP2* reverse primer | | CAGACACCTCCTCTGCTGTT |
| *NFIA* forward primer | | ACAGGTGGGGTTCCTCAATC |
| *NFIA* reverse primer | | GTGGGACGCTGCAACTTTT |
| **Softwares** | | |
| Fiji | https://imagej.net/software/fiji/ | |
| CellSens Dimensions software (Olympus, build 16686) | https://evidentscientific.com/en/products/software/cellsens | |
| Python | | |
| Scanpy | https://scanpy.readthedocs.io/en/stable/ | |

| Reagent/Resource | Reference or Source | Identifier or Catalog Number |
|---|---|---|
| SCENIC+ | https://scenicplus.readthedocs.io/en/latest/ | |
| pycisTopic | https://pycistopic.readthedocs.io/en/latest/tutorials.html | |
| Clampfit 11.1 | | |
| snapseed | https://github.com/devsystemslab/snapseed | |
| SnapATAC2 | https://scverse.org/SnapATAC2/tutorials/index.html | |

## Maintenance of iPSCs

The generation and use of human iPS cell lines was approved by National Centre for Biological Sciences Institutional Human Ethics Committee (NCBS/ICE-8/002/31E). The generation and characterization of LS patient iPSC has been previously described (Akhtar et al, 2022). hiPSC were maintained StemFlex medium (Gibco, # A3349401) and maintained on hESC-qualified Matrigel-coated surface (Corning # 354277). They were routinely passaged with EDTA (Sigma #E8008) and frozen stocks were prepared using PSC cryomix (Gibco, #A26444-01). The cultures were regularly checked for mycoplasma using the Lonza mycoplasma detection kit (# LT07-318).

## Cloning of sgRNAs

Two sgRNAs (OCRL-688-G1, 5′-AAACAATACCCAATCTGGG-CAGC-3′ and OCRL-688-G2, 5′-CACCGGGTCTCATCAAACA-TATCC-3′) were designed to target OCRL exon 8. The two sgRNA fragments under the control of their independent U6 promoters were cloned into the plasmid PX458, pSpCas9(BB)-2A-GFP, (Addgene #48138) by Gibson assembly. First, the annealed OCRL-688-G1 was ligated into the linearized PX458 vector using the Bbs1 site. The confirmed clone of PX458-OCRL-688-G1 was then digested with Xba1 and Kpn1 to produce a linearized vector for Gibson assembly. For generation of the insert, the U6-OCRL-688-G2 cassette was amplified from a previously generated clone of PX461-OCRL-688-G2 using custom primers (GA_FP, 5′-ctgcaga-caaatggctctagaGAGGGCCTATTTCCCATG-3′ and GA_RP, 5′-agttatgtaacgggtaccGCCATTTGTCTGCAGAATTG-3′) to introduce a spacer into the amplicon. The U6-OCRL-688-G2 cassette was then inserted into the PX458-OCRL-688-G1 vector using NEBuilder HiFi DNA Assembly Cloning Kit (New England Biolabs), according to the manufacturer's instructions. The resulting plasmid, PX458-OCRL-688-G1-G2, was purified and presence of both sgRNAs in the final CRISPR-Cas9 vector was confirmed by Sanger sequencing.

## Generation of OCRL knockout iPSC line

The OCRL knockout iPSC line (iPSC-OCRL$^{KO}$) was generated using CRISPR-Cas9 gene editing in the iPSC wildtype cell line NCRM5/WT2 (Reagents and Tools table). Two sgRNAs, OCRL-688-G1 and OCRL-688-G2, were designed to target OCRL exon 8, based on the c.688 C > T truncating mutation identified in the LS

patients. WT2 cells ($3 \times 10^6$) were transfected with the assembled CRISPR-Cas9 vector, PX458-OCRL-688-G1-G2, (18 µg) using the Neon transfection system and Neon transfection 10 µL kit (Invitrogen Cat # MPK1025). Briefly, cells pre-incubated in fresh E8 Complete media with 1x Revita Cell Supplement for 2 h were harvested by enzymatic dissociation using StemPro Accutase, for 5 min at 37 °C. After neutralizing Accutase with the spent media, the cells were spun down at 1200 RPM for 3 min at room temperature. The cells were resuspended in buffer-R such that each 10 µL hit contained 3 µg of the CRISPR-Cas9 vector and $0.5 \times 10^6$ cells. The cell suspension was electroplated using the Neon Electroporation System as per manufacturer's protocol under the following standardized parameters (1100 V, 20 ms width, 3 pulses) and plated in a hESC-qualified Matrigel matrix coated 35 mm plate pre-incubated with fresh E8 Complete media and Revita Cell Supplement. A complete media change was performed 4 h post electroporation to remove dead cells and debris. After 24 h, the cells were checked for GFP expression and were subjected to fluorescence-activated cell sorting (FACS) to enrich for GFP-positive cells. The mixed cell culture was enzymatically dissociated using StemPro Accutase, washed once with PBS and resuspended in media with Penicillin Streptomycin at a concentration of $2 \times 10^6$ cells per mL. The cells were sorted using FACS-Aria Fusion (BD Biosciences). Forward and side scatter parameters were adjusted to eliminate cell clumps and debris. The gating parameter threshold for GFP-positive cells was set using control cells electroplated with pCXLE-EGFP. The GFP-positive cells were plated in a pre-incubated Matrigel coated 60 mm dish. Clonal expansion of isolated colonies was performed manually. gDNA was isolated from stable colonies using QIAamp DNA Mini Kit (Qiagen Cat # 51304) and the target region was amplified using specific primers (OCRL_E8_PCR_F, 5′-TCAAAGCCCTGTTACCCTGG-3′ and OCRL_E8_PCR_R, 5′-GACAGGAGCTTGAAACAGGC-3′). The PCR product was purified and clones with the desired out-of-frame indel modifications were identified by Sanger Sequencing.

## Generation of neural progenitor cells from iPSCs and terminal differentiation into neurons

Differentiation of forebrain cortical neurons from iPSC was carried out as per the published protocol (Shi et al, 2012). Briefly, to generate dorsal forebrain neural progenitor cells, iPS cells after 3 passages were plated on 6-well tissue culture plates coated with Matrigel (hESC qualified). When cells reach 100% confluency, medium was replaced with neural induction (NI) medium marking 0 days in vitro (DIV) and maintained for 10 DIV. The NI medium consisted of Neural Maintainence Media (NMM): (1:1) Neurobasal and DMEM/F12+Glutamax, insulin (2.5 µg/ml), B27 with vitamin A (0.5X), N2 supplement (0.5X), betamercaptoethanol, sodium pyruvate (1 mM), non-essential amino acids (NEAA, 0.5X), penstrep (1000 U/ml), Glutamax (1 mM), supplemented with SB-43142 (10 µM), and LDN (200 nM). On 10th day, cells were dissociated using dispase (Life Technologies) and resuspended in NI medium with 1X revita supplement. Cells were plated on air-dried poly-L-ornithine and 10ug/ml laminin-coated in 6-well plates. The following day, NIM was replaced with neural maintenance (NM) medium supplemented with 20 ng/ml FGF2, which was added for 4–6 days. Lastly, NPCs were frozen at ~1 million density in PSC cryomix. For terminal differentiation of

NPCs into cortical neurons, ~400,000–500,000 NPCs were plated in glass bottom dishes and 6-well dishes coated with poly-L-ornithine/laminin-coated plates and maintained in NM media supplemented with 20 ng/ml BDNF, 20 ng/ml GDNF, ascorbic acid (200 µM), and dbcAMP (50 µM). During the first 14 days after plating, cells were treated with 10 µM DAPT to synchronize the neuronal maturation process.

## Generation of dorsal forebrain organoids

Brain organoids were generated from iPSC using the established protocol (Sloan et al, 2018). Briefly, iPSC were dissociated with accutase to obtain a single-cell suspension. Approximately $2.5–3 \times 10^6$ cells were placed in each well of an AggreWell 800 plate containing Stemflex medium enriched with 1X Revita supplement to allow formation of embryoid bodies (EBs). The plate underwent centrifugation at $100 \times g$ for 3 min and was then incubated at 37 °C with 5% $CO_2$. The following day, fresh Stemflex was supplemented without Revita supplement. The EBs were maintained in Stemflex media for another day or two.

### Neural induction

After 48–72 h, using 1 ml cut-tips EBs were transferred to ultra-low attachment dishes (6-well or 60 mm and 100 mm) and cultured in the neural induction medium (NIM) containing Essential 6 medium supplemented with LDN (200 nM) and SB-431542 (10 µM). This day denoted as '0 DIV'. To prevent spontaneous fusion of EBs, dishes were transferred to shaker from day 0 onwards with orbital speed of 65 rpm. From days 2 to 5, organoids were maintained in NIM.

### Expansion of neural progenitor cells

On day 6, organoids were moved to a neural maintenance medium (NMM) composed of Neurobasal A (Gibco #10888022), B-27 supplement (vitamin A-free), GlutaMax (1:100), and penicillin-streptomycin (1:100). This medium was further supplemented with EGF (20 ng/ml) and FGF2 (20 ng/ml) from day 6 to 24. Medium changes were performed daily for the initial 10 days, followed by every other day for the subsequent 9 days.

### Neuronal maturation

From day 25 to day 43, EGF and FGF were withdrawn from the NMM, and was replaced with 20 ng/ml BDNF and 20 ng/ml NT-3. From day 44, only NMM without growth factors was used for medium changes every 4 days and B27 (vitamin A-free) was replaced with B27 plus supplement.

### Cryopreservation, sectioning, IHC of organoids and ICC for monolayer cultures

90 DIV Organoids were fixed in 4% PFA-PBS overnight at 4 °C followed by dehydration in 30% sucrose for 24–48 h. Subsequently, samples were embedded in optimal cutting temperature (OCT) compound (Tissue-Tek OCT Compound 4583, Sakura Finetek) and 30% sucrose–PBS (1:1) for cryo-sectioning (20-µm-thick sections) using a Leica Cryostat (Leica, catalogue no. CM1860). For immunofluorescence staining, cryosections were washed with PBS to remove excess OCT. Sections were first permeabilised with 0.5% Triton X-100 diluted in 1X phosphate-buffered saline (PBS with $Ca^{2+}/Mg^{2+}$) for 30 min at room temperature (RT). The sections were blocked in 5% BSA containing 0.1% Triton X-100 diluted in PBS for 1 h at RT. Sections were then incubated overnight at 4 °C with primary antibodies diluted in PBS containing 5% BSA. PBS was used to wash away excess primary antibodies, and the cryosections were incubated with secondary antibodies in PBS containing 5% BSA for 1 h. The nuclei were visualized with DAPI. Primary antibodies used for IHC: GFAP, S100β (please refer to Reagents and Tools table for dilutions).

NSC and neuronal cultures were fixed using 4% formaldehyde in PBS for 20 min and permeabilized using 0.1% TX-100 for 5–10 min. These cultures were incubated at room temperature for 1 h in a blocking solution of 5% BSA in PBS. Primary antibodies at respective dilutions were added and incubated overnight at 4 °C in blocking solution, followed by incubation with secondary antibodies in blocking solution (Invitrogen) for 1 h. List as well of dilutions of primary and secondary antibodies used are provided in the Reagents and Tools table. Confocal images were captured using FV3000 and a range of z-stack was captured. The z-stack so obtained was merged and processed using Z-project (maximum intensity projection) function using ImageJ (National Institute of Health, USA, http://imagej.nih.gov/ij).

## Calcium imaging

Calcium imaging was according to our previously published protocol with minor modifications (Sharma et al, 2020). Briefly, neurons were washed with Tyrode's buffer solution (5 mM KCl, 129 mM NaCl, 2 mM $CaCl_2$, 1 mM $MgCl_2$, 30 mM glucose and 25 mM HEPES, pH 7.4) for 10 min and later incubated with 4 µM fluo-4/AM (1 mM, Molecular probes, #F14201) and 0.02% pluronic F-127 (Sigma-Aldrich, #P2443) in the Tyrode's buffer solution in the dark for 30–45 min at RT. Following dye loading, the cells were washed again with the buffer thrice, each wash for 5 min. Finally, cells were incubated for an additional 20 min at RT to facilitate de-esterification. $Ca^{2+}$ imaging was performed for 10 min with a time interval of 1 s at 20X objective using a wide-field fluorescence microscope Olympus IX-83. A 4-min baseline measurement was recorded to capture spotaneous calcium transients, followed by the addition of TTX to abolish calcium transients for another 4 min, and finally high KCL was added to depolarise the neurons, which served as an internal control. Calcium traces were obtained using the Imagej software by drawing a region of interest (ROI) manually around each neuronal soma. Timelapse imaging was carried out in regions where neurons were distributed uniformly across geno-types. To plot the calcium traces, the raw fluorescence intensity values from each neuron were normalized to the first fluorescence intensity signal of the baseline recording. Frequency of calcium transients was measured by counting the number of spikes above a 2 times the standard deviation above the baseline.

## Electrophysiology

Whole-cell patch clamp recordings were performed on 60DIV iPSC-derived cortical neurons. Recordings were performed at room temperature. Neurons used for all patch-clamp recordings were chosen from regions where cells were sparsely distributed. Briefly, coverslips containing the cultures were transferred to a recording chamber perfused with an external solution (in mM: 152 NaCl, 2.8 KCl, 10 HEPES, 2 $CaCl_2$, 10 glucose; pH 7.3–7.4, 300–320 mOsm).

A MultiClamp 700B amplifier was used to collect recordings, with data sampled at 10 kHz and digitized at 20 kHz via a Digidata 1550. Patch pipettes were crafted from thick-walled borosilicate glass and filled with an internal solution (in mM: 155 K-gluconate, 2 MgCl$_2$, 10 HEPES, 10 Na-PiCreatine, 2 Mg$^{2+}$-ATP, 0.3 Na3-GTP; pH 7.3, 280–290 mOsm). Series resistance was typically 5–8 MΩ. Membrane voltage was held at −70 mV. Currents were recorded using a voltage-step protocol. The membrane potential was initially held at −20 mV and then incrementally increased in 5 mV steps up to +100 mV. Stimulation protocols were designed using pClamp 10.5 software, with subsequent analysis performed offline using Clampfit 11.1 software.

## Western blotting

Cultures were harvested using Stempro Accutase and pelleted at $1000 \times g$ for 5 min then washed three times with ice-cold PBS. The pelleted cells were homogenized in 1X RIPA lysis buffer containing freshly added phosphatase and protease inhibitor cocktail (Roche). To remove cellular debris, crude RIPA lysates were centrifuged at 13,000 rpm for 20 min at 4 °C. The supernatant was transferred to a new tube and quantified with a Pierce BCA protein assay (Thermo Fisher Scientific, #23225). Thereafter, the samples were heated at 95 °C with Laemmli loading buffer for 5 min and 20 µg protein was loaded onto Bolt™ 4 to 12%, Bis-Tris SDS gel (Invitrogen, #NW04120BOX). The proteins were then transferred onto a nitrocellulose membrane and incubated overnight at 4 °C with indicated antibodies. The blots were then washed three times with Tris Buffer Saline containing 0.1% Tween-20 (0.1% TBS-T) and incubated with 1:10,000 concentration of appropriate HRP-conjugated secondary antibodies (Jackson Laboratories, Inc.) for 45 min. After three washes with 0.1% TBS-T, blots were developed using Clarity Western ECL substrate (Bio-Rad) on a GE ImageQuant LAS 4000 system.

## Mass spectrometry analysis

Lipid extraction followed by liquid chromatography-mass spectrometry was used to estimate PIP$_2$ levels. The methods used are identical to that reported in Akhtar et al (2022).

## 10x Multiomics

### Nuclei isolation from NSC

NSC were revived and nuclei were isolated as per 10x genomics recommended protocol. Briefly, approximately $1.2 \times 10^6$ NSCs were thawed, resuspended in PBS + 0.04% BSA and spun down at 1250 rpm for 5 min. The cell pellet was resuspended in lysis buffer containing Tris-HCl (pH 7.4) 10 mM, NaCl 10 mM, MgCl$_2$ 3 mM, Tween-20 0.1%, IGEPAL CA-630 0.1%, Digitonin 0.01%, BSA 1%, DTT 1 mM, RNase inhibitor (Roche) 1U/µl. The cell pellet was allowed to lyse for a period 3 min. Chilled wash buffer containing Tris-HCl (pH 7.4) 10 mM, NaCl 10 mM, MgCl$_2$ 3 mM, BSA 1%, Tween-20 0.1%, DTT 1 mM, RNase inhibitor 1U/µl was added to lysed cells and centrifuged 1250 rpm for 5 min at 4 °C. The cells were washed for two more times. Finally, the cells were resuspended in nuclei buffer containing DPBS with DTT 1 mM, and RNase inhibitor 1 U/µl. Nuclei were counted using a hemocytometer, diluted to 10,000 nuclei, and further processed

following 10x Genomics Chromium Next GEM Single Cell Multiome ATAC + Gene Expression Reagent Kits user guide. We targeted 10,000 nuclei per sample per reaction. Libraries from individual samples were pooled and sequenced on the NovaSeq 6000 sequencing system, targeting $2 \times 100$ base pairs reads for ATAC as well as RNA.

## Pre-processing of snMultiome data

The raw sequencing signals generated from NovaSeq 6000 in the BCL format were demultiplexed into fastq format using the "mkfastq" function in the Cell Ranger ARC suite (10x Genomics). Cell Ranger-ARC count pipeline was implemented for cell barcode calling, read alignment, and quality assessment using the human reference genome (GRCh38) following the protocols described by 10x Genomics. The final outs file so generated for each sample were used for downstream analysis using Scanpy for RNA and SnapATAC2 and pycisTopic for ATAC.

## RNAseq pre-processing

The resulting matrix from the Cell Ranger ARC pipeline was imported into Scanpy using the scanpy.read_10x_mtx() function for WT1, LSP2 and LSP3 NSCs. The three samples were concatenated and used for further processing. Quality control metrics were calculated for each nucleus, including the number of genes expressed, total count of unique molecular identifiers (UMIs), and the percentage of mitochondrial gene expression. After initial preprocessing, we performed quality control and filtering steps to remove low-quality nuclei and genes with low detection rates. This process was implemented using Scanpy preprocessing functions. Specifically, we applied the following filters: we removed nuclei expressing fewer than 200 genes using the sc.pp.filter_cells() function and genes detected in fewer than 5 nuclei were filtered out using the sc.pp.filter_genes() function. Potential doublets were identified using Scrublet, implemented via sc.pp.scrublet(), with sample information provided as a batch key to account for sample-specific doublet rates. Data was then normalized to the median of total counts across all cells using sc.pp.normalize_total(). A log-transformation ($\log(1+x)$) was applied using sc.pp.log1p() to account for the large dynamic range of expression values. Highly variable genes were identified using sc.pp.highly_variable_genes(), selecting the top 2000 genes across all samples. Principal Component Analysis (PCA) was performed using sc.tl.pca(). A neighborhood graph was constructed using sc.pp.neighbors() based on the PCA representation. Uniform Manifold Approximation and Projection (UMAP) was applied for visualization using sc.tl.umap(). Cells were clustered using the Leiden algorithm, implemented with sc.tl.leiden(). To identify cluster-specific marker genes, we performed differential expression analysis using the Wilcoxon rank-sum test, implemented with sc.tl.rank_genes_groups(). The Leiden based clustering obtained was then used for cell-state annotation using snapseed and scVI-scanVI framework below.

## Cell-state annotation

We used two ways to annotate cells in our dataset. First, we used Snapseed that uses a pre-defined marker list to provide annotation

results. To this end, we used the cell-type marker list from Braun et al (2023) and filtered other regions to retain only forebrain, which resulted in neuronal IPC, radial-glia, glioblast, and neuron cell-types. This list was converted to .yaml file as per Snapseed instructions. Second, we employed the scvi-scanvi framework for cell-type annotation of our single-cell RNA sequencing (scRNA-seq) data, to leverage both unsupervised and supervised learning techniques to integrate reference and query datasets and transfer cell-type labels. We first prepared the reference datasets Braun et al (2023) (https://github.com/linnarsson-lab/developing-human-brain) and Wang et al (2025) (https://cell.ucsf.edu/snMultiome/) using scanpy. Both the datasets were filtered to include specific cell populations of interest. Specifically, for Wang et al, we retained cells from first, second, and third trimesters, and focusing on cells identified in their 'subclass' category: Astrocytes, Cajal-Retzius cell, GABAergic neuron, Glutamatergic neuron, IPC-EN, IPC-Glia, OPC, Oligodendrocyte and Radial-glia. To ease computing resource, we down sampled the dataset to 80,000 cells for Wang et al and $\sim 2 \times 10^5$ cells for Braun et al dataset. Quality control was performed by filtering out cells with fewer than 300 expressed genes. Highly variable genes (HVGs) were identified using Scanpy's highly_variable_genes function, selecting the top 2000 genes. We employed the SCVI (Single-Cell Variational Inference) model for dimensionality reduction and data integration. The data was log-normalized, and the SCVI model was set up and trained on the reference dataset. Latent representations were extracted for downstream analysis. Using the SCVI latent representations, we constructed a neighbor graph and performed Leiden clustering for unsupervised cell grouping. The cell type labels were prepared for supervised learning using the SCANVI model [ref]. The WT1, LSP2, LSP3 NSCs processed query dataset was loaded and preprocessed similarly to the reference dataset as outlined. We identified common genes between the reference and query datasets, subsetting both to these shared highly variable genes to ensure compatibility for integration. For dataset integration and cell type prediction, we employed the SCANVI (Single-Cell Annotation using Variational Inference) model. The model was trained on the reference dataset and then applied to the query dataset. SCANVI latent representations were extracted, and cell types were predicted for the query cells based on the reference dataset annotations. We selected highly confident predictions for Braun et al (2023), followed by Wang et al (2025), through the confusion matrix. Finally, based on the consensus from Snapseed, scVI-scanVI framework, the 14 cell clusters in our NSC-dataset were labeled.

## SCENIC+ analysis

We employed the SCENIC+ workflow to construct gene regulatory networks for the clusters in our NSC dataset. The analysis pipeline began with consensus peak calling for each cell type using MACS2 with the default parameters. We implemented the Latent Dirichlet Allocation (LDA) algorithm using MALLET. The default parameters were used, with 500 iterations for topic modeling. Topics were binarized using the Otsu method to convert continuous distributions into discrete region sets. Both differentially accessible regions (DARs) and topics (sets of co-accessible regions) across cell types were used as candidate enhancers. Outputs of pycisTopic cistopic object and scanpy were used as input for SCENIC+

analysis as outlined in their tutorial default settings (Bravo González-Blas et al, 2023).

## snapATAC2

ATAC-seq data was imported using SnapATAC2's import_data function, utilizing the hg38 genome as reference. Initial quality control metrics, including the number of fragments, fraction of duplicates, and fraction of mitochondrial reads, were analyzed with default settings as per the tutorial. Cells were filtered based on the number of counts (5000–100,000) and minimum TSSE score (>10). A tile matrix was generated, and 250,000 features were selected for downstream analysis. Potential doublets were identified and filtered using the Scrublet algorithm. Dimensionality reduction was performed using spectral decomposition, followed by k-nearest neighbor graph construction and Leiden clustering for initial cell grouping. A gene activity matrix was created using the make_gene_matrix function, associating ATAC-seq peaks with gene annotations from the hg38 genome.

To annotate and compare ATAC cell clusters, we integrated our ATAC-seq data with its paired RNA NSC dataset annotated above. The query (ATAC-seq) and reference (RNA-seq) datasets were concatenated, retaining only genes present in both datasets. Highly variable genes (top 5000) were identified using Seurat v3 methodology. We utilized the scvi-tools package for data integration and cell type prediction. A Variational Autoencoder (VAE) model was trained on the combined dataset using the SCVI class, with 2 layers, 30 latent dimensions, and negative binomial gene likelihood. The model was trained for up to 1000 epochs with early stopping. Subsequently, a semi-supervised SCANVI model was initialized from the trained SCVI model. Known cell type labels from the reference dataset were used to guide the annotation of query cells. The SCANVI model was trained for up to 1000 epochs, sampling 100 cells per label in each epoch. Cell type predictions for the query dataset were obtained using the trained SCANVI model. The latent representations from SCANVI were used for neighbor graph construction and UMAP visualization. The predicted cell types were then mapped back to the original ATAC-seq object.

## RNA isolation, cDNA conversion and quantitative PCR

RNA from multiple biological replicates of brain organoids were extracted using TRIzol (Ambion, Life Technologies), as per manufacturer's instructions. The total RNA was then quantified using NanoDrop 1000 spectrophotometer (Thermo Fisher Scientific). 1 μg of RNA was used for DNase I treatment in a reaction mixture of 10 mM DTT and 40U RNase inhibitor (RNaseOUT, Thermo Fisher Scientific). This reaction mixture was then incubated at 37 °C for 30 min followed by 70 °C heat inactivation for 10 min. For cDNA synthesis, 200U of Superscript II Reverse Transcriptase (Invitrogen) was added to the reaction mixture along with 2.5 μM random hexamers and 0.5 mM dNTPs. The reaction mixture was then incubated at 25 °C for 10 min, followed by 42 °C for 60 min and heat inactivated at 70 °C for 10 min, along with a no reverse-transcriptase control sample. Real-time quantitative PCR of the cDNA samples was performed using primers for genes of interest and control gene, GAPDH, on Applied Biosystems 7500 fast qRT-PCR system. The reporter used here is Power SYBR Green Master mix (Applied Biosystems). The reaction was run for 40

**The paper explained**

**Problem**

Lowe syndrome (LS), a rare X-linked disorder caused by mutations in OCRL, with an estimated prevalence of 1 in 500,000 people. LS leads to severe neurodevelopmental delays, intellectual disability, hypotonia, seizures, and behavioral problems, which significantly impact quality of life. The mechanisms behind the neurological symptoms are complex and not well elucidated. Presently, no treatments exist that can ameliorate these clinical problems.

**Results**

LS patient-derived neural models (2D and 3D brain organoids) revealed that loss of OCRL elevates PI(4,5)P$_2$ levels and leads to increased DLK1 levels and hyperactive Notch signaling. This shifts the LS neural stem cells (NSCs) towards glial progenitor/astrocyte states. Developing LS neurons show reduced excitability that is dependent on PI(4,5)P$_2$ levels. PIP5K inhibition rescued neuronal excitability and normalized hyperactive Notch pathway components.

**Impact**

Our study demonstrates that pharmacological modulation using of PI(4,5)P$_2$ rescues the Lowe syndrome brain phenotypes at physiological level, using a 'disease-in-a-dish' model, that recapitulates early neurodevelopment. This work provides a foundation for exploring therapies that involve modulating neural development and excitability by targeting PI(4,5)P$_2$ dependent mechanisms.

cycles of 95 °C for 30 s (denaturation), 60 °C for 30 s (annealing) and 72 °C for 45 s (extension). The relative mRNA expression of different genes was then calculated by $\Delta C_t$ method, normalizing their $2^{-\Delta Ct}$ values to GAPDH.

## Statistical analysis

Total number of samples included in each experiment is included in the results section/figure legends. All statistical comparisons were performed in Graphpad Prism (v10.2.3). For Fig. 6E, outliers were identified using the ROUT method (Q = 1%) in GraphPad Prism and excluded from statistical analyses. All statistical test used for assessing significance in each experiment is reported in the figure legends. For each experiment, all genotypes with their respective controls were processed together on the same day to minimize batch effects. Each experiment was independently repeated using samples generated in separate rounds of differentiations. No randomization or blinding was performed. *p*-values indicated are as follows: *$p < 0.05$; **$p < 0.01$; ***$p < 0.001$; ****$p < 0.0001$. Error bars indicated in the plots throughout the manuscript represents: Mean $+/-$ SEM.

## Data availability

The datasets produced in this study are available in the GEO data base. Single nuclei multiome data containing gene expression (GEX) and chromatin accessibility files have been deposited to the Gene Expression Omnibus (GEO) and can be found under the accession number: GSE298342. The source data of this paper has been uploaded on BioStudies and assigned the BioStudies accession number S-BSST2141 (https://doi.org/10.6019/S-BSST2141).

The source data of this paper are collected in the following database record: biostudies:S-SCDT-10_1038-S44321-025-00327-y.

## Peer review information

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

## Acknowledgements

This work was supported by the Department of Atomic Energy, Government of India, under Project Identification No. RTI 4006, a Wellcome-DBT India Alliance Senior Fellowship to PR (IA/S/14/2/501540), Department of Biotechnology, Government of India (BT/PR17316/MED/31/326/2015), the Pratiksha Trust and Rohini Nilekani Philanthropies. We thank the NCBS Imaging, Mass spectrometry, Stem Cell and NGS sequencing facility for support. We thank Dr. Zubin Rashid for his help in the electrophysiological analysis in this study.

## Author contributions

**Yojet Sharma**: Conceptualization; Investigation; Methodology; Writing—original draft; Writing—review and editing. **Priyanka Bhatia**: Investigation; Writing—review and editing. **Gagana Rangappa**: Investigation; Writing—review and editing. **Sankhanil Saha**: Investigation; Writing—review and editing. **Padinjat Raghu**: Conceptualization; Supervision; Funding acquisition; Writing—original draft; Project administration; Writing—review and editing.

Source data underlying figure panels in this paper may have individual authorship assigned. Where available, figure panel/source data authorship is listed in the following database record: biostudies:S-SCDT-10_1038-S44321-025-00327-y.

## Funding

## Disclosure and competing interests statement

The authors declare no competing interests.

# Expanded View Figures

**Figure EV1.   Generation of OCRL^KO cell line.**

(A) iPSC-derived NSCs using dual-SMAD inhibition exhibit canonical NSC markers Nestin (red), Pax6 and FOXG1 (green). Nuclei were stained with DAPI, Scale bar = 50 μm. Immunofluorescence images obtained from WT1, LSP2, LSP3 and LSP4. (B) CRISPR-Cas9 targeting strategy: Two sgRNAs (G1 and G2) were designed to target exon 8 of *OCRL*, prior to the 5′-Phosphatase domain of the protein. Target sites of *OCRL*-688-G1 and *OCRL*-688-G2 are highlighted in yellow and green, respectively. (C) Sequence chromatogram confirming insertion of 11 bp (red outlined box) in OCRL^KO iPSC (lower panel), compared to non-edited WT2 iPSC (upper panel). (D) iPSC from WT2 and OCRL^KO displaying pluripotent markers viz., SOX2 (green), SSEA4 (red), Oct-4 (green), TRA-160 (red); iPSCs were counter-stained with DAPI (blue), scale bar = 50 μm. (E) Normal karyogram confirming chromosomal integrity of OCRL^KO iPSC. (F) iPSC-derived NSC from WT2 and OCRL^KO exhibit canonical NSC markers Nestin (red), Pax6 and FOXG1 (green). Nuclei were stained with DAPI, scale bar = 50 μm.

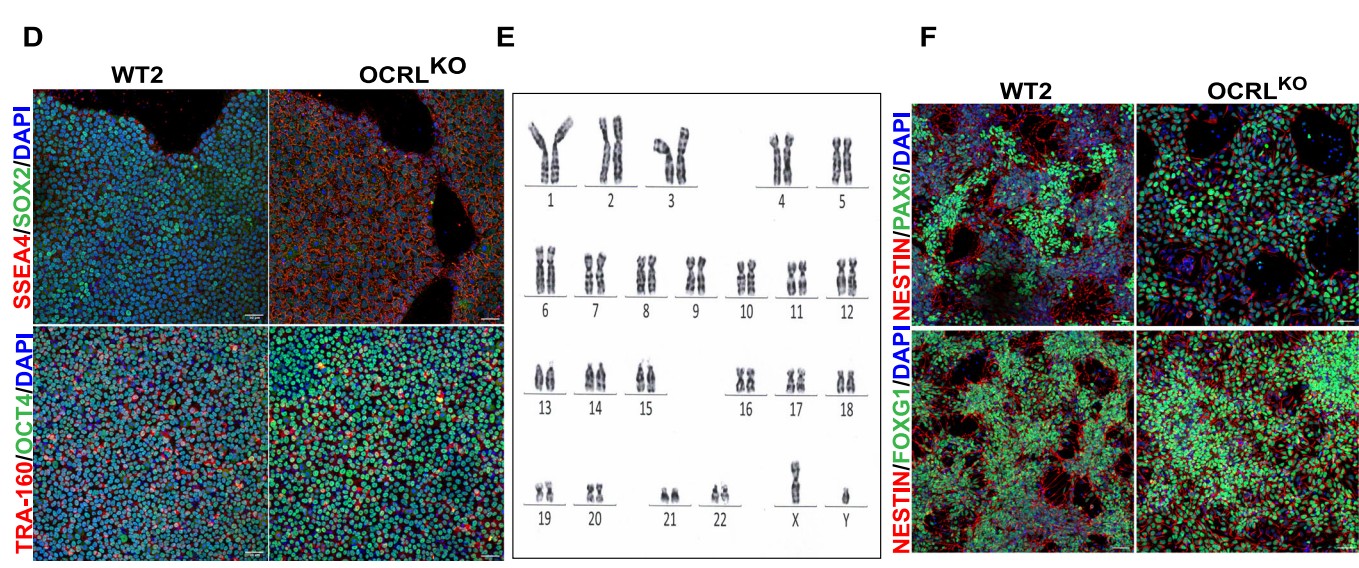

**A**

WT1　LSP2　LSP3　LSP4

NESTIN/PAX6/DAPI

NESTIN/FOXG1/DAPI

**B**

X chromosome　　q26.1

PH | 5'-Phosphatase | ASH | Rho-GAP
1　119　237　563　564　678　721　901

1 2 3 4 5 6 7 8 9 10 11 12 13 14 15 16 17 18 19 20 21 22 23

G1 G2

GCTTCCACGTGAAAAAGAAGCTTCTAACAAGGAGCAGCCCAAAGTGACCAACACCATGCGGAAGCT
CTTTGTACCAAATACCCAATCTGGGCAGCGGGAGGGTCTCATCAAACATATCCTGGCAAAGCGAGAGA
AAGAATATGTCAACATTCAGACTTTCAG

**C**

WT2
ACCAAATACCCAATCTGGGCAGCGGGAGGGTCTCATCAAACATATCCTGGCAA
200　190　180　170　160　150

OCRL^KO
ACCCAATCTGGGCAGCGGGAGGGTCTCATCAAACAT TTGC A GGA GGGT CCTGGCAA
200　190　180　170　160　150

**D**

WT2　OCRL^KO

SSEA4/SOX2/DAPI

TRA-160/OCT4/DAPI

**E**

1 2 3　4 5
6 7 8 9 10 11 12
13 14 15　16 17 18
19 20　21 22　X Y

**F**

WT2　OCRL^KO

NESTIN/PAX6/DAPI

NESTIN/FOXG1/DAPI

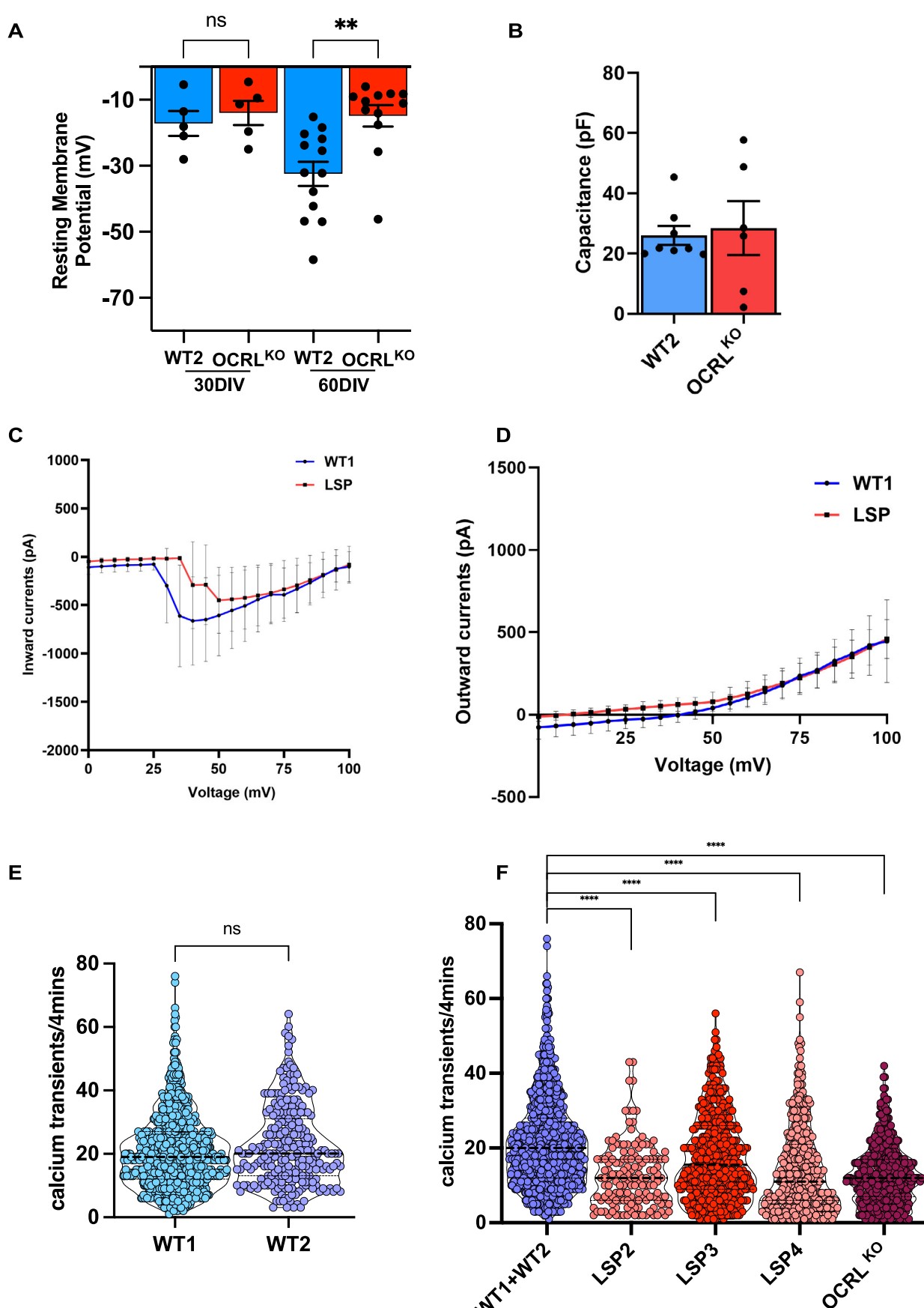

**Figure EV2. Physiological properties of iPSC derived neurons.**

(A) Resting membrane potential from WT2 and OCRL$^{KO}$ derived from 30 and 60 DIV neurons measured during whole cell recordings. Y-axis show resting membrane potential in mV. Each point represents recording from a single neuron obtained through multiple differentiations ($n = 2$). Statistical significance was assessed using ordinary one-way ANOVA followed by Tukey's multiple comparison test, OCRL$^{KO}$ vs WT2 60DIV $p = 0.0043$**. Error bars: Mean $+/-$ SEM shown. (B) Capacitance measurements from WT2 and OCRL$^{KO}$ 60 DIV neurons measured during whole cell recordings ($n = 2$). Y-axis shows capacitance in pF. Each point represents a single cell. Error bars: Mean $+/-$ SEM. Whole-cell patch clamp recordings in voltage-clamp mode from 60DIV WT1 and pooled LSP neurons showing ($n = 2$): (C) Inward and (D) outward currents. Y-axis shows currents in pA. X-axis voltage in mV. Averaged traces from WT1 (11 cells) and LSP (6 cells) is shown ($n = 2$). Current amplitude at each voltage is shown as mean $+/-$ SEM. (E) [Ca$^{2+}$]$_i$ transients data comparing two control cell lines WT1 and WT2 show no statistical difference in the frequency of calcium transients (WT1 $n = 5$; WT2 $n = 2$). Statistical significance was assessed by using Mann–Whitney t-test. (F) WT1 and WT2 combined using the available data and plotted against three individual LS patient lines and OCRL$^{KO}$ (WT1, LSP2, LSP3, $n = 3$; WT1, LSP4, WT2, OCRL$^{KO}$ $n = 2$). Statistical test used: one-way ANOVA followed by Dunnett's multiple comparison test, WT1 $+$ WT2 vs LSP2, WT1 $+$ WT2 vs LSP3, WT1 $+$ WT2 vs LSP4, WT1 $+$ WT2 vs OCRL$^{KO}$ $p = <0.0001$****.

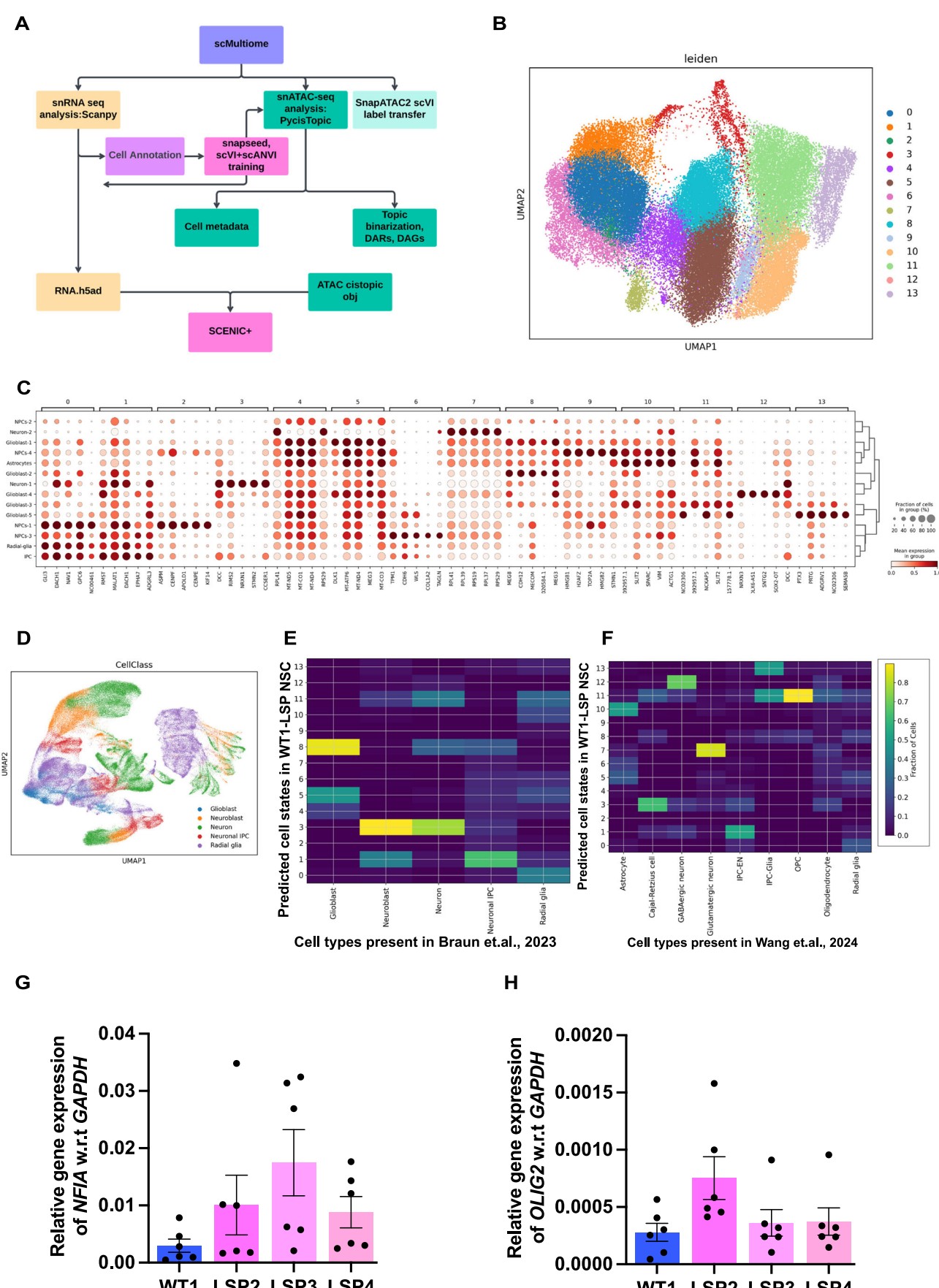

**Figure EV3.   Overview of single nuclei multiome data analysis.**

(A) Diagram illustrating the workflow and steps for generating and analyzing single nuclei multiome data. (B) UMAP projection showing cell clustering based on the Leiden algorithm using the scanpy pipeline. Each color represents a distinct cell cluster. (C) Dot plot displaying expression levels of marker genes across cell clusters generated using scanpy. Dot size indicates the percentage of cells expressing each gene, while color intensity reflects expression level. Upper X-axis shows cell clusters numbered 0–13; Lower X-axis included names of transcripts whose enrichment is being plotted. Y-axis show annotated cell clusters. (D) UMAP colored by cell class, highlighting distribution of major cell types such as glioblasts, neuroblasts, neurons, neuronal IPCs, and radial glia; dataset from Braun et al (2023). (E) Confusion matrix from scVI-scANVI mapping; Y-axis depicts the 14 unannotated clusters obtained from our multiome analysis; X-axis are the cell types seen in the Braun et al dataset showing prediction accuracy for different cell types across conditions. (F) Confusion matrix from scVI-scANVI mapping, detailing prediction outcomes for various cell clusters in Wang et al (2024), dataset. (G, H) RT-PCR analysis of glia specific transcripts in WT1, LSP2, LSP3 and LSP4 NSC. Each point represents analysis from NSCs generated from multiple neural inductions ($n = 6$) from iPSCs. Y-Axis represents transcript levels for a glial marker mRNA normalized to *GAPDH* levels. (G) *NF1A* and (H) *OLIG2* transcripts. Error bars: Mean $+/-$ SEM.

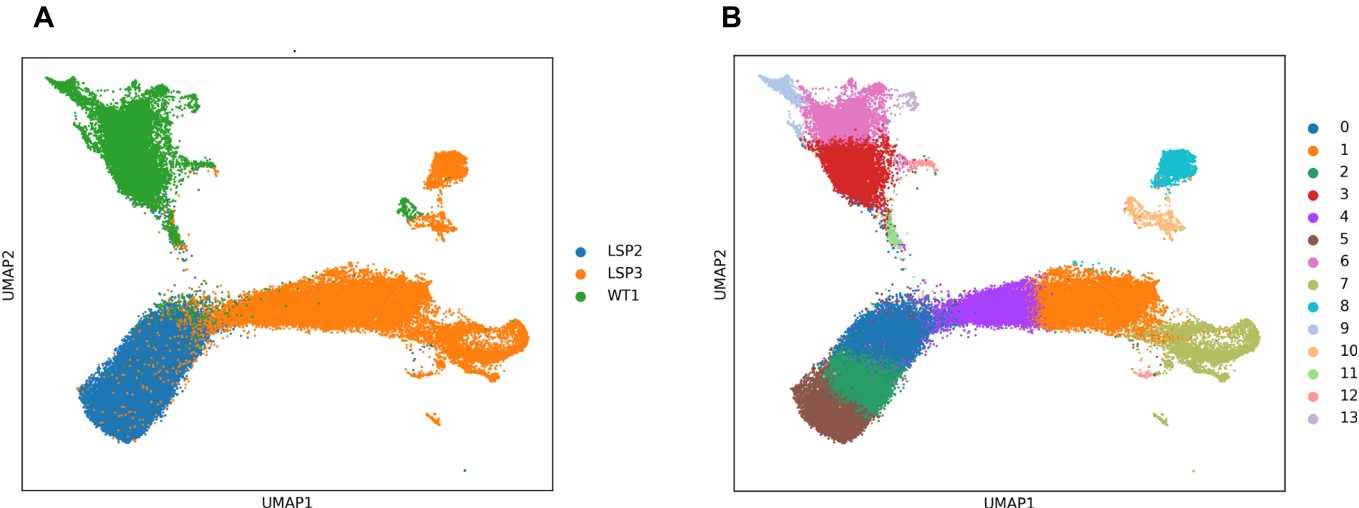

**A**

**B**

**C**

| Motif logo | TF | Cell-cluster | NES | AUC |
|---|---|---|---|---|
| | LHX9 | Radial-glia | 7.269415 | 0.015211 |
| | EMX1 | Radial-glia | 6.380436 | 0.013670 |
| | SOX2, POU2F1 | Glioblast | 7.025670 | 0.013287 |
| | NFIX, NFIC, NFIB, NFIA | Astrocytes | 6.759462 | 0.012162 |
| | RFX2, RFX3, RFX4, RFX5, RFX1 | Astrocytes | 18.842255 | 0.028898 |

Figure EV4. **Validation of gliogenic bias in LSP NSC and transcription factor motif analysis from multiome data.**

(A) UMAP projection for ATAC-seq dataset showing sample distribution (WT1, LSP2, LSP3) using the snapATAC2 pipeline. (B) UMAP projection for ATAC-seq dataset with Leiden clustering across all samples, indicating distinct clusters identified through the snapATAC2 analysis. (C) The table describes enriched motifs for each cell cluster obtained through SCENIC+ analysis. Motif logo shows y-axis representing the information content measured in bits, ranging from 0 to 2 bits for DNA sequences. Specifically, a value of 0 bits indicates a position where all nucleotides occur with equal probability. A value of 2 bits indicates a position where only a single nucleotide occurs. The height of each stack shows how conserved that position is - taller stacks mean higher conservation. The x-axis shows the nucleotide positions in the DNA sequence alignment. Each position contains a stack of letters (A, C, G, T) where: the letters are stacked according to their relative frequency at that position. The height of each individual letter within a stack represents how frequently that nucleotide appears at that position and the most common base appears as the largest letter at the top of each stack. In summary, In the motif logo, the x-axis represents sequential nucleotide positions (5′ to 3′), and the y-axis shows information content in bits (0-2.0). Letter height indicates the relative frequency of each nucleotide (A, T, C, G) at each position, with taller letters representing more frequently occurring bases.

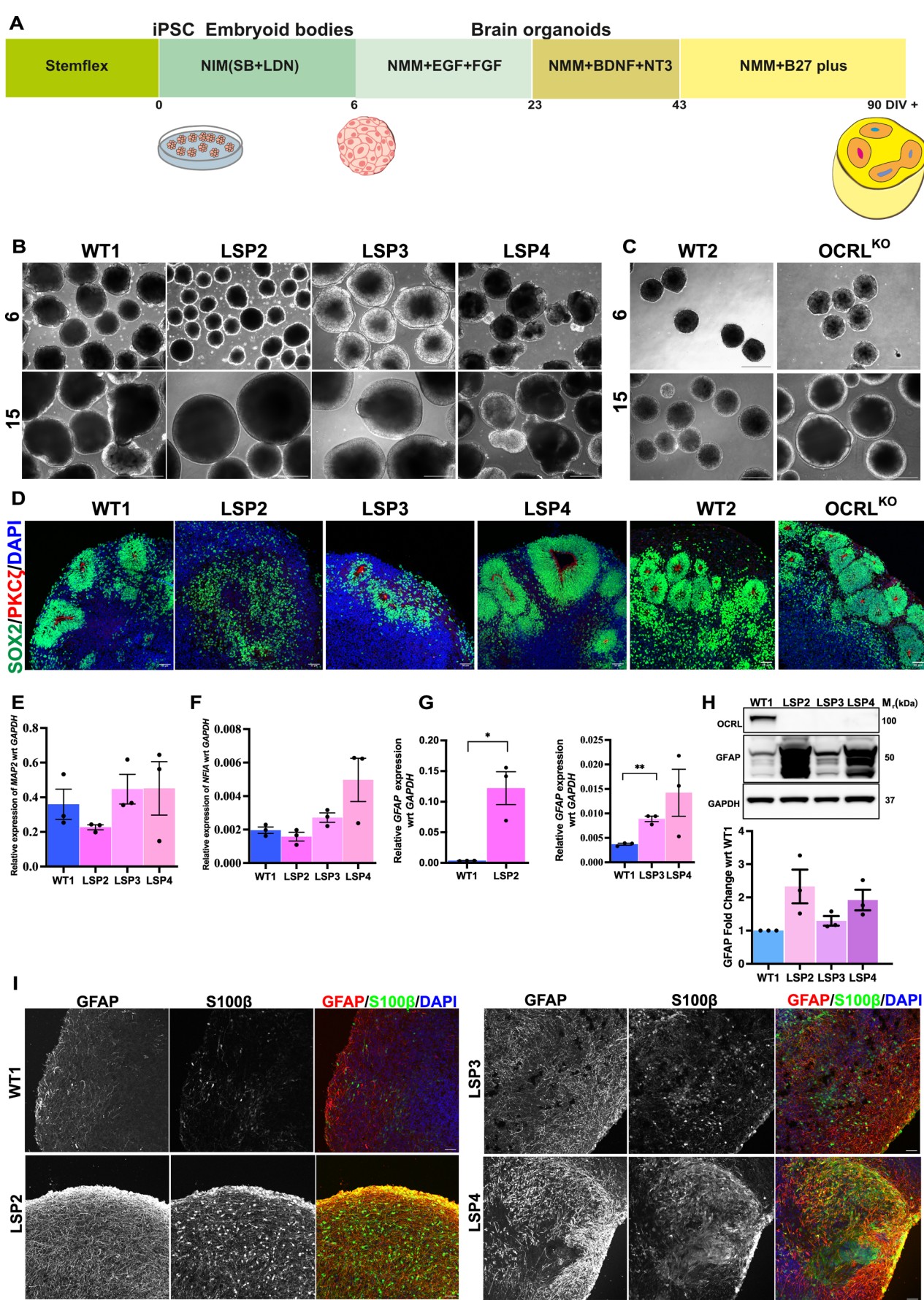

◀ **Figure EV5. Loss of OCRL results in increased gliogenesis in 90DIV LSP brain organoids.**

(A) Schematic for brain organoid generation. iPSC are dissociated into single cells to allow formation of embryoid bodies (EBs). EBs are suspended in neural induction medium (NIM) with TGFβ and BMP inhibitors, followed by changing to neural maintenance medium (NMM) with EGF and FGF. For maturation period, the brain organoids are maintained in NMM with BDNF and NT3. (B) Phase contrast images showing the morphology of WT1, LSP2, LSP3, LSP4, and (C) WT2 and OCRL$^{KO}$ neural spheroids across specific developmental timepoints: 6 and 15 DIV. Scale bar: 500 μm. (D) Confocal images of WT1, LSP2, LSP3, LSP4, WT2 and OCRL$^{KO}$ 60DIV showing typical neural rosettes consisting of apical marker protein kinase-C (PKC-ζ, red) and SOX2+ (green) neural stem cells. Nuclei were counterstained with DAPI. Scale bar = 50 μm. RT-PCR analysis of neural spheroids derived from the LSP iPSC lines compared to WT1. Transcript levels for the neuronal marker (E) MAP2 and (F) NFIA are shown. Y-axis depicts transcript levels normalized to the housekeeping gene GAPDH. Error bars: Mean $+/-$ SEM. (G) GFAP levels for WT1, LSP2, LSP3, and LSP4 are shown. For MAP2, NFIA and GFAP each point depicts transcripts measured from extracts of 15–20 spheroids ($n = 3$). Error bars: Mean $+/-$ SEM. Statistical test: Unpaired t-test with Welch correction, for GFAP, WT1 vs LSP2 $p = 0.0047$*; WT1 vs LSP3 $p = 0.0064$**. (H) Western blot of 90DIV neural spheroids cultures WT1, LSP2, LSP3, LSP4; immunoblotting for OCRL (100 kDa) and GFAP (50 kDa) proteins are shown; GAPDH (37 kDa) was used as a loading control. GFAP fold-change in LSP derived 90 DIV neural spheroids analyzed and plotted w.r.t. the control WT1. Y-axis shows the fold change in GFAP levels in LSP relative to WT1 ($n = 3$). Error bars: Mean $+/-$ SEM. (I) Individual maximum z-projection of confocal images of GFAP (red) S100β (green) from 90 DIV WT1, LSP2, LSP3 and LSP4 brain organoids. Nuclei were stained with DAPI (blue). Scale bar: 50 μm.

