## [Peer Review File · EMBO Molecular Medicine]

Enhanced gliogenesis and delayed maturation underpin the neurodevelopmental defects in Lowe syndrome

Yojet Sharma, Priyanka Bhatia, Gagana Rangappa, Sankhanil Saha, and Padinjat Raghu

Corresponding author: Padinjat Raghu (praghu@ncbs.res.in)

Review Timeline:

Transferred from Review Commons:	16th May 25
Editorial Decision:	19th May 25
Revision Received:	27th Aug 25
Editorial Decision:	18th Sep 25
Revision Received:	1st Oct 25
Accepted:	7th Oct 25

Editor: *Zeljko Durdevic*

Transaction Report:

Review
COMMONS

This manuscript was transferred to EMBO Molecular Medicine following peer review at Review Commons.

Review #1**1. Evidence, reproducibility and clarity:****Evidence, reproducibility and clarity (Required)******Summary****

The study by Sharma et al uses iPSC and neural differentiation in 2D and 3D to investigate how mutation in the OCRL gene affects neural differentiation and neurons. Mutation in the OCRL gene the cause of Lowe Syndrome (LS), a neurodevelopmental disorder. Neural cultures derived from LS patient iPSCs exhibited reduced excitability and increased glial markers expression. Additional data show increased levels of DLK1, cleaved Notch protein, and HES5 indicate upregulated Notch signaling in OCRL mutated neural cells. Treatment of brain organoids with a PIP5K inhibitor restored calcium signalling in neurons. These findings describe new dysregulated phenotypes in neural cultures of OCRL mutated cell shedding light on the underlying caus of Lowe Syndrome.

****Major comments****

In general, I think the use of iNeurons usually means direct reprogramming from a somatic cell to neurons without the iPSC stage. Could be confusing to use this term for iPSC derived neurons.

Please add at least one more replicate of WP cell line to the single nuclei RNAseq.

The WT1 and the patient lines are rarely analysed together with the WT2 and KO lines, thus it is tricky to understand if the KO line is mimicking the patient lines? Please, add more merged analyses.

Co-analysing all lines:

i) would show if the KO line is more similar to the patient lines or to the WT1 or somewhere inbetween.

ii) Could answer questions about the variation in phenotypes between the genetic backgrounds.

iii) Elucidate how much variability there is between the the two WT lines in your assays. If

the two WT lines vary much then conclusions about phenotypes in the patients and KO lines might need to be rethought?

Please include more discussion and rationale around the link between the expression pattern of OCRL and the various phenotypes shown. From the RNAseq data performed at the NSC state where the expression of OCRL is lower than in neurons there are considerable differences in cell type distribution between lines. How can this skew cell type distribution affect downstream differentiation and neuronal function? Also, OCRL is expressed also at the iPSC state as shown in Figure 1I, do you see any phenotypes in iPSC? If not, explain how that could be.

There is not enough data in the manuscript to show mechanistic links between OCRL, DLK1 and Notch so be aware not to overstate the conclusions.

Line 173, please describe what mutation in the OCRL these patients have, is it a biallelic deletion? Is the protein totally absent? Please show western blot analyses of the protein in the patient lines.

Would be good with a bit of clinical explanation of these patients? Do they have the same level of severity? Are there any differences between their clinical symptoms? This could be interesting to link to differences in cellular phenotypes.

Line 185, please be clear on how the CRISPR KO line genetically mimics the patients' lines.

Figure 1I, could the protein levels at the different stages be quantified?

Figure 3A, there seem to be much more cells in LSP2, making it tricky to compare with the other cell lines. Density during differentiation can affect the cell fate. Please, provide images from the different lines that are comparable with similar density. Please provide quantification to the statement that there is fewer number of S100B cells in the LSP lines. Figure 3B, the images show cells very different, and it is tricky to compare similarities and differences, please provide images that look more similar to each other. Avoid images with clusters of cells or make sure to select representative images with clusters from each cell line. If the clustering is a phenotype explain and quantify that. Make sure the density is similar in all pictures.

Line 2018, the statement "In the same cultures, there was no change in the staining pattern of the neuronal markers MAP2 and CTIP2 (Fig 3B)" is not strengthened by the figure. Please provide new pictures or data to prove the statement.

Figure 3E, please describe all markers in the picture, thus also MAP2, S100B, CTIP2 and draw conclusions. Try to show comparable pictures.

Fig 3D and G, what are the replicates? please explain.

Figure 4 A, C, there is a large difference in the ratio of different cell types between the different cell lines, also between the LSP2 and LSP3. This would indicate either that the genetic background affects the phenotype to a large extent or that there is large variability between rounds of differentiation. To understand how much variability that comes from the differentiation and culturing:

i) another replicate of WP cell from another donor (WT2) should be included (single nuclei RNAseq)

ii) Confirm that three independent rounds of differentiation of the WT1, WT2, LSP2, LSP3, LSP4, and OCRL-KO result in similar outcome when it comes to cell type distribution. Could be done with qPCR marker.

Line 309, "astrocytic transcripts NF1A and GFAP was elevated" It is unclear from this sentence in which cell lines NF1A and GFAP is elevated? Please explain.

Figure 5C, E, G, there is a large variation of Cnotch and Hes5 expression between the different

Figure 5H, unclear which of the bands that is DLK1 and how the bands relate to the quantification. The band at 50 kDa seems to be stronger in the WT2 than in the OCRL-KO but in the quantification in Figure 5I, it shows 2x more in the KO. Thus, the other way around.

Figure 6, please show that the inhibitor is inhibiting PIP5KC.
Have you titrated the added concentration of the inhibitor?

Figure 6B, why do the calcium data for the WT2+1Ci look so different to the other, the dots are much more spread and seem to fewer replicates than for the other sample, please explain.

Figure 6F, there is no significant differences between the bars but the statement in the text (sentence starts on line 332) indicate it is, please update the figure or remove the

statement.

Figure 6G, the HES5 expression seem to behave very similar in both WT2 and OCRL-KO cells when the inhibitor is used. What does this mean? Seems to not be linked to OCRL. Explain.

****Minor comments****

The panels in Figure 6 are not completely referred to correctly in the text, please check. Double check that all figure panels are referred to properly in the text

2. Significance:

Significance (Required)

The manuscript is an interesting addition to the in vitro iPSC derived cellular modelling of neurodevelopmental disorder.

***Strengths:**

The use of both patient iPSC lines and CRISPR edited lines
The use of both monolayer and 3D cultures

***Weaknesses:** the significance decrease a bit due too few replicates (only 1 WT line in each experiment) and the variability between the patients' cell lines.

3. How much time do you estimate the authors will need to complete the suggested revisions:

Estimated time to Complete Revisions (Required)

(Decision Recommendation)

Between 3 and 6 months

4. Review Commons values the work of reviewers and encourages them to get credit for their work. Select 'Yes' below to register your reviewing activity at Web of Science Reviewer Recognition Service (formerly Publons); note that the content of your review will not be visible on Web of Science.

Yes

Review #2

1. Evidence, reproducibility and clarity:

Evidence, reproducibility and clarity (Required)

This paper describes the effects of loss of OCRL (the Lowe syndrome protein) upon the function and differentiation of neurones, using an in vitro iPSC model system. Cells derived from three related Lowe syndrome patients and an OCRL knockout, generated using CRISPR, were used for these experiments. The results show that upon loss of OCRL, differentiation of stem cells into neurones is reduced, with an increased number of cells adopting glial and astrocytic fates. The neurones that are generated have reduced calcium transients and electrical activity. Gene expression data combined with biochemical analysis indicate altered Notch activity, which may account for the altered cell fate data seen in the in vitro differentiation model. Finally, rescue of cell fate and neuronal activity is seen upon knockdown of a PIP5K, which indicates that these phenotypes are due to the elevated PIP2 levels seen on the OCRL-deficient cells.

The results provide new insights into the pathogenesis of Lowe syndrome. I found the paper to be well done, and the data supports the conclusions of the authors. I have a few comments below that may improve the manuscript:

****Major points****

1. The UMAP and ATAC-Seq data indicate different maps for the two different Lowe syndrome patient-derived cells (Fig 4 and Fig S3). This suggests that the cells are quite different, and therefore that changes seen in one Lowe syndrome patient may not be applicable to the others. I think this heterogeneity has important implications for the paper i.e. how general are findings obtained? Several different glioblast types are described (numbered 1-5)- how different or similar are these?
2. The PIP5K inhibitor seems to have a very strong effect on both WT and KO cells in terms of Notch activity (Fig 5G). This strongly suggests the effects of this inhibitor are not through OCRL and that changes in PIP2 induced by the inhibitor override those of OCRL. Thus, the experiments shown in Fig 5 seem not to be due to a rescue of OCRL activity as such.

****Minor points****

1. The main text needs to say what synapsin is and why it was analysed. In Fig 1I, synapsin abundance declines at 90 days. This appears quite strange. The authors should comment on it in the text.
2. In Fig 2A and 3B there are clumps of green cells (CTIP2 positive). I am concerned that the lack of uniformity in the cell distribution could impact other analysis performed, where certain fields of view have been analysed e.g. by imaging or electrophysiology e.g. calcium measurements.
3. In Fig 2J and 2K are the differences between samples significant? The error bars are huge.
4. In Fig S4- it would be good to show gene expression analysis and GFAP staining for organoids made using the OCRL KO cells
5. Fig 5A needs more annotation- fold change comparing what to what?
6. There should be more information provided in the main text relating to DLK1. For example, it is shown to be secreted, but no information is provided on whether this is expected. Secreted? The DLK1 blot in Fig 5F is not convincing.
7. Of the 3 PIP5Ks, only PIP5Kc was targeted. The rationale for picking only this one needs to be provided.
8. Fig 6D is confusing. I suspect the figure labelling is not correct- it does not correlate with the graphs.

2. Significance:

Significance (Required)

This paper is significant because it provides important new information on the neurological features of Lowe syndrome. The approach is novel in terms of studying this condition. The findings are likely to be of interest to clinicians, cell biologists, neurobiologists and those studying human development. My expertise is in membrane traffic and OCRL/Lowe syndrome. I am not a neurobiologist.

3. How much time do you estimate the authors will need to complete the suggested revisions:

Estimated time to Complete Revisions (Required)

(Decision Recommendation)

Less than 1 month

4. Review Commons values the work of reviewers and encourages them to get credit for their work. Select 'Yes' below to register your reviewing activity at Web of Science

Reviewer Recognition Service (formerly Publons); note that the content of your review will not be visible on Web of Science.

Yes

Review #3

1. Evidence, reproducibility and clarity:

Evidence, reproducibility and clarity (Required)

This paper by Sharma et al describes findings in an iPSC model of Lowe Syndrome. This is an important line of research because no mouse models phenocopy the neurodevelopmental aspects of the condition. They identified a potential role of Notch signaling in pathogenesis, a potentially druggable target. However, several issues need to be addressed.

****Major issues****

1. The sample size is very small, which is understandable to some extent given the expense and difficulty doing research using iPSCs. However, there are a couple of opportunities to improve the sample size. For example, in the analysis of DLK1 and other proteins shown in Figure 5, the analysis amounts to a single control vs the 3 patient lines, and a single control vs the KO line. The separation is justified because a complete KO of the gene might result in differences compared to hypomorphic mutation that apparently affects the 3 cases. However, there is no reason why WT1 and WT2 shouldn't be combined. They are both random controls. This might not affect the results of the other proteins analyzed, NOTCH and HES5, but the significance of DLK1 could change. DLK1 expression brings up another point. This, along with MEG3 and MEG8 are imprinted genes, two of the top differentially expressed genes in this study. However, these findings can be questioned by the well-known phenomenon that the expression of some imprinted genes may not be properly maintained during iPSC reprogramming. Thus, the differential expression of these imprinted genes might be due to a reprogramming artifact rather than the effects of OCRL per se. Analyzing both controls together could mitigate this objection. However, even if it does, the potential dysregulation of imprinted genes in the development of iPSCs should be acknowledged and addressed.
2. Similarly, in the calcium signaling experiment shown in fig.2, the KO and patient lines are justifiably separated. However, again, why not combine both controls in the comparison

with the patient samples?

3. Regarding the hypomorphic nature of the patient-specific iPSC, I do not see the OCRL variant that was found in the family. Please correct me if I missed that, and if it was omitted, it should be included. I suspect that the variant generates a hypomorphic OCRL protein because the authors show expression in Figure 1D. Hypomorphic OCRL mutations compared with complete KO could show differences in molecular phenotypes, as found in Barnes et al. (PMID: 30147856) in an analysis of F-actin and WAVE-1 expression.

****Minor issue****

The authors use the term mental retardation on line 102 to describe the cognitive phenotype in Lowe Syndrome. Medical, legal, and advocacy groups have abandoned this term because it is viewed as offensive. It is being replaced by intellectual disability, although this term also is problematic. In any event, many conferences on autism and intellectual disabilities are attended by families, and high-functioning cases sometimes address an audience of scientists. They would object to the use of this term if presented in a talk by one of the co-authors.

2. Significance:

Significance (Required)

as noted above, this study is significant because there are no mammalian models available to address the neurodevelopmental aspects of Lowe Syndrome

3. How much time do you estimate the authors will need to complete the suggested revisions:

Estimated time to Complete Revisions (Required)

(Decision Recommendation)

Cannot tell / Not applicable

4. Review Commons values the work of reviewers and encourages them to get credit for their work. Select 'Yes' below to register your reviewing activity at Web of Science Reviewer Recognition Service (formerly Publons); note that the content of your review will not be visible on Web of Science.

Yes

Revision Plan

Manuscript number: RC-2024-02825

Corresponding author(s): Padinjat, Raghu

Key to revision plan document:

Black: reviewer comments

Red: response to reviewer comment-authors

Blue: specific changes that will be done in a revision-authors

1. General Statements [optional]

We thank the reviewers for their detailed comments on our manuscript and appreciating the novelty, quality and thoroughness of the work. Detailed responses to individual queries and revision plans are indicated below.

2. Description of the planned revisions

Reviewer 1:

Summary

The study by Sharma et al uses iPSC and neural differentiation in 2D and 3D to investigate how mutation in the OCRL gene affects neural differentiation and neurons. Mutation in the OCRL gene the cause of Lowe Syndrome (LS), a neurodevelopmental disorder. Neural cultures derived from LS patient iPSCs exhibited reduced excitability and increased glial markers expression. Additional data show increased levels of DLK1, cleaved Notch protein, and HES5 indicate upregulated Notch signaling in OCRL mutated neural cells. Treatment of brain organoids with a PIP5K inhibitor restored calcium signalling in neurons. These findings describe new dysregulated phenotypes in neural cultures of OCRL mutated cell shedding light on the underlying caus of Lowe Syndrome.

Major comments

1. In general, I think the use of iNeurons usually means direct reprogramming from a somatic cell to neurons without the iPSC stage. Could be confusing to use this term for iPSC derived neurons. Thank you for pointing this out. We agree and will remove this term and replace it with a more suitable one in the revised manuscript.
2. Please add at least one more replicate of WP cell line to the single nuclei RNAseq.

There is no cell line called WP1 in the manuscript. We believe the reviewer was likely referring to WT1 (wild-type 1).

10xgenomics guidelines highlight that the statistical power of a multiome experiment relies on several factors including sequencing depth, total number of cells per sample, sample size and number of cells per cell type of interest (10xgenomics). In this study, we performed a multiome experiment and obtained high-quality reads from 20,000 nuclei for each sample for both the modalities: snRNA seq and snATAC seq. The multiome kit recommends a lower limit is 10,000

nuclei per sample. Thus the number of cells sampled per cell line is double the suggested minimum. Therefore, and consistent with other single-cell seq studies already published, our study followed the approach where biological replicates were not included (for e.g see PMID: 39487141, GSE238206; PMID: 31651061; PMID: 32109367, GSE144477; PMID: 40056913, GSE279894; PMID: 38280846 GSE250386; PMID: 36430334, GSE213798; PMID: 33333020, GSE123722; PMID: 32989314, GSE145122; PMID: 38711218, GSE243015, PMID: 38652563, GSE236197). Furthermore, single-cell RNA-seq inherently treats each individual cell as as a replicate (Satija lab guidelines, PMID: 29567991; Wellcome Sanger Institute), reducing the necessity for additional biological replicates. Overall this appears to be the current standard in the field which we have followed.

Importantly, we took additional steps to validate the predictions our single-nuclei RNA-seq findings experimentally. For this we used a 3D brain organoid system. We confirmed key observations noted initially in 2D neural stem cells using a brain organoid model. This approach allowed us to confirm key predictions from the single cell sequencing data set. For example, in Lowe Syndrome patient derived organoids and OCRL-KO organoids, we noted increased DLK1 levels (Fig5.C-D, H-I) as well as increased GFAP+ cells and gene expression in brain organoids (Fig.S4E,F). These complementary approaches strengthen our confidence in the biological relevance of our findings from the single nuclei sequencing experiments.

3. The WT1 and the patient lines are rarely analysed together with the WT2 and KO lines, thus it is tricky to understand if the KO line is mimicking the patient lines? Please, add more merged analyses. Co-analysing all lines:
 - (i) would show if the KO line is more similar to the patient lines or to the WT1 or somewhere in between.
 - ii) Could answer questions about the variation in phenotypes between the genetic backgrounds.
 - iii) Elucidate how much variability there is between the two WT lines in your assays. If the two WT lines vary much then conclusions about phenotypes in the patients and KO lines might need to be rethought?

The reviewer is right is noting that throughout the manuscript we have analysed the patient lines with WT1 and the KO line with WT2. This was a conscious decision which we believe is the correct one for the following reasons:

It is well recognized and discussed in the literature that genetic background can be a key factor contributing to phenotypes observed in cells differentiated from iPSC (Anderson et al., 2021, PMID: 33861989; Brunner et al., 2023, PMID: 36385170; Hockemeyer and Jaenisch, 2016, PMID: 27152442; Soldner and Jaenisch, 2012, PMID: 30340033; Volpato and Webber, 2020, PMID: 31953356). Therefore, as a matter of abundant precaution, in this study we have tried to use the closest possible genetically matched control lines for analysis.

The patient lines used in this study for Lowe syndrome were all derived from a family in India of Indian ethnic origin. Therefore, in order to reduce the potential impact of genetic background contributing to potential phenotypes, we have used a control line derived from an individual of Indian ethnic background; this line has previously been developed and published by our group (PMID: 29778976 DOI: [10.1016/j.scr.2018.05.001](https://doi.org/10.1016/j.scr.2018.05.001)). By contrast, the OCRL^{KO} line was generated using the control line NCRM5 (WT2); this line is derived from a Caucasian male (RRID:

Revision Plan

CVCL_1E75). Therefore, whenever we have analyzed OCRL^{KO}, we have used NCRM5 as the control; throughout the manuscript, NCRM5 is referred to as WT2.

However, in deference to the reviewer's concerns we have performed a few analyses to compare the extent of variability between the two control lines.

Figure Legend: Replotted $[Ca^{2+}]_i$ transients data from LS patient lines, OCRL^{KO} and two control cell lines WT1 and WT2. (A) There is no statistical difference in the frequency of $[Ca^{2+}]_i$ transients between WT 1 and WT2. Test used-Mann Whitney test. (B) Plot with WT1 and WT2 data combined versus all three LS lines and OCRL^{KO} combined. Test used-Mann Whitney test. (C) WT1 and WT2 combined plotted against three individual patient lines and OCRL^{KO}. Statistical test used One-way ANOVA. (total neurons analysed: WT1:808; WT2:267; LSP2:150; LSP3:462; LSP4:463; OCRL^{KO}:411)

(i) We compared the frequency of calcium transients between neurons of age 30 DIV between WT1 and WT2 (Panel A above). **We found no significant difference between these.**

Additionally, as suggest we combined the data from both control lines into a single set and that from all the LSP patient lines and OCRL^{KO} into another one (Panel B above). At the end of the analysis the difference between control and OCRL depleted cells remains. Please note the large number of cells studied in each genotype.

We also combined both control lines into a single control data set and compared it to each patient line and OCRL^{KO}. We find that each patient line and OCRL^{KO} is still significantly different from the control set (panel C above).

We did not find that OCRL^{KO} to be significantly different from LSP2 or LSP4, indicating that the OCRL^{KO} line closely aligns with the patient-derived lines, supporting the idea that the observed phenotype is primarily disease-driven rather than background-dependent. However, we did observe a significant difference between LSP3 and OCRL^{KO}, highlighting some degree of inter-patient variability. Therefore, the key point is that the disease phenotype remains stable across

different backgrounds, reinforcing the idea that the observed differences are driven by OCRL loss rather than background variability. This will be discussed in the revision.

(ii) In our RTPCR assay for *HES5*, when WT1 and WT2 are plotted together, there is no significant difference observed (panel A below). Similarly, western blotting data for cNotch (panel C) and DLK1 (panel B) of pooled WT1 and WT2 together on one plot shows no significant difference (Unpaired t-test, Welch's correction). Overall, based on the above data, WT1 and WT2 are not statistically different.

Figure legend: Comparison of control lines WT1 and WT2. (A) comparison of *HES5* transcripts. (B) Western blot for DLK1 levels. (C) Western blot for cleaved notch protein levels. Statistical test: Unpaired t-test, Welch's correction.

- Please include more discussion and rationale around the link between the expression pattern of OCRL and the various phenotypes shown. From the RNAseq data performed at the NSC state where the expression of OCRL is lower than in neurons there are considerable differences in cell type distribution between lines. How can this skew cell type distribution affect downstream differentiation and neuronal function?

We would like to highlight that we did not perform bulk RNAseq in NSC and neurons; rather, we performed snRNA seq in NSCs (Fig3). The data in Fig.1E is mined from a publicly available resource dataset (Sidhaye et.al., 2023, PMID: 36989136) as mentioned in line 155, which is an integrated proteomics and transcriptomics generated from iPSC-derived human brain organoids at different stages of development *in-vitro*.

Fig 1D and 1E do indeed show lower levels of OCRL expression in NSC compared to neurons. However, it is important to bear in mind that even though OCRL may be expressed at relatively low levels during the NSC stage, its enzymatic activity could still have a substantial impact. Therefore, even at low expression levels, OCRL could be modulating the PI(4,5)P₂ pool in ways that significantly influence cellular functions, especially during early stages of neurodevelopment that alter cell-fate decisions thereby affecting neuronal excitability.

Our working model posits that loss of OCRL leads to increased levels of PI(4,5)P₂ which upregulates Notch pathway thereby leading to an increase in its downstream effector *HES5*. *HES5* is a known transcription factor influencing gliogenesis and thus leading to a precocious glial shift in OCRL deficient NSCs as seen in our multiome dataset. This temporal perturbation in differentiation affects maturation of LS/OCRL-KO neurons and/or astrocytes leading to a defective neuronal excitability.

5. Also, OCRL is expressed also at the iPSC state as shown in Figure 1I, do you see any phenotypes in iPSC? If not, explain how that could be.

Yes, OCRL is indeed expressed in iPSCs as shown in Figure 1I. In an earlier paper from our lab that described the generation of these patient derived iPSC from Lowe syndrome patients (Akhtar et.al 2022 PMID: 35023542), we have reported that PIP₂ levels are elevated at the iPSC stage as well as NSC stage in OCRL patient lines. We have not performed a detailed analysis of the iPSC stage for these lines as the focus of our investigation was primarily on the later stages of differentiation, particularly in neural progenitors and differentiated neurons. However, in response to the reviewer's questions on why there are no obvious phenotypes at the iPSC we would suggest that this is due to compensation from the activity of other genes of the 5-phosphatase family. In support of this, we would cite our previous study (Akhtar et.al 2022 PMID: 35023542), in which we show that in LS patient derived lines, at the iPSC Stage, at least six other 5-phosphatases are upregulated.

6. There is not enough data in the manuscript to show mechanistic links between OCRL, DLK1 and Notch so be aware not to overstate the conclusions.

We appreciate the reviewer's constructive comment regarding the mechanistic links between OCRL, DLK1, and Notch. Treatment of organoids and neurons with UNC-3230 PIP5K1C inhibitor rescues the observed phenotypes suggesting a role for a PIP₂ dependent process, this process itself remains to be identified. We will adjust the wording in the manuscript during the revision to ensure that this comes through and the conclusions do not appear overstated.

7. Line 173, please describe what mutation in the OCRL these patients have, is it a biallelic deletion? Is the protein totally absents? Please show western blot analyses of the protein in the patient lines.

The patients from whom these LS lines were generated, the nature of the OCRL allele in them and the status of OCRL protein have all been previously been described in detail in a paper from our lab. This paper (Akhtar et.al 2022 PMID: 35023542) has been cited in the present manuscript at the very first occasion that the lines are described (Line 174, references 26 and 27). In addition, in the present manuscript, the protein status of OCRL in all the three patient lines is shown with a Western blot in Figure 3C.

Would be good with a bit of clinical explanation of these patients? Do they have the same level of severity? Are there any differences between their clinical symptoms? This could be interesting to link to differences in cellular phenotypes.

Revision Plan

The clinical details of each patient are described in a preprint from our lab (Pallikonda et.al., 2021 bioRxiv 2021.06.22.449382). The potential reasons for the difference in severity, a very interesting scientific question, is also addressed in this preprint. Currently experimental analysis to support the proposed likely reasons is ongoing in our lab. We feel those analysis are beyond the scope of this manuscript and will be published later this year as a separate study.

As described in in ref 26 and 27, LSP patients have a mutation in exon 8 leading to a stop codon. We mimicked this by CRISPR based genome editing to introduce a stop codon and protein truncation in exon 8 to generate of WT2 to OCRL^{KO}. This is also described in supplementary Fig 1 of the present manuscript and the technical details of line generation are fully described in the materials and methods.

Like the patient lines OCRL^{KO} is a protein null allele-this is shown by Western blot in Fig 2D. Also in OCRL^{KO}, the PIP₂ levels are elevated (Fig 2E) recapitulating what has been described by us in (Akhtar et.al 2022 PMID: 35023542). We will explicitly state this detail around line 185.

8. Figure 1I, could the protein levels at the different stages be quantified?

Yes, we can and will do it in the revision

9. Figure 3A, there seem to be much more cells in LSP2, making it tricky to compare with the other cell lines. Density during differentiation can affect the cell fate. Please, provide images from the different lines that are comparable with similar density.

We controlled for cell density by seeding equal number of cells 50,000 cells/cm² for all the genotypes, as mentioned in the material and methods. However, heterogeneity between lines during terminal differentiation is well-established, leading to crowding in some genotypes while not in others. Additionally, different growth rates during terminal differentiation also leads to crowded neural cultures as a function of genotype. Therefore, to complement our immunostaining data, we have provided western blot analyses showing increased GFAP protein levels in LS patient lines compared to controls. We will provide images from different lines that are comparable in density during the revision.

Please provide quantification to the statement that there is fewer number of S100B cells in the LSP lines.

As we haven't quantified the number of S100B cells, we will remove that statement.

10. Figure 3B, the images show cells very different, and it is tricky to compare similarities and differences, please provide images that look more similar to each other. Avoid images with clusters of cells or make sure to select representative images with clusters from each cell line. If the clustering is a phenotype explain and quantify that. Make sure the density is similar in all pictures.

We will provide images of matched density during the revision. Also see response to comment above.

Revision Plan

11. Line 2018, the statement "In the same cultures, there was no change in the staining pattern of the neuronal markers MAP2 and CTIP2 (Fig 3B)" is not strengthened by the figure. Please provide new pictures or data to prove the statement.

As CTIP2 staining is inherently observed in either clumps or sparsely distributed regions across WT1 and LSP genotypes, we will replace the CTIP2 marker with TBR1, which is also a deep layer cortical marker (layer VI-V), as shown below. Using this additional marker for neurons, we continue to see no change in staining pattern of neuronal markers MAP2 and TBR1. Corresponding images for each genotype are optically zoomed-in images of individual neurons positive for MAP2 and TBR1. Scale bar=50µm, 20µm.

12. Figure 3E, please describe all markers in the picture, thus also MAP2, S100B, CTIP2 and draw conclusions. Try to show comparable pictures.
This will be attended in the revision

13. Fig 3D and G, what are the replicates? please explain.

Each point represents a single neural induction done on iPSCs to generate NSCs and then terminally differentiated 30DIV cultures. Experiments were done across 3-6 independent neural inductions. This detail will be included in the revised figure legend.

14. Figure 4 A, C, there is a large difference in the ratio of different cell types between the different cell lines, also between the LSP2 and LSP3. This would indicate either that the genetic background affects the phenotype to a large extent or that there is large variability between rounds of differentiation. To understand how much variability that comes from the differentiation and culturing: another replicate of WP cell from another donor (WT2) should be included (single nuclei RNAseq). Confirm that three independent rounds of differentiation of the WT1, WT2, LSP2, LSP3, LSP4, and OCRL-KO result in similar outcome when it comes to cell type distribution. Could be done with qPCR marker.

Revision Plan

For scientific reasons explained in response to the reviewer's comment #2 we feel it is not necessary to perform replicates of the single nucleus multiome seq. However to allay the reviewer's concern of variability between differentiations leading to a conclusion of altered cell state we present the following three suggestions for a revised manuscript:

- (i) We will perform multiple differentiations from iPSC to NSC and test the altered cell state using Q-PCR for transcripts of glial lineage markers.
- (ii) Shown below are western blot analyses for WT1, LSP2, LSP3 and LSP4 NSCs (left). Analyses were done from 4 independent rounds of neural inductions and exhibit a significant increase in the levels of a astrocytic fate-determinant marker NF1A in LSP NSCs wrt to WT1 (Mann Whitney test used to measure statistical significance). Each point represents sample from an independent neural differentiation.

- (iii) We would also like to highlight that we have already demonstrated increased GFAP levels in LS patient derived differentiated cultures and OCRL^{KO}. These data, quantified in Fig 3D are done using samples derived from multiple differentiations of iPSC to NSC and then terminally differentiated. Thus the phenotype of enhanced glial cells in LS derived cultures, is most likely a consequence of the increased number of glial precursor cells is seen across multiple differentiations.

15. Line 309, "astrocytic transcripts NF1A and GFAP was elevated" It is unclear from this sentence in which cell lines NF1A and GFAP is elevated? Please explain.

We acknowledge the incompleteness in the statement. We will add the complete statement explaining the graphs. The levels of astrocytic transcripts NF1A and GFAP were elevated in LSP3 and LSP4 compared to WT1.

16. Figure 5C, E, G, there is a large variation of Notch and Hes5 expression between the different

This comment is incomplete.

17. Figure 5H, unclear which of the bands that is DLK1 and how the bands relate to the quantification. The band at 50 kDa seems to be stronger in the WT2 than in the OCRL-KO but in the quantification in Figure 5I, it shows 2x more in the KO. Thus, the other way around.

The datasheet of DLK1 antibody used (Abcam ab21682; RRID_AB731965) describes bands seen at 50,48, 45 and 15kDa. We have quantified the bands at 50kDa and 48-45kDa for all the genotypes. This will be explicitly stated in the revised figure legend.

18. Figure 6, please show that the inhibitor is inhibiting PIP5KC. Have you titrated the added concentration of the inhibitor?

Figure legend: Fields of view from WT1 derived NSC expressing the plasma membrane PIP₂ reporter. Plasma membrane distribution of the probe indicating PIP₂ levels is shown in (A) untreated cells (B) treatment with 10µM and (C) 50µM UNC-3230 PIP5K1C inhibitor. Scale bar=50µm (D) Quantification of plasma membrane PIP₂ levels using this reporter. Y-axis shows probe levels at PM; X-axis shows treatment conditions.

Yes, we used a previously generated plasma membrane PH-PLC::mCherry reporter WT1-NSCs (Akhtar et.al., 2021) and carried out a dose-response experiment using 10µM and 50µM of the UNC-3230 PIP5K1C inhibitor as shown above. We quantified intensity of PI(4,5)P₂::mCherry at the plasma membrane and plotted the mean intensity. We observed a significant decrease in plasma PI(4,5)P₂ levels at 50µM (Statistical used: Mann Whitney test) but not 10µM and therefore we selected that concentration for our experiments.

19. Figure 6B, why do the calcium data for the WT2+1Ci look so different to the other, the dots are much more spread and seem to fewer replicates than for the other sample, please explain.

We had only analysed a few replicates for WT2+1Ci genotype. We analysed the remaining replicates and have updated the data as shown below. The revised data set resolves the reviewer's concern. The revised data set will be included in the revision.

Revision Plan

20. Figure 6F, there is no significant differences between the bars but the statement in the text (sentence starts on line 332) indicate it is, please update the figure or remove the statement.

We added more replicates (now total is 7-10 biological replicates each with 15-20 organoids) and updated the figure (panel B) is shown below. The differences between treated and untreated of OCRL^{KO} are significant whereas there is no significant difference between wild type, treated and untreated (statistical test: Mann Whitney test).

Revised figure will be included in the revision

21. Figure 6G, the HES5 expression seem to behave very similar in both WT2 and OCRL-KO cells when the inhibitor is used. What does this mean? Seems to not be linked to OCRL. Explain.

Thank you for your comment. In our initial experiment (shown in original version of manuscript), we observed a reduction in *HES5* expression upon inhibitor treatment in both WT2 and OCRL-KO cells. However, to ensure robustness of our findings, we repeated the experiment across multiple, additional independent organoid differentiation batches. In this redone experiment, we no longer observe the previous trend. Instead, we see no significant changes in WT2 on inhibitor treatment, while OCRL^{KO} cells show a reduction in *HES5* expression upon inhibitor treatment (Panel A). Similarly, the protein levels of cNotch and DLK1 are not different between WT2 and WT2+1Ci (panel B and C). This strongly suggests loss of OCRL leading to elevated levels of PIP₂ perturbs Notch pathway, resulting in higher cNotch and thereby increased effector expression of *HES5*. New data set will be included in the revision.

Minor comments

The panels in Figure 6 are not completely referred to correctly in the text, please check.

Double check that all figure panels are referred to properly in the text

Yes, we will correct it in the revised manuscript.

Revision Plan

Reviewer #1 (Significance (Required)):

The manuscript is an interesting addition to the in vitro iPSC derived cellular modelling of neurodevelopmental disorder.

Strengths: The use of both patient iPSC lines and CRISPR edited lines

The use of both monolayer and 3D cultures

We thanks the reviewer for their detailed critique. Addressing these has helped improve the manuscript. We thank the reviewer for appreciating the strengths of the manuscript.

Weaknesses: the significance decrease a bit due too few replicates (only 1 WT line in each experiment) and the variability between the patients' cell lines.

We thank the reviewer for this comment. As explained above we have added substantially more data and revised the analysis which should remove this concern.

Reviewer 2:

This paper describes the effects of loss of OCRL (the Lowe syndrome protein) upon the function and differentiation of neurones, using an in vitro iPSC model system. Cells derived from three related Lowe syndrome patients and an OCRL knockout, generated using CRISPR, were used for these experiments. The results show that upon loss of OCRL, differentiation of stem cells into neurones is reduced, with an increased number of cells adopting glial and astrocytic fates. The neurones that are generated have reduced calcium transients and electrical activity. Gene expression data combined with biochemical analysis indicate altered Notch activity, which may account for the altered cell fate data seen in the in vitro differentiation model. Finally, rescue of cell fate and neuronal activity is seen upon knockdown of a PIP5K, which indicates that these phenotypes are due to the elevated PIP2 levels seen on the OCRL-deficient cells.

The results provide new insights into the pathogenesis of Lowe syndrome. I found the paper to be well done, and the data supports the conclusions of the authors. I have a few comments below that may improve the manuscript:

We thank the reviewer for summarizing the comprehensive nature of our study and appreciating the value of our study in providing new insights into the pathogenesis of Lowe syndrome with respect to the brain. Thank you for appreciating that our study is well done, and that the data supports the conclusions of the authors.

Major points

1. The UMAP and ATAC-Seq data indicate different maps for the two different Lowe syndrome patient-derived cells (Fig 4 and Fig S3). This suggests that the cells are quite different, and therefore that changes seen in one Lowe syndrome patient may not be applicable to the others. I think this heterogeneity has important implications for the paper i.e. how general are findings

Revision Plan

obtained? Several different glioblast types are described (numbered 1-5)- how different or similar are these?

We are unclear what the reviewer means by “ the UMAP and ATAC seq data indicate different maps.....”.

UMAP is a technique for visually representing data generated by single cell analysis methods be it RNAseq or ATAC seq. Perhaps what the reviewer means is that the UMAP generated from RNA seq and ATAC seq data looks different from each other.

We would like to reiterate that the UMAP generated from single cell RNA seq data is based on the complement of transcripts in each cell of the analysis compared to an existing single cell RNAseq data set, whereas the UMAP generated from ATACseq is generated from regions of open chromatin detected in and around genes and therefore presumably also reflecting ongoing gene expression. In principle the two analyses for any set of cells should indicate overall clustering into similar groups on UMAPs generated using both data sets, if the ATACseq based read out of transcription largely maps the RNAseq based read out of differences in transcription. However, it may not be reasonable to expect them to be identical. This is indeed what we see for our data set, and this has been represented in Fig 4E. The cell clusters detected based on GEX (gene expression i.e single cell RNA seq) analysis are plotted against the cells clusters detected from ATACseq data using a confusion matrix. As can be seen from this panel (Fig 4E), a very large fraction of cells falls on the diagonal indicated a large degree of similarity between clusters detected by both methods (GEX and ATACseq) of analysis. This can be reiterated more strongly during the revision by strengthening this statement.

2. The PIP5K inhibitor seems to have a very strong effect on both WT and KO cells in terms of Notch activity (Fig 5G). This strongly suggests the effects of this inhibitor are not through OCRL and that changes in PIP2 induced by the inhibitor override those of OCRL. Thus, the experiments shown in Fig 5 seem not to be due to a rescue of OCRL activity as such.

We think reviewer means Fig 6G and our response is as follows:

In our initial experiment (shown in the current version of manuscript), we observed a reduction in HES5 expression upon inhibitor treatment in both WT2 and OCRL^{KO} cells. However, to ensure robustness of our findings, we repeated the experiment across multiple, additional independent organoid differentiation batches. In this redone experiment, we no longer observe the previous trend. Instead, we see no significant changes in WT2 on inhibitor treatment, while OCRL^{KO} cells

Revision Plan

show a reduction in HES5 expression upon inhibitor treatment (Panel A). Similarly, the protein levels of cNotch and DLK1 are not different between WT2 and WT2+1Cⁱ (panel B and C). This strongly suggests loss of OCRL leading to elevated levels of PIP₂ perturbs Notch pathway, resulting in higher cNotch and thereby increased effector expression of HES5. The figures updated with the new data will be included in the revision.

Minor points

1. The main text needs to say what synapsin is and why it was analysed. In Fig 1I, synapsin abundance declines at 90 days. This appears quite strange. The authors should comment on it in the text.

We will add a line about use of synapsin in the western. Synapsin is only used qualitatively to highlight mature neuronal culture age, as was done in Sidhaye et.al PMID: 36989136.

In the revised main text, we will add the following explanation: "We also analyzed the expression of synapsin-1, a synaptic vesicle protein that serves as a marker for mature synapses and functional neuronal networks. The presence of synapsin-1 indicates the development of synaptic connections in our cultures, providing evidence of neuronal maturation."

The decline and thereby variability in synapsin-1 protein levels has been reported before. Regarding the decline in synapsin-1 at 90 days, we can add the following discussion:

"We observed a decline in synapsin-1 levels at 90 days in vitro (DIV) compared to earlier time points. This pattern has been previously reported in iPSC-derived neuronal models (Togo et.al PMID: 34629097 and Nazir et.al PMID: 30342961). Such variability in synapsin-1 expression over extended culture periods may reflect the dynamic nature of synaptic remodeling and maturation processes in vitro. It's important to note that synapsin-1 levels can fluctuate due to various factors, including culture conditions and the heterogeneity of neuronal populations present at different time points."

2. In Fig 2A and 3B there are clumps of green cells (CTIP2 positive). I am concerned that the lack of uniformity in the cell distribution could impact other analysis performed, where certain fields of view have been analysed e.g. by imaging or electrophysiology e.g. calcium measurements.

To address the reviewers concern about uniformity, in the revised manuscript, we will provide/replace the representative images of deep layer markers along with MAP2 from all genotypes showing the areas selected for analysis to demonstrate that data collection was performed in comparable regions across all experimental conditions. As answered in the response to reviewer 1, comment 11.

The clumps of neurons (as seen in Fig2A) poses challenges for obtaining high-quality seals during patch-clamp recordings. To address this, we primarily selected areas with sparsely distributed neurons for electrophysiology experiments. This approach ensured robust recordings. To address this, we can provide a clarification in the Methods section to explicitly state that neurons used for all patch-clamp recordings were chosen from regions where cells were sparsely distributed.

In case of calcium imaging experiments, we focused on both crowded and sparse fields of views across genotypes to avoid potential biases introduced by clumped cells. However, it is to be noted

that during the stages of terminal differentiation there are NSCs undergoing proliferation, which makes the neuronal culture denser. We can provide video files as a supplementary material to demonstrate the types of areas used for calcium imaging experiments. Additionally, we will include a statement in the Methods section specifying that regions with uniform neuronal distribution were selected for calcium imaging to ensure consistency in our analysis.

3. In Fig 2J and 2K are the differences between samples significant? The error bars are huge.

From line 204-209, we have not used the word “significantly different”. We acknowledge that the error bars in Figures 2J and 2K are indeed large, which is not uncommon in electrophysiological recordings from iPSC-derived neurons due to their inherent variability. We have intentionally refrained from claiming statistical significance for these specific comparisons. Instead, we describe the data as showing a pattern or trend of reduced currents in OCRL^{KO} neurons compared to WT2. To improve clarity, we propose to add a statement in the results section acknowledging the variability in these measurements and explaining our interpretation of the data as a trend rather than a statistically significant difference.

4. In Fig S4- it would be good to show gene expression analysis and GFAP staining
We are not completely sure what this comment means. However the present figure shows double staining with GFAP and S100beta. These will be split and shown separately to enhance clarity.
5. Fig 5A needs more annotation- fold change comparing what to what?
We will add the annotation “fold change wrt to WT1”.
6. There should be more information provided in the main text relating to DLK1. For example, it is shown to be secreted, but no information is provided on whether this is expected. Secreted? The DLK1 blot in Fig 5F is not convincing.

We will add more information relating to DLK1 and secretion status.

DLK1 is a non-canonical notch ligand that is indeed known to be secreted by neighboring cells to either activate/inhibit notch pathway. While we acknowledge the blot could have been better, however, variability in the blot could arise due to differences in secretion efficiency, or protein stability in the cell culture media that could have led to inconsistencies across LSP genotypes. However, as shown in the blot, the OCRL^{KO} shows a clear enrichment of secreted-DLK1 compared to WT2.

We have performed the western blot analyses across two independent differentiations of organoids from WT1, LSP2, LSP3, LSP4, WT2, OCRL-KO iPSCs in phenol-free neurobasal-A medium, and quantified secreted protein. We then loaded 40µg of protein per genotype. Shown below is the quantification. The quantification of mean intensity of DLK1 band shows a moderate increase in LSP2, and substantial increase in LSP3 and LSP4 organoids as compared to WT1. While OCRL-KO a substantial increase compared to its control, WT2. A revised figure will be used in the revision.

7. Rationale for choosing PIP5K1C
PIP5K1C is one of the major regulators maintaining appropriate levels of the synaptic pool of PI(4,5)P₂, synaptic transmission and synaptic vesicle trafficking (Hara et al., 2013 PMID: 23802628; Morleo et al., 2023 PMID: 37451268; Wenk et al., 2001 PMID: 11604140). Therefore, we were interested in rescuing the physiological phenotype, we chose PIP5K1C. Additionally, in initial experiments we found that inhibiting PIP5K1B using ISA-2011B killed the organoids or lead to detachment of 2D neuronal cultures.
8. Fig 6D is confusing. I suspect the figure labelling is not correct- it does not correlate with the graphs.
We apologise for the error and will correct this.

Reviewer #2 (Significance (Required)):

This paper is significant because it provides important new information on the neurological features of Lowe syndrome. The approach is novel in terms of studying this condition. The findings are likely to be of interest to clinicians, cell biologists, neurobiologists and those studying human development. My expertise is in membrane traffic and OCRL/Lowe syndrome. I am not a neurobiologist.

We thank the reviewer for appreciating the importance of our study, novelty of findings and new of our approach we have used. We would light to highlight that while extensive work has been done with respect to the renal phenotype of Lowe syndrome, the brain phenotypes have remained largely a black box. This is in part because mouse knockouts of OCRL have failed to recapitulate the brain related clinical phenotypes displayed by Lowe syndrome patients (for e.g. PMID: 30590522; PMCID: PMC6548226; DOI: 10.1093/hmg/ddy449). Our study of brain development defects in Lowe syndrome depleted cells provides the first insight into the cellular and developmental changes in this disorder.

Reviewer 3:

This paper by Sharma et al describes findings in an iPSC model of Lowe Syndrome. This is an important line of research because no mouse models phenocopy the neurodevelopmental aspects of the condition. They identified a potential role of Notch signaling in pathogenesis, a potentially druggable target. However, several issues need to be addressed.

We thank the reviewer for appreciating the importance of our study in covering the basis of the neurodevelopmental phenotype of Lowe syndrome. Due to a lack of a mouse model, there was previously no understanding of how the clinical features related to the brain arise.

Major issues

1. The sample size is very small, which is understandable to some extent given the expense and difficulty doing research using iPSCs. However, there are a couple of opportunities to improve the sample size. For example, in the analysis of DLK1 and other proteins shown in Figure 5, the analysis amounts to a single control vs the 3 patient lines, and a single control vs the KO line. The separation is justified because a complete KO of the gene might result in differences compared to hypomorphic mutation that apparently affects the 3 cases. However, there is no reason why WT1 and WT2 shouldn't be combined. They are both random controls. This might not affect the results of the other proteins analyzed, NOTCH and HES5, but the significance of DLK1 could change.

Nature of the allele in LS patient lines

There is a misconception in the reviewer comment that the OCRL allele in the three Lowe syndrome lines is a hypomorph. This is not correct. In the patients from whom these LS lines were generated, the nature of the OCRL allele and the status of OCRL protein in cells have been previously described in detail in a peer-reviewed, published paper from our lab. This paper (Akhtar et.al 2022 PMID: 35023542) has been cited in the present manuscript at the very first occasion that the LS patient lines are described (Line 174, references 26 and 27). As described in in ref 26 and 27, LSP patients have a mutation in exon 8 leading to a stop codon. This results in a protein null allele of OCRL in all three patient lines. This has been shown in Fig 1B of Akhtar et.al 2022 by immunofluorescence using an OCRL specific antibody (PMID: 35023542). It has also been demonstrated by Western blot using an OCRL specific antibody for all three LS patient lines in Fig 3C and 5C of the present manuscript. The nature of the allele will be highlighted more clearly in the revision.

Combining WT1 and WT2

We are not in favour of combining WT1 and WT2. The reason for this is as follows.

It is well recognized and discussed that genetic background can be a key factor contributing to phenotypes observed in cells differentiated from iPSC (Anderson et al., 2021, PMID: 33861989; Brunner et al., 2023, PMID: 36385170; Hockemeyer and Jaenisch, 2016, PMID: 27152442; Soldner and Jaenisch, 2012, PMID: 30340033; Volpato and Webber, 2020, PMID: 31953356). As a result, it is recommended that a line closely matched for genetic background be used when assessing the validity of observed phenotypes. The patient lines used in this study for Lowe syndrome were all derived from a family in India of Indian ethnic origin. Therefore, in order to reduce the impact of genetic background contributing to potential phenotypes, we have used a control line (referred to in this manuscript as WT1) derived from

Revision Plan

an individual of Indian ethnic background; this line has previously been developed and published by our group (PMID: 29778976 DOI: [10.1016/j.scr.2018.05.001](https://doi.org/10.1016/j.scr.2018.05.001)).”

In the case of OCRL^{KO} we have genome edited NCRM5 (a white Caucasian male control line) to introduce a stop codon in exon 8 to mimic the truncation seen in our LS patient lines. This allele is also protein null as shown by Western blot using an OCRL specific antibody. The data is shown in Fig 2D of the present manuscript. Therefore, we reiterate that all the LS patient lines in this study and OCRL^{KO} are **protein null alleles**.

Status of DLK1 levels

We have performed a combined analysis of DLK1 levels in the two control lines and all the patient lines as well as OCRL^{KO}. As shown below the result remains unchanged, namely that DLK1 levels are elevated in OCRL depleted cells in this model system.

Figure legend: Quantification of DLK1 protein levels in control, LS patient and OCRL^{KO} iPSC lines. Western blot intensities for each patient line and OCRL^{KO} were normalized to GAPDH and then to the respective internal WT control (WT1 or WT2) resulting in fold-change values. For statistical analysis across genotypes, normalized fold-change values from different gels were pooled post hoc. All statistical testing was performed on fold-change values. Statistical test used: Mann Whitney test. (A) Values for WT1 and WT2 have been combined and plotted against individual values for three patient lines and OCRL^{KO} (B) Values for WT1 and WT2 have been combined and plotted against combined values for all three LSP lines and OCRL^{KO}.

Reviewer comment: DLK1 expression brings up another point. This, along with MEG3 and MEG8 are imprinted genes, two of the top differentially expressed genes in this study. However, these findings can be questioned by the well-known phenomenon that the expression of some imprinted genes may not be properly maintained during iPSC reprogramming. Thus, the differential expression of these imprinted genes might be due to a reprogramming artifact rather than the effects of OCRL per se. Analyzing both controls together could mitigate this objection. However, even if it does, the potential dysregulation of imprinted genes in the development of iPSCs should be acknowledged and addressed.

Revision Plan

We are aware that the DLK1 locus is imprinted. However, we feel that reprogramming artifacts are very unlikely to explain the observed changes in DLK1 levels.

It is important to note that the patient lines and WT1 were not directly re-programmed from White blood cells to iPSC and then used for differentiation and analysis. As detailed in our previous peer-reviewed publications WT1 (PMID: 29778976) and the patient LSP lines (PMID: 35023542) were first converted to lymphoblastoid cell lines and subsequently reprogrammed into iPSC.

We think that re-programming induced imprinting changes are unlikely to be responsible for the elevated levels of DLK1 seen in LS patient lines. The reason is as follows:

We compared DLK1 levels in WT2 and OCRL^{KO} which is a CRISPR edited line that introduces a stop codon in exon 8. NCRM-5/WT2 was derived from CD34+ cord blood cells. What we found is that levels of DLK1 are elevated in OCRL^{KO} compared to WT2. Since OCRL^{KO} was generated by genome editing WT2, it must be the case that the level of imprinting of the DLK1-DIO3 locus is comparable if not identical between the two lines. Therefore, the difference in DLK1 levels between WT2 and OCRL^{KO} cannot be a consequence of different imprinting status of the DLK1 locus between these two lines. Rather, it strongly suggests a causal link to OCRL deficiency. Following on from this, the DLK1 levels are elevated in patient lines compared to the OCRL^{KO}. We will highlight and discuss and explain this in the revised version.

2. Similarly, in the calcium signaling experiment shown in fig.2, the KO and patient lines are justifiably separated. However, again, why not combine both controls in the comparison with the patient samples?

The data has been reanalyzed and presented as requested by the reviewer. There is no change in the conclusion.

For the reasons described above, it remains our preference to present each set of lines with the appropriate control; i.e WT1 and the three LS patient lines and WT2 with OCRL^{KO}. However, as the reviewer has asked for it, we also present below analysis in which WT1 and WT2 and combined and LS patient lines and OCRL^{KO} are combined. The replotted data is shown below. The essential conclusion shown in the main manuscript remains, namely that $[Ca^{2+}]_i$ transients in LS depleted developing neurons is lower than in wild type.

Revision Plan

Figure Legend: Replotted $[Ca^{2+}]_i$ transients from LS patient lines, OCRL^{KO} and two control cell lines WT1 and WT2 (A) There is no statistical difference in the frequency of $[Ca^{2+}]_i$ transients between WT 1 and WT2. Test used-Mann Whitney test. (B) Plot with WT1 and WT2 data combined v all three LS lines and OCRL^{KO} combined. Test used-Mann Whitney test. (C) WT1 and WT2 combined plotted against three individual patient lines and OCRL^{KO}. Statistical test used One-way ANOVA. (total neurons analyzed: WT1:808; WT2:267; LSP2:150; LSP3:462; LSP4:463; OCRL^{KO}:411)

3. Regarding the hypomorphic nature of the patient-specific iPSC, I do not see the OCRL variant that was found in the family. Please correct me if I missed that, and if it was omitted, it should be included. I suspect that the variant generates a hypomorphic OCRL protein because the authors show expression in Figure 1D. Hypomorphic OCRL mutations compared with complete KO could show differences in molecular phenotypes, as found in Barnes et al. (PMID: 30147856) in an analysis of F-actin and WAVE-1 expression.

Nature of the allele in LS patient lines

There is a misconception in the reviewer's comment that the OCRL allele in the three Lowe syndrome lines is a hypomorph. This is incorrect. In the patients from whom these LS lines were generated, the nature of the OCRL allele in them and the status of OCRL protein have all previously been described in detail in a peer-reviewed, published paper from our lab. This paper (Akhtar et.al 2022 PMID: 35023542) has been cited in the present manuscript at the very first occasion that the LS patient lines are described (Line 174, references 26 and 27). As described in in ref 26 and 27, LSP patients have a mutation in exon 8 leading to a stop codon. This results in a protein null allele of OCRL in all three patient lines. This has been shown in Fig 1B of Akhtar et.al 2022 by immunofluorescence using an OCRL specific antibody (PMID: 35023542). It has also been demonstrated by Western blot using an OCRL specific antibody for all three LS patient lines in Fig 3C and 5C of the present manuscript.

The data presented in Fig.1D, E is a publicly available resource data PMID: 36989136 as mentioned in line 155, which is an integrated proteomics and transcriptomics generated from control iPSC-derived human brain organoids at different stages of development *in-vitro*.

Minor issue

The authors use the term mental retardation on line 102 to describe the cognitive phenotype in Lowe Syndrome. Medical, legal, and advocacy groups have abandoned this term because it is viewed as offensive. It is being

Revision Plan

replaced by intellectual disability, although this term also is problematic. In any event, many conferences on autism and intellectual disabilities are attended by families, and high-functioning cases sometimes address an audience of scientists. They would object to the use of this term if presented in a talk by one of the co-authors.

Thank you. We will rephrase this line.

3. Description of the revisions that have already been incorporated in the transferred manuscript

Not applicable at this stage. The above is a revision plan.

4. Description of analyses that authors prefer not to carry out

We prefer to not carry out replicates of the single cell multiome analysis. As explained above the state of the art in the single cell analysis field is to not do so. The scientific reasons as to why such replicates are not required have been explained in the response to the reviewer comment.

19th May 2025

Dear Prof. Padinjat,

Thank you for the submission of your research manuscript to our editorial offices. I have now had the opportunity to read the manuscript, the referee reports and your point-by-point response and to discuss it with the other members of our editorial team. We all agreed that the manuscript fits the scope of EMBO Molecular Medicine but also that the referees raised important concerns that should be addressed in major revision. In our opinion your revision plan addresses the referees' criticism adequately and therefore we would like to invite submission of the revised manuscript to our journal.

We would welcome the submission of a revised version within three months for further consideration. Please let us know if you require longer to complete the revision.

Please use this link to login to the manuscript system and submit your revision: <https://embomolmed.msubmit.net/cgi-bin/main.plex>

I look forward to receiving your revised manuscript.

Yours sincerely,

Zeljko Durdevic

We require:

- 1) A .docx formatted version of the manuscript text (including legends for main figures, EV figures and tables). Please make sure that the changes are highlighted to be clearly visible.
- 2) Individual production quality figure files as .eps, .tif, .jpg (one file per figure). For guidance, download the 'Figure Guide PDF': (<https://www.embopress.org/page/journal/17574684/authorguide#figureformat>).
- 3) A .docx formatted letter INCLUDING the reviewers' reports and your detailed point-by-point responses to their comments. As part of the EMBO Press transparent editorial process, the point-by-point response is part of the Review Process File (RPF), which will be published alongside your paper.
- 4) A complete author checklist, which you can download from our author guidelines (<https://www.embopress.org/page/journal/17574684/authorguide#submissionofrevisions>). Please insert information in the checklist that is also reflected in the manuscript. The completed author checklist will also be part of the RPF.
- 5) Please note that all corresponding authors are required to supply an ORCID ID for their name upon submission of a revised manuscript.
- 6) It is mandatory to include a 'Data Availability' section after the Materials and Methods. Before submitting your revision, primary

datasets produced in this study need to be deposited in an appropriate public database, and the accession numbers and database listed under 'Data Availability'. Please remember to provide a reviewer password if the datasets are not yet public (see <https://www.embopress.org/page/journal/17574684/authorguide#dataavailability>).

12) Author contributions: the contribution of every author must be detailed in a separate section (before the acknowledgments).

13) A Conflict of Interest statement should be provided in the main text.

14) Every published paper now includes a 'Synopsis' to further enhance discoverability. Synopses are displayed on the journal webpage and are freely accessible to all readers. They include a short stand first (maximum of 300 characters, including space) as well as 2-5 one-sentences bullet points that summarizes the paper. Please write the bullet points to summarize the key NEW findings. They should be designed to be complementary to the abstract - i.e. not repeat the same text. We encourage inclusion of key acronyms and quantitative information (maximum of 30 words / bullet point). Please use the passive voice. Please attach these in a separate file or send them by email, we will incorporate them accordingly.

15) Include a Reagents and Tools Table as part of the Methods section, which can be downloaded from our author guidelines (<https://www.embopress.org/page/journal/17574684/authorguide#structuredmethods>)

Rev_Com_number: RC-2024-02825

New_manu_number: EMM-2025-21962-T

Corr_author: Padinjat

Title: Enhanced gliogenesis and delayed maturation underpin the neurodevelopmental defects in Lowe syndrome

Revision Plan

Manuscript number: RC-2024-02825

Corresponding author(s): Padinjat, Raghu

Key to revision plan document:

Black: reviewer comments

Red: response to reviewer comment by authors

Blue: specific changes done in the revised version by authors

1. General Statements [optional]

We thank the reviewers for their detailed comments on our manuscript and appreciating the novelty, quality and thoroughness of the work. Detailed responses to individual queries and revision plans are indicated below.

2. Description of the planned revisions

Reviewer 1:

Summary

The study by Sharma et al uses iPSC and neural differentiation in 2D and 3D to investigate how mutation in the OCRL gene affects neural differentiation and neurons. Mutation in the OCRL gene the cause of Lowe Syndrome (LS), a neurodevelopmental disorder. Neural cultures derived from LS patient iPSCs exhibited reduced excitability and increased glial markers expression. Additional data show increased levels of DLK1, cleaved Notch protein, and HES5 indicate upregulated Notch signaling in OCRL mutated neural cells. Treatment of brain organoids with a PIP5K inhibitor restored calcium signalling in neurons. These findings describe new dysregulated phenotypes in neural cultures of OCRL mutated cell shedding light on the underlying caus of Lowe Syndrome.

Major comments

1. In general, I think the use of iNeurons usually means direct reprogramming from a somatic cell to neurons without the iPSC stage. Could be confusing to use this term for iPSC derived neurons. Thank you for pointing this out. We have removed this term in the revised manuscript.
2. Please add at least one more replicate of WP cell line to the single nuclei RNAseq.

There is no cell line called WP1 in the manuscript. We believe the reviewer was likely referring to WT1 (wild-type 1).

10xgenomics guidelines highlight that the statistical power of a multiome experiment relies on several factors including sequencing depth, total number of cells per sample, sample size and number of cells per cell type of interest (10xgenomics). In this study, we performed a multiome experiment and obtained high-quality reads from 20,000 nuclei for each sample for both the modalities: snRNA seq and snATAC seq. The multiome kit recommends a lower limit is 10,000 nuclei per sample. Thus the number of cells sampled per cell line is double the suggested

minimum. Therefore, and consistent with other single-cell seq studies already published, our study followed the approach where biological replicates were not included (for e.g see PMID: 39487141, GSE238206; PMID: 31651061; PMID: 32109367, GSE144477; PMID: 40056913, GSE279894; PMID: 38280846 GSE250386; PMID: 36430334, GSE213798; PMID: 33333020, GSE123722; PMID: 32989314, GSE145122; PMID: 38711218, GSE243015, PMID: 38652563, GSE236197). Furthermore, single-cell RNA-seq inherently treats each individual cell as a replicate (Satija lab guidelines, PMID: 29567991; Wellcome Sanger Institute), reducing the necessity for additional biological replicates. Overall this appears to be the current standard in the field which we have followed.

Importantly, we have performed additional experiments to validate the predictions our single-nuclei RNA-seq findings experimentally. To confirm the results of the single cell RNA seq, we performed RT-PCR using RNA from NSC for 2 different glial genes NF1A and OLIG2. The result confirm elevate levels of these glial transcripts in LSP2, LSP3 and LSP4 NSC consistent with the observation of increased glial precursor cells from the single cell data. This data is present here and has also been included in the manuscript as Fig EV4 A.

We confirmed key observations noted initially in 2D neural stem cells using a 3D brain organoid model. This approach allowed us to confirm key predictions from the single cell sequencing data set. For example, in LS patient derived organoids and OCRL^{KO} organoids, we noted increased DLK1 levels (Fig5.C-D, G-H) as well as increased GFAP+ cells and GFAP gene and protein expression in LS brain organoids (Fig EV5 F, G, H).

Collectively, these results strengthen our confidence in the biological relevance of our findings from the single nuclei sequencing experiments.

3. The WT1 and the patient lines are rarely analysed together with the WT2 and KO lines, thus it is tricky to understand if the KO line is mimicking the patient lines? Please, add more merged analyses. Co-analysing all lines:
 - (i) would show if the KO line is more similar to the patient lines or to the WT1 or somewhere in between.
 - ii) Could answer questions about the variation in phenotypes between the genetic backgrounds.
 - iii) Elucidate how much variability there is between the two WT lines in your assays. If the two WT lines vary much then conclusions about phenotypes in the patients and KO lines might need to be rethought?

The reviewer is right is noting that throughout the manuscript we have analysed the patient lines with WT1 and the KO line with WT2. This was a conscious decision which we believe is the correct one for the following reasons:

It is well recognized and discussed in the literature that genetic background can be a key factor contributing to phenotypes observed in cells differentiated from iPSC (Anderson et al., 2021, PMID: 33861989; Brunner et al., 2023, PMID: 36385170; Hockemeyer and Jaenisch, 2016, PMID: 27152442; Soldner and Jaenisch, 2012, PMID: 30340033; Volpato and Webber, 2020, PMID: 31953356). Therefore, as a matter of abundant precaution, in this study we have tried to use the closest possible genetically matched control lines for analysis.

The patient lines used in this study for Lowe syndrome were all derived from a family in India of Indian ethnic origin. Therefore, in order to reduce the potential impact of genetic background contributing to potential phenotypes, we have used a control line derived from an individual of Indian ethnic background; this line has previously been developed and published by our group (PMID: 29778976 DOI: [10.1016/j.scr.2018.05.001](https://doi.org/10.1016/j.scr.2018.05.001)). By contrast, the OCRL^{KO} line was generated using the control line NCRM5 (WT2); this line is derived from a Caucasian male (RRID: CVCL_1E75). Therefore, whenever we have analyzed OCRL^{KO}, we have used NCRM5 as the control; throughout the manuscript, NCRM5 is referred to as WT2.

However, in deference to the reviewer's concerns we have performed a few analyses to compare the extent of variability between the two control lines.

Figure Legend: Replotted $[Ca^{2+}]_i$ transients' data from LS patient lines, OCRL^{KO} and two control cell lines WT1 and WT2. (A) There is no statistical difference in the frequency of $[Ca^{2+}]_i$ transients between WT 1 and WT2. Test used-Mann Whitney test. (B) Plot with WT1 and WT2 data combined versus all three LS lines and OCRL^{KO} combined. Test used-Mann Whitney test. (C) WT1 and WT2 combined plotted against three individual patient lines and OCRL^{KO}. Statistical test used One-way ANOVA. (total neurons analysed: WT1:808; WT2:267; LSP2:150; LSP3:462; LSP4:463; OCRL^{KO}:411)

(i) We compared the frequency of calcium transients between neurons of age 30 DIV between WT1 and WT2 (Panel A above). **We found no significant difference between these.**

Additionally, as suggested we combined the data from both control lines into a single set and that from all the LSP patient lines and OCRL^{KO} into another one (Panel B above). At the end of the analysis the difference between control and OCRL depleted cells remains. Please note the large number of cells studied in each genotype.

We also combined both control lines into a single control data set and compared it to each patient line and OCRL^{KO}. We find that each patient line and OCRL^{KO} is still significantly different from the control set (panel C above).

We did not find OCRL^{KO} to be significantly different from LSP2 or LSP4, indicating that the OCRL^{KO} line closely aligns with the patient-derived lines, supporting the idea that the observed phenotype is primarily disease-driven rather than background-dependent. However, we did observe a significant difference between LSP3 and OCRL^{KO}, highlighting some degree of inter-patient variability. Therefore, the key point is that the disease phenotype remains stable across different backgrounds, reinforcing the idea that the observed differences are driven by OCRL loss rather than background variability. This had been added to the revised discussion.

(ii) In our RTPCR assay for *HES5*, when WT1 and WT2 are plotted together, there is no significant difference observed (panel A below). Similarly, western blotting data for cNotch (panel C) and DLK1 (panel B) of pooled WT1 and WT2 together on one plot shows no significant difference (Unpaired t-test, Welch's correction). Overall, based on the above data, WT1 and WT2 are not statistically different.

Figure legend: Comparison of control lines WT1 and WT2. (A) comparison of *HES5* transcripts. (B) Western blot for DLK1 levels. (C) Western blot for cleaved notch protein levels. Statistical test: Unpaired t-test, Welch's correction.

- Please include more discussion and rationale around the link between the expression pattern of OCRL and the various phenotypes shown. From the RNAseq data performed at the NSC state where the expression of OCRL is lower than in neurons there are considerable differences in cell type distribution between lines. How can this skew cell type distribution affect downstream differentiation and neuronal function?

We would like to highlight that we did not perform bulk RNAseq in NSC and neurons; rather, we performed snRNA seq in NSCs (Fig3). The data in Fig.1E is mined from a publicly available resource dataset (Sidhaye et.al., 2023, PMID: 36989136) as mentioned in line 155, which is integrated proteomics and transcriptomics generated from iPSC-derived human brain organoids at different stages of development *in-vitro*.

Fig 1D and 1E do indeed show lower levels of OCRL expression in NSC compared to neurons. However, it is important to bear in mind that even though OCRL may be expressed at relatively low levels during the NSC stage, its enzymatic activity could still have a substantial impact.

Revision Plan

Therefore, even at low expression levels, OCRL could be modulating the PI(4,5)P₂ pool in ways that significantly influence cellular functions, especially during early stages of neurodevelopment that alter cell-fate decisions thereby affecting neuronal excitability. This has been added to the revised discussion.

Our working model posits that loss of OCRL leads to increased levels of PI(4,5)P₂ which upregulates Notch pathway thereby leading to an increase in its downstream effector *HES5*. *HES5* is a known transcription factor influencing gliogenesis and thus leading to a precocious glial shift in OCRL deficient NSCs as seen in our multiome dataset. This temporal perturbation in differentiation affects maturation of LS/OCRL^{KO} neurons and/or astrocytes leading to a defective neuronal excitability.

5. Also, OCRL is expressed also at the iPSC state as shown in Figure 1I, do you see any phenotypes in iPSC? If not, explain how that could be.

Yes, OCRL is indeed expressed in iPSCs as shown in Figure 1I. In an earlier paper from our lab that described the generation of these patient-derived iPSC from Lowe syndrome patients (Akhtar et.al 2022 PMID: 35023542), we have reported that PIP₂ levels are elevated at the iPSC stage as well as NSC stage in OCRL patient lines. We have not performed a detailed analysis of the iPSC stage for these lines as the focus of our investigation was primarily on the later stages of differentiation, particularly in neural progenitors and differentiated neurons. However, in response to the reviewer's questions on why there are no obvious phenotypes at the iPSC we would suggest that this is due to compensation from the activity of other genes of the 5-phosphatase family. In support of this, we would cite our previous study (Akhtar et.al 2022 PMID: 35023542), in which we show that in LS patient derived lines, at the iPSC Stage, at least six other 5-phosphatases are upregulated. A line on this has been added to the revised discussion.

6. There is not enough data in the manuscript to show mechanistic links between OCRL, DLK1 and Notch so be aware not to overstate the conclusions.

We appreciate the reviewer's constructive comment regarding the mechanistic links between OCRL, DLK1, and Notch. Treatment of organoids and neurons with UNC-3230 PIP5K1C inhibitor rescues the observed phenotypes suggesting a role for a PIP₂ dependent process, this process itself remains to be identified. We have adjusted the wording in the manuscript during the revision to ensure that this comes through and the conclusions do not appear overstated.

7. Line 173, please describe what mutation in the OCRL these patients have, is it a biallelic deletion? Is the protein totally absent? Please show western blot analyses of the protein in the patient lines.

The patients from whom these LS lines were generated, the nature of the OCRL allele in them and the status of OCRL protein have all been previously been described in detail in a paper from our lab. This paper (Akhtar et.al 2022 PMID: 35023542) has been cited in the present manuscript at the very first occasion that the lines are described (Line 291, references 26 and 27). In addition, in the present manuscript, the protein status of OCRL in all the three patient lines is shown with a Western blot in Figure 3C.

Revision Plan

Would be good with a bit of clinical explanation of these patients? Do they have the same level of severity? Are there any differences between their clinical symptoms? This could be interesting to link to differences in cellular phenotypes.

The clinical details of each patient are described in a preprint from our lab (Pallikonda et.al., 2021 bioRxiv 2021.06.22.449382). The potential reasons for the difference in severity, a very interesting scientific question, is also addressed in this preprint. Currently experimental analysis to support the proposed likely reasons is ongoing in our lab. We feel those analysis are beyond the scope of this manuscript and will be published later this year as a separate study.

8. Figure 1I, could the protein levels at the different stages be quantified?
This has been included as Fig 1J in the revision.

9. Figure 3A, there seem to be much more cells in LSP2, making it tricky to compare with the other cell lines. Density during differentiation can affect the cell fate. Please, provide images from the different lines that are comparable with similar density.

We controlled for cell density by seeding equal number of cells 50,000 cells/cm² for all the genotypes, as mentioned in the material and methods. However, heterogeneity between lines during terminal differentiation is well-established, leading to crowding in some genotypes while not in others. Additionally, different growth rates during terminal differentiation also leads to crowded neural cultures as a function of genotype. Therefore, to complement our immunostaining data, we have provided western blot analyses showing increased GFAP protein levels in LS patient lines compared to controls. We have provided new images (Fig 3B) from different lines that are comparable in density in the revision.

Please provide quantification to the statement that there is fewer number of S100B cells in the LSP lines.

As we haven't quantified the number of S100B cells, we have removed that statement.

10. Figure 3B, the images show cells very different, and it is tricky to compare similarities and differences, please provide images that look more similar to each other. Avoid images with clusters of cells or make sure to select representative images with clusters from each cell line. If the clustering is a phenotype explain and quantify that. Make sure the density is similar in all pictures.

We have provided images of matched density during the revision (Fig 3C). Also see response to comment above.

11. Line 2018, the statement "In the same cultures, there was no change in the staining pattern of the neuronal markers MAP2 and CTIP2 (Fig 3B)" is not strengthened by the figure. Please provide new pictures or data to prove the statement.

As CTIP2 staining is inherently observed in either clumps or sparsely distributed regions across WT1 and LSP genotypes, we have replaced the CTIP2 marker with TBR1, which is also a deep layer cortical marker (layer VI-V), as shown below. Using this additional marker for neurons, we continue to see no change in staining pattern of neuronal markers MAP2 and TBR1. Corresponding images for each genotype are optically zoomed-in images of individual neurons positive for MAP2 and TBR1. Scale bar=50µm, 20µm.

12. Figure 3E, please describe all markers in the picture, thus also MAP2, S100B, CTIP2 and draw conclusions. Try to show comparable pictures.

This has been attended in the revision

13. Fig 3D and G, what are the replicates? please explain.

Each point represents a single neural induction done on iPSCs to generate NSCs and then terminally differentiated 30DIV cultures. Experiments were done across 3-6 independent neural inductions. This detail has been included in the revised figure legend.

14. Figure 4 A, C, there is a large difference in the ratio of different cell types between the different cell lines, also between the LSP2 and LSP3. This would indicate either that the genetic background affects the phenotype to a large extent or that there is large variability between rounds of differentiation. To understand how much variability that comes from the differentiation and culturing: another replicate of WP cell from another donor (WT2) should be included (single nuclei RNAseq). Confirm that three independent rounds of differentiation of the WT1, WT2, LSP2, LSP3, LSP4, and OCRL-KO result in similar outcome when it comes to cell type distribution. Could be done with qPCR marker.

For scientific reasons explained in response to the reviewer's comment #2 we feel it is not necessary to perform replicates of the single nucleus multiome seq. However to allay the

Revision Plan

reviewer's concern of variability between differentiations leading to a conclusion of altered cell state we present the following three suggestions for a revised manuscript:

- (i) We have performed multiple differentiations from iPSC to NSC and tested the altered cell state using Q-PCR for transcripts of glial lineage markers. This is shown below and this data is included as Fig EV4A in the revised manuscript.

Figure Legend: RT-QPCR analysis of glia specific transcripts in RNA samples of NSC generated from wild type (WT1) and the patient lines LSP2, LSP3 and LSP4. Each point represents analysis of NSC generated from iPSC in an independent induction. Y-Axis represents transcript levels for a glial markers (A) NF1A transcripts and (B) OLIG2 transcripts normalized to GAPDH levels.

- (ii) Shown below are western blot analyses for WT1, LSP2, LSP3 and LSP4 NSCs (left). Analyses were done from 4 independent rounds of neural inductions and exhibit a significant increase in the levels of a astrocytic fate-determinant marker NF1A in LSP NSCs compared to WT1 (Mann Whitney test used to measure statistical significance). Each point represents sample from an independent neural differentiation.

- (iii) We would also like to highlight that we have already demonstrated increased GFAP levels in LS patient derived differentiated cultures and OCRL^{KO}. These data, quantified in Fig 3D are done using samples derived from multiple differentiations of iPSC to NSC and then terminally differentiated. Thus the phenotype of enhanced glial cells in LS derived cultures, is most likely a consequence of the increased number of glial precursor cells is seen across multiple differentiations.

Revision Plan

15. Line 309, "astrocytic transcripts NF1A and GFAP was elevated" It is unclear from this sentence in which cell lines NF1A and GFAP is elevated? Please explain.

We acknowledge the incompleteness in the statement. We have added a complete statement explaining the graphs. The levels of astrocytic transcripts NF1A and GFAP were elevated in LSP3 and LSP4 compared to WT1.

16. Figure 5C, E, G, there is a large variation of Notch and Hes5 expression between the different

This comment is incomplete.

17. Figure 5H, unclear which of the bands that is DLK1 and how the bands relate to the quantification. The band at 50 kDa seems to be stronger in the WT2 than in the OCRL-KO but in the quantification in Figure 5I, it shows 2x more in the KO. Thus, the other way around.

The datasheet of DLK1 antibody used (Abcam ab21682; RRID_AB731965) describes bands seen at 50,48, 45 and 15kDa. We have quantified the bands at 50kDa and 48-45kDa for all the genotypes. This has been explicitly stated in the revised figure legend.

18. Figure 6, please show that the inhibitor is inhibiting PIP5KC. Have you titered the added concentration of the inhibitor?

Figure legend: Fields of view from WT1 derived NSC expressing the plasma membrane PIP₂ reporter. Plasma membrane distribution of the probe indicating PIP₂ levels is shown in (A) untreated cells (B) treatment with 10 μM and (C) 50 μM UNC-3230 PIP5K1C inhibitor. Scale bar=50 μm (D) Quantification of plasma membrane PIP₂ levels using this reporter. Y-axis shows probe levels at PM; X-axis shows treatment conditions.

Yes, we used a previously generated plasma membrane PH-PLC::mCherry reporter WT1-NSCs (Akhtar et.al., 2021) and carried out a dose-response experiment using 10 μM and 50 μM of the UNC-3230 PIP5K1C inhibitor as shown above. We quantified intensity of PI(4,5)P₂::mCherry at the plasma membrane and plotted the mean intensity. We observed a significant decrease in

Revision Plan

plasma PI(4,5)P₂ levels at 50μM (Statistical used: Mann Whitney test) but not 10μM and therefore we selected that concentration for our experiments.

19. Figure 6B, why do the calcium data for the WT2+1Ci look so different to the other, the dots are much more spread and seem to fewer replicates that for the other sample, please explain.

We had only analysed a few replicates for WT2+1Ci genotype. We analysed the remaining replicates and have updated the data as shown below. The revised data set resolves the reviewer's concern. The revised data set is included in the revision.

20. Figure 6F, there is no significant differences between the bars but the statement in the text (sentence starts on line 332) indicate it is, please update the figure or remove the statement.

We added more replicates (now total is 7-10 biological replicates each with 15-20 organoids) and updated the figure (panel B) is shown below. The differences between treated and untreated of OCRL^{KO} are significant whereas there is no significant difference between wild type, treated and untreated (statistical test: Mann Whitney test).

Revised figure is included in the revision

21. Figure 6G, the HES5 expression seem to behave very similar in both WT2 and OCRL-KO cells when the inhibitor is used. What does this mean? Seems to not be linked to OCRL. Explain.

Thank you for your comment. In our initial experiment (shown in original version of manuscript), we observed a reduction in HES5 expression upon inhibitor treatment in both WT2 and OCRL-

KO cells. However, to ensure robustness of our findings, we repeated the experiment across multiple, additional independent organoid differentiation batches. In this redone experiment, we no longer observe the previous trend. Instead, we see no significant changes in WT2 on inhibitor treatment, while OCRL^{KO} cells show a reduction in HES5 expression upon inhibitor treatment (Panel A). Similarly, the protein levels of cNotch and DLK1 are not different between WT2 and WT2+1Cⁱ (panel B and C). This strongly suggests loss of OCRL leading to elevated levels of PIP₂ perturbs Notch pathway, resulting in higher cNotch and thereby increased effector expression of HES5. New data set is included in the revision.

Minor comments

The panels in Figure 6 are not completely referred to correctly in the text, please check. Double check that all figure panels are referred to properly in the text
This has been corrected in the revised manuscript.

Reviewer #1 (Significance (Required)):

The manuscript is an interesting addition to the in vitro iPSC derived cellular modelling of neurodevelopmental disorder.

Strengths: The use of both patient iPSC lines and CRISPR edited lines

The use of both monolayer and 3D cultures

We thanks the reviewer for their detailed critique. Addressing these has helped improve the manuscript. We thank the reviewer for appreciating the strengths of the manuscript.

Weaknesses: the significance decrease a bit due too few replicates (only 1 WT line in each experiment) and the variability between the patients' cell lines.

We thank the reviewer for this comment. As explained above we have added substantially more data and revised the analysis which should remove this concern.

Reviewer 2:

This paper describes the effects of loss of OCRL (the Lowe syndrome protein) upon the function and differentiation of neurones, using an in vitro iPSC model system. Cells derived from three related Lowe syndrome patients and an OCRL knockout, generated using CRISPR, were used for these experiments. The results show that upon loss of OCRL, differentiation of stem cells into neurones is reduced, with an increased number of cells adopting glial and astrocytic fates. The neurones that are generated have reduced calcium transients and electrical activity. Gene expression data combined with biochemical analysis indicate altered Notch activity, which may account for the altered cell fate data seen in the in vitro differentiation model. Finally, rescue of cell fate and neuronal activity is seen upon knockdown of a PIP5K, which indicates that these phenotypes are due to the elevated PIP₂ levels seen on the OCRL-deficient cells.

Revision Plan

The results provide new insights into the pathogenesis of Lowe syndrome. I found the paper to be well done, and the data supports the conclusions of the authors. I have a few comments below that may improve the manuscript:

We thank the reviewer for summarizing the comprehensive nature of our study and appreciating the value of our study in providing new insights into the pathogenesis of Lowe syndrome with respect to the brain. Thank you for appreciating that our study is well done, and that the data supports the conclusions of the authors.

Major points

1. The UMAP and ATAC-Seq data indicate different maps for the two different Lowe syndrome patient-derived cells (Fig 4 and Fig S3). This suggests that the cells are quite different, and therefore that changes seen in one Lowe syndrome patient may not be applicable to the others. I think this heterogeneity has important implications for the paper i.e. how general are findings obtained? Several different glioblast types are described (numbered 1-5)- how different or similar are these?

We are unclear what the reviewer means by “ the UMAP and ATAC seq data indicate different maps.....”.

UMAP is a technique for visually representing data generated by single cell analysis methods be it RNAseq or ATAC seq. Perhaps what the reviewer means is that the UMAP generated from RNA seq and ATAC seq data looks different from each other.

We would like to reiterate that the UMAP generated from single cell RNA seq data is based on the complement of transcripts in each cell of the analysis compared to an existing single cell RNAseq data set, whereas the UMAP generated from ATACseq is generated from regions of open chromatin detected in and around genes and therefore presumably also reflecting ongoing gene expression. In principle the two analyses for any set of cells should indicate overall clustering into similar groups on UMAPs generated using both data sets, if the ATACseq based read out of transcription largely maps the RNAseq based read out of differences in transcription. However, it may not be reasonable to expect them to be identical. This is indeed what we see for our data set, and this has been represented in Fig 4E. The cell clusters detected based on GEX (gene expression i.e single cell RNA seq) analysis are plotted against the cells clusters detected from ATACseq data using a confusion matrix. As can be seen from this panel (Fig 4E), a very large fraction of cells falls on the diagonal indicated a large degree of similarity between clusters detected by both methods (GEX and ATACseq) of analysis. This has been clearly stated in the revision.

2. The PIP5K inhibitor seems to have a very strong effect on both WT and KO cells in terms of Notch activity (Fig 5G). This strongly suggests the effects of this inhibitor are not through OCRL and that changes in PIP2 induced by the inhibitor override those of OCRL. Thus, the experiments shown in Fig 5 seem not to be due to a rescue of OCRL activity as such.

We think reviewer means Fig 6G and our response is as follows:

Revision Plan

In our initial experiment (shown in the current version of manuscript), we observed a reduction in *HES5* expression upon inhibitor treatment in both WT2 and OCRL^{KO} cells. However, to ensure robustness of our findings, we repeated the experiment across multiple, additional independent organoid differentiation batches. In this redone experiment, we no longer observe the previous trend. Instead, we see no significant changes in WT2 on inhibitor treatment, while OCRL^{KO} cells show a reduction in *HES5* expression upon inhibitor treatment (Panel A). Similarly, the protein levels of cNotch and DLK1 are not different between WT2 and WT2+1C¹ (panel B and C). This strongly suggests loss of OCRL leading to elevated levels of PIP₂ perturbs Notch pathway, resulting in higher cNotch and thereby increased effector expression of *HES5*. The figures have been updated with the new data in the revision.

Minor points

1. The main text needs to say what synapsin is and why it was analysed. In Fig 11, synapsin abundance declines at 90 days. This appears quite strange. The authors should comment on it in the text.

We will add a line about use of synapsin in the western. Synapsin is only used qualitatively to highlight mature neuronal culture age, as was done in Sidhaye et.al PMID: 36989136.

In the revised main text, we have added the following explanation: "We also analyzed the expression of synapsin-1, a synaptic vesicle protein that serves as a marker for mature synapses and functional neuronal networks. The presence of synapsin-1 indicates the development of synaptic connections in our cultures, providing evidence of neuronal maturation."

The decline and thereby variability in synapsin-1 protein levels has been reported before. Regarding the decline in synapsin-1 at 90 days, we have added the following discussion:

"We observed a decline in synapsin-1 levels at 90 days in vitro (DIV) compared to earlier time points. This pattern has been previously reported in iPSC-derived neuronal models (Togo et.al PMID: 34629097 and Nazir et.al PMID: 30342961). Such variability in synapsin-1 expression over extended culture periods may reflect the dynamic nature of synaptic remodeling and maturation processes in vitro. It's important to note that synapsin-1 levels can fluctuate due to various factors, including culture conditions and the heterogeneity of neuronal populations present at different time points."

2. In Fig 2A and 3B there are clumps of green cells (CTIP2 positive). I am concerned that the lack of uniformity in the cell distribution could impact other analysis performed, where certain fields of view have been analysed e.g. by imaging or electrophysiology e.g. calcium measurements.

To address the reviewers concern about uniformity, in the revised manuscript, we have provided representative images of deep layer markers along with MAP2 from all genotypes showing the areas selected for analysis to demonstrate that data collection was performed in comparable regions across all experimental conditions. As answered in the response to reviewer 1, comment 11.

The clumps of neurons (as seen in Fig2A) poses challenges for obtaining high-quality seals during patch-clamp recordings. To address this, we primarily selected areas with sparsely distributed neurons for electrophysiology experiments. This approach ensured robust recordings. To address this, we have provided a clarification in the Methods section to explicitly state that neurons used for all patch-clamp recordings were chosen from regions where cells were sparsely distributed.

In case of calcium imaging experiments, we focused on both crowded and sparse fields of views across genotypes to avoid potential biases introduced by clumped cells. However, it is to be noted that during the stages of terminal differentiation there are NSCs undergoing proliferation, which makes the neuronal culture denser. We can provide video files as a supplementary material to demonstrate the types of areas used for calcium imaging experiments. Additionally, we have included a statement in the Methods section specifying that regions with uniform neuronal distribution were selected for calcium imaging to ensure consistency in our analysis.

3. In Fig 2J and 2K are the differences between sampels significant? The error bars are huge.

From line 204-209, we have not used the word “significantly different”. We acknowledge that the error bars in Figures 2J and 2K are indeed large, which is not uncommon in electrophysiological recordings from iPSC-derived neurons due to their inherent variability. We have intentionally refrained from claiming statistical significance for these specific comparisons. Instead, we describe the data as showing a pattern or trend of reduced currents in OCRL^{KO} neurons compared to WT2. To improve clarity, we have added a statement in the results section acknowledging the variability in these measurements and explaining our interpretation of the data as a trend rather than a statistically significant difference.

4. In Fig S4- it would be good to show gene expression analysis and GFAP staining
We are not completely sure what this comment means. However the present figure shows double staining with GFAP and S100beta. These have been split and shown separately to enhance clarity.
5. Fig 5A needs more annotation- fold change comparing what to what?
We have added the annotation “fold change wrt to WT1”.
6. There should be more information provided in the main text relating to DLK1. For example, it is shown to be secreted, but no information is provided on whether this is expected. Secreted? The DLK1 blot in Fig 5F is not convincing.

We will add more information relating to DLK1 and secretion status.

DLK1 is a non-canonical notch ligand that is indeed known to be secreted by neighboring cells to either activate/inhibit notch pathway. While we acknowledge the blot could have been better, however, variability in the blot could arise due to differences in secretion efficiency, or protein stability in the cell culture media that could have led to inconsistencies across LSP genotypes. However, as shown in the blot, the OCRL^{KO} shows a clear enrichment of secreted-DLK1 compared to WT2.

We have performed the western blot analyses across two independent differentiations of organoids from WT1, LSP2, LSP3, LSP4, WT2, OCRL-KO iPSCs in phenol-free neurobasal-A medium, and quantified secreted protein. We then loaded 40µg of protein per genotype. Shown below is the quantification. The quantification of mean intensity of DLK1 band shows a moderate increase in LSP2, and substantial increase in LSP3 and LSP4 organoids as compared to WT1. While OCRL-KO a substantial increase compared to its control, WT2. A revised figure has been provided in the revision.

7. Rationale for choosing PIP5K1C

PIP5K1C is one of the major regulators maintaining appropriate levels of the synaptic pool of PI(4,5)P₂, synaptic transmission and synaptic vesicle trafficking (Hara et al., 2013 PMID: 23802628; Morleo et al., 2023 PMID: 37451268; Wenk et al., 2001 PMID: 11604140). Therefore, we were interested in rescuing the physiological phenotype, we chose PIP5K1C. Additionally, in initial experiments we found that inhibiting PIP5K1B using ISA-2011B killed the organoids or lead to detachment of 2D neuronal cultures.

8. Fig 6D is confusing. I suspect the figure labelling is not correct- it does not correlate with the graphs.

We have corrected this.

Reviewer #2 (Significance (Required)):

This paper is significant because it provides important new information on the neurological features of Lowe syndrome. The approach is novel in terms of studying this condition. The

findings are likely to be of interest to clinicians, cell biologists, neurobiologists and those studying human development. My expertise is in membrane traffic and OCRL/Lowe syndrome. I am not a neurobiologist.

We thank the reviewer for appreciating the importance of our study, novelty of findings and new of our approach we have used. We would like to highlight that while extensive work has been done with respect to the renal phenotype of Lowe syndrome, the brain phenotypes have remained largely a black box. This is in part because mouse knockouts of OCRL have failed to recapitulate the brain related clinical phenotypes displayed by Lowe syndrome patients (for e.g. PMID: 30590522; PMCID: PMC6548226; DOI: 10.1093/hmg/ddy449). Our study of brain development defects in Lowe syndrome depleted cells provides the first insight into the cellular and developmental changes in this disorder.

Reviewer 3:

This paper by Sharma et al describes findings in an iPSC model of Lowe Syndrome. This is an important line of research because no mouse models phenocopy the neurodevelopmental aspects of the condition. They identified a potential role of Notch signaling in pathogenesis, a potentially druggable target. However, several issues need to be addressed.

We thank the reviewer for appreciating the importance of our study in covering the basis of the neurodevelopmental phenotype of Lowe syndrome. Due to a lack of a mouse model, there was previously no understanding of how the clinical features related to the brain arise.

Major issues

1. The sample size is very small, which is understandable to some extent given the expense and difficulty doing research using iPSCs. However, there are a couple of opportunities to improve the sample size. For example, in the analysis of DLK1 and other proteins shown in Figure 5, the analysis amounts to a single control vs the 3 patient lines, and a single control vs the KO line. The separation is justified because a complete KO of the gene might result in differences compared to hypomorphic mutation that apparently affects the 3 cases. However, there is no reason why WT1 and WT2 shouldn't be combined. They are both random controls. This might not affect the results of the other proteins analyzed, NOTCH and HES5, but the significance of DLK1 could change.

Nature of the allele in LS patient lines

There is a misconception in the reviewer comment that the OCRL allele in the three Lowe syndrome lines is a hypomorph. This is not correct. In the patients from whom these LS lines were generated, the nature of the OCRL allele and the status of OCRL protein in cells have been previously described in detail in a peer-reviewed, published paper from our lab. This paper (Akhtar et.al 2022 PMID: 35023542) has been cited in the present manuscript at the very first occasion that the LS patient lines are described (Line 174, references 26 and 27). As described in in ref 26 and 27, LSP patients have a mutation in exon 8 leading to a stop codon. This results in a protein null allele of OCRL in all three patient lines. This has been shown in Fig 1B of Akhtar et.al 2022 by immunofluorescence using an OCRL specific antibody (PMID: 35023542). It has also been demonstrated by Western blot using an OCRL specific antibody for

Revision Plan

all three LS patient lines in Fig 3C and 5C of the present manuscript. The nature of the allele has been explicitly stated in line 271 of the revision.

Combining WT1 and WT2

We are not in favour of combining WT1 and WT2. The reason for this is as follows.

It is well recognized and discussed that genetic background can be a key factor contributing to phenotypes observed in cells differentiated from iPSC (Anderson et al., 2021, PMID: 33861989; Brunner et al., 2023, PMID: 36385170; Hockemeyer and Jaenisch, 2016, PMID: 27152442; Soldner and Jaenisch, 2012, PMID: 30340033; Volpato and Webber, 2020, PMID: 31953356). As a result, it is recommended that a line closely matched for genetic background be used when assessing the validity of observed phenotypes. The patient lines used in this study for Lowe syndrome were all derived from a family in India of Indian ethnic origin. Therefore, in order to reduce the impact of genetic background contributing to potential phenotypes, we have used a control line (referred to in this manuscript as WT1) derived from an individual of Indian ethnic background; this line has previously been developed and published by our group (PMID: 29778976 DOI: [10.1016/j.scr.2018.05.001](https://doi.org/10.1016/j.scr.2018.05.001))."

In the case of OCRL^{KO} we have genome edited NCRM5 (a white Caucasian male control line) to introduce a stop codon in exon 8 to mimic the truncation seen in our LS patient lines. This allele is also protein null as shown by Western blot using an OCRL specific antibody. The data is shown in Fig 2D of the present manuscript. Therefore, we reiterate that all the LS patient lines in this study and OCRL^{KO} are **protein null alleles**.

Status of DLK1 levels

We have performed a combined analysis of DLK1 levels in the two control lines and all the patient lines as well as OCRL^{KO}. As shown below the result remains unchanged, namely that DLK1 levels are elevated in OCRL depleted cells in this model system.

Figure legend: Quantification of DLK1 protein levels in control, LS patient and OCRL^{KO} iPSC lines. Western blot intensities for each patient line and OCRL^{KO} were normalized to GAPDH and then to the respective internal WT

Revision Plan

control (WT1 or WT2) resulting in fold-change values. For statistical analysis across genotypes, normalized fold-change values from different gels were pooled post hoc. All statistical testing was performed on fold-change values. Statistical test used: Mann Whitney test. (A) Values for WT1 and WT2 have been combined and plotted against individual values for three patient lines and OCRL^{KO} (B) Values for WT1 and WT2 have been combined and plotted against combined values for all three LSP lines and OCRL^{KO}.

Reviewer comment: DLK1 expression brings up another point. This, along with MEG3 and MEG8 are imprinted genes, two of the top differentially expressed genes in this study. However, these findings can be questioned by the well-known phenomenon that the expression of some imprinted genes may not be properly maintained during iPSC reprogramming. Thus, the differential expression of these imprinted genes might be due to a reprogramming artifact rather than the effects of OCRL per se. Analyzing both controls together could mitigate this objection. However, even if it does, the potential dysregulation of imprinted genes in the development of iPSCs should be acknowledged and addressed.

We are aware that the DLK1 locus is imprinted. However, we feel that reprogramming artifacts are very unlikely to explain the observed changes in DLK1 levels.

It is important to note that the patient lines and WT1 were not directly re-programmed from White blood cells to iPSC and then used for differentiation and analysis. As detailed in our previous peer-reviewed publications WT1 (PMID: 29778976) and the patient LSP lines (PMID: 35023542) were first converted to lymphoblastoid cell lines and subsequently reprogrammed into iPSC.

We think that re-programming induced imprinting changes are unlikely to be responsible for the elevated levels of DLK1 seen in LS patient lines. The reason is as follows:

We compared DLK1 levels in WT2 and OCRL^{KO} which is a CRISPR edited line that introduces a stop codon in exon 8. NCRM-5/WT2 was derived from CD34+ cord blood cells. What we found is that levels of DLK1 are elevated in OCRL^{KO} compared to WT2. Since OCRL^{KO} was generated by genome editing WT2, it must be the case that the level of imprinting of the DLK-DIO3 locus is comparable if not identical between the two lines. Therefore, the difference in DLK1 levels between WT2 and OCRL^{KO} cannot be a consequence of different imprinting status of the DLK1 locus between these two lines. Rather, it strongly suggests a causal link to OCRL deficiency. Following on from this, the DLK1 levels are elevated in patient lines compared to the OCRL^{KO}. We have highlighted this in the revised version.

2. Similarly, in the calcium signaling experiment shown in fig.2, the KO and patient lines are justifiably separated. However, again, why not combine both controls in the comparison with the patient samples?

The data has been reanalyzed and presented as requested by the reviewer. There is no change in the conclusion.

For the reasons described above, it remains our preference to present each set of lines with the appropriate control; i.e WT1 and the three LS patient lines and WT2 with OCRL^{KO}. However, as the reviewer has asked for it, we also present below analysis in which WT1 and WT2 and combined and LS patient lines and OCRL^{KO} are combined. The replotted data is shown below. The essential conclusion shown in the main manuscript remains, namely that $[Ca^{2+}]_i$ transients in LS depleted developing neurons is lower than in wild type.

Revision Plan

Figure Legend: Replotted $[Ca^{2+}]_i$ transients from LS patient lines, OCRL^{KO} and two control cell lines WT1 and WT2 (A) There is no statistical difference in the frequency of $[Ca^{2+}]_i$ transients between WT 1 and WT2. Test used-Mann Whitney test. (B) Plot with WT1 and WT2 data combined v all three LS lines and OCRL^{KO} combined. Test used-Mann Whitney test. (C) WT1 and WT2 combined plotted against three individual patient lines and OCRL^{KO}. Statistical test used One-way ANOVA. (total neurons analyzed: WT1:808; WT2:267; LSP2:150; LSP3:462; LSP4:463; OCRL^{KO}:411)

3. Regarding the hypomorphic nature of the patient-specific iPSC, I do not see the OCRL variant that was found in the family. Please correct me if I missed that, and if it was omitted, it should be included. I suspect that the variant generates a hypomorphic OCRL protein because the authors show expression in Figure 1D. Hypomorphic OCRL mutations compared with complete KO could show differences in molecular phenotypes, as found in Barnes et al. (PMID: 30147856) in an analysis of F-actin and WAVE-1 expression.

Nature of the allele in LS patient lines

There is a misconception in the reviewer's comment that the OCRL allele in the three Lowe syndrome lines is a hypomorph. This is incorrect. In the patients from whom these LS lines were generated, the nature of the OCRL allele in them and the status of OCRL protein have all previously been described in detail in a peer-reviewed, published paper from our lab. This paper (Akhtar et.al 2022 PMID: 35023542) has been cited in the present manuscript at the very first occasion that the LS patient lines are described (Line 174, references 26 and 27). As described in in ref 26 and 27, LSP patients have a mutation in exon 8 leading to a stop codon. This results in a protein null allele of OCRL in all three patient lines. This has been shown in Fig 1B of Akhtar et.al 2022 by immunofluorescence using an OCRL specific antibody (PMID: 35023542). It has also been demonstrated by Western blot using an OCRL specific antibody for all three LS patient lines in Fig 3C and 5C of the present manuscript.

The data presented in Fig.1D, E is a publicly available resource data PMID: 36989136 as mentioned in line 155, which is an integrated proteomics and transcriptomics generated from control iPSC-derived human brain organoids at different stages of development *in-vitro*.

Minor issue

The authors use the term mental retardation on line 102 to describe the cognitive phenotype in Lowe Syndrome. Medical, legal, and advocacy groups have abandoned this term because it is viewed as offensive. It is being

Revision Plan

replaced by intellectual disability, although this term also is problematic. In any event, many conferences on autism and intellectual disabilities are attended by families, and high-functioning cases sometimes address an audience of scientists. They would object to the use of this term if presented in a talk by one of the co-authors.

Thank you. We have rephrased this line.

3. Description of the revisions that have already been incorporated in the transferred manuscript

All revisions incorporated have been marked in blue in this document and highlighted in yellow in the revised manuscript.

4. Description of analyses that authors prefer not to carry out

We prefer to not carry out replicates of the single cell multiome analysis. As explained above the state of the art in the single cell analysis field is to not do so. The scientific reasons as to why such replicates are not required have been explained in the response to the reviewer comment.

18th Sep 2025

Dear Prof. Padinjat,

Thank you for the submission of your revised manuscript to EMBO Molecular Medicine. I am pleased to inform you that we will be able to accept your manuscript pending the following final amendments:

- 1) Please address referee #1 minor point and implement his/her suggestion regarding the multi-omics data shown in Fig 4 and EV3.
- 2) Authors: There is a discrepancy in how the name is displayed in our system (Raghu Padinjat) and in the manuscript (Padinjat Raghu), please adjust.
- 3) Figures: Please submit EV Figures as individual, high resolution figure files. Please update the heading of the suppl. figure legends to "Expanded View Figure Legends".
- 4) Author checklist: Please submit a complete checklist. <https://www.embopress.org/pb-assets/embosite/EMBO%20Press%20Author%20Checklist-1642513524327.xlsx>
- 5) In the main manuscript file, please do the following:
 - Please address all comments suggested by our data editors listed below:
 - o Data availability statement:
 1. Please note that the specific URLs for GSE298342, S-BSST2141 datasets are not provided in the data availability statement.
 - o Figure legends:
 1. Please define the annotated p values ****/****/* as well as provide the exact p-values for the same in the legend of figures 1E, 2C, F, H; 3E, H; 5D, E F, H, I, K; 6B, E-G; EV2 A, F; EV5 G, as appropriate.
 2. Please note that the box plots need to be defined in terms of minima, maxima, centre, bounds of box and whiskers, and percentile in the legends of figures 2F.
 3. Please note that information related to n is missing in the legends of figures 1E, F; 2C, F, H, J, K; 3E, H; 5D-F, H-K; 6B, EV2 A-F; EV4 A, B; EV5 G, H.
 4. Please note that the error bars are not defined in the legends of figures 1E, F; 5D-F; EV4 A, B; EV5 G.
 5. Please note that the scale bar needs to be defined for figure 2A, D; 3B.
 6. Please note that scale bar and its definition are missing for figure 2I.
 - Add up to 5 keywords.
 - Add callouts Fig 5H and Fig EV2E-F. Figures should be called out in a sequential order. Currently Fig EV4A-B are called out before Fig EV3G-H. Please correct.
 - Place Acknowledgements after Methods.
 - In Methods, provide the antibody dilutions that were used for each antibody. You can also do that in a table. The table should be then cited at appropriate places in Methods.
 - Please remove Reagents and Tools Table from the manuscript file and uploaded it as a separate file. More information on how to adhere to this format as well as downloadable templates (.docx) for the Reagents and Tools Table can be found in our author guidelines: <https://www.embopress.org/page/journal/17574684/authorguide#structuredmethods>
An example of a paper with Structured Methods can be found here:
<https://www.embopress.org/doi/full/10.1038/s44320-024-00037-6#sec-4>
 - In Methods, add statistical paragraph that should reflect all information that you have filled in the Authors Checklist, especially regarding randomization, blinding, replication.
 - Indicate in legends number and nature of replicates and exact p= values, not a range, along with the statistical test used. To keep the figures "clear" some authors found providing an Appendix table Sx with all exact p-values preferable. You are welcome to do this if you want to.
 - Please add "Disclosure and competing interests statement". We updated our journal's competing interests policy in January 2022 and request authors to consider both actual and perceived competing interests. Please review the policy <https://www.embopress.org/competing-interests> and update your competing interests if necessary.
 - In data availability section please remove information about data generated in other studies. The "snRNA seq data from Wang et.al., study was obtained from cell.ucsf.edu/snMultiome/ and the Braun et.al., data was9 obtained from github.com/linnarsson-lab/developing-human-brain" should be cited at an appropriate place in Methods. Data availability section should contain information only for the raw data generated in this study and deposited in public databases. Please use the following format to report the accession number of your data:

[data type]: [full name of the resource] [accession number/identifier] ([doi or URL or identifiers.org/DATABASE:ACCESSION])

Please check "Author Guidelines" for more information.

<https://www.embopress.org/page/journal/17574684/authorguide#availabilityofpublishedmaterial>

- Correct the reference citation in the reference list. Where there are more than 10 authors on a paper, 10 will be listed, followed by "et al.". Also, please remove DOIs. Please check "Author Guidelines" for more information.

<https://www.embopress.org/page/journal/17574684/authorguide#referencesformat>

6) Synopsis:

- Synopsis text: Please remove and upload as a separate .doc file and reduce the number of bullet points to max. 5.
- Synopsis image: Please provide a visual abstract as a high-resolution .jpeg file 550 px-wide x 300-600 pixels high.
- Please check your synopsis text and image before submission with your revised manuscript. Please be aware that in the proof stage minor corrections only are allowed (e.g., typos).

7) As part of the EMBO Publications transparent editorial process initiative (see our Editorial at <http://embomolmed.embopress.org/content/2/9/329>), EMBO Molecular Medicine will publish online a Review Process File (RPF) to accompany accepted manuscripts. This file will be published in conjunction with your paper and will include the anonymous referee reports, your point-by-point response and all pertinent correspondence relating to the manuscript. Let us know whether you agree with the publication of the RPF and as here, if you want to remove or not any figures from it prior to publication. Please note that the Authors checklist will be published at the end of the RPF.

8) Please provide a point-by-point letter INCLUDING my comments as well as the reviewer's reports and your detailed responses (as Word file).

I look forward to reading a new revised version of your manuscript as soon as possible.

Yours sincerely,

Zeljko Durdevic

Zeljko Durdevic
Senior Editor
EMBO Molecular Medicine

*** Instructions to submit your revised manuscript ***

- 1) a .docx formatted version of the manuscript text (including Figure legends and tables)
- 2) Separate figure files*
- 3) supplemental information as Expanded View and/or Appendix. Please carefully check the authors guidelines for formatting Expanded view and Appendix figures and tables at <https://www.embopress.org/page/journal/17574684/authorguide#expandedview>
- 4) a letter INCLUDING the reviewer's reports and your detailed responses to their comments (as Word file).
- 5) The paper explained: EMBO Molecular Medicine articles are accompanied by a summary of the articles to emphasize the major findings in the paper and their medical implications for the non-specialist reader. Please provide a draft summary of your article highlighting
 - the medical issue you are addressing,
 - the results obtained and
 - their clinical impact.This may be edited to ensure that readers understand the significance and context of the research.

Please refer to any of our published articles for an example.

6) Author contributions: the contribution of every author must be detailed in a separate section.

7) EMBO Molecular Medicine now requires a complete author checklist (<https://www.embopress.org/page/journal/17574684/authorguide>) to be submitted with all revised manuscripts. Please use the checklist as guideline for the sort of information we need WITHIN the manuscript. The checklist should only be filled with page numbers where the information can be found. This is particularly important for animal reporting, antibody dilutions (missing) and exact values and n that should be indicated instead of a range.

8) Every published paper now includes a 'Synopsis' to further enhance discoverability. Synopses are displayed on the journal webpage and are freely accessible to all readers. They include a short stand first (maximum of 300 characters, including space) as well as 2-5 one sentence bullet points that summarise the paper. Please write the bullet points to summarise the key NEW findings. They should be designed to be complementary to the abstract - i.e. not repeat the same text. We encourage inclusion of key acronyms and quantitative information (maximum of 30 words / bullet point). Please use the passive voice. Please attach these in a separate file or send them by email, we will incorporate them accordingly.

You are also welcome to suggest a striking image or visual abstract to illustrate your article. If you do please provide a jpeg file 550 px-wide x 300-600px high.

9) A Conflict of Interest statement should be provided in the main text

10) Please note that we now mandate that all corresponding authors list an ORCID digital identifier. This takes <90 seconds to complete. We encourage all authors to supply an ORCID identifier, which will be linked to their name for unambiguous name identification.

Currently, our records indicate that the ORCID for your account is 0000-0003-3578-6413.

Link Not Available

11) Include a Reagents and Tools Table as part of the Methods section, which can be downloaded from our author guidelines (<https://www.embopress.org/page/journal/17574684/authorguide#structuredmethods>)

Photos 400-800 DPI

*Additional important information regarding figures and illustrations can be found at

<https://bit.ly/EMBOPressFigurePreparationGuideline>. See also figure legend preparation guidelines:

<https://www.embopress.org/page/journal/17574684/authorguide#figureformat>

***** Reviewer's comments *****

Referee #1 (Remarks for Author):

The authors have largely addressed my points in this revised version of the manuscript. However, for the my first major point, I think the authors have misunderstood what I was asking. My point related to the difference in UMAP profiles between the two LS cultures i.e. LSP2 compared to LSP3, as shown in Fig 4A-C (derived from RNASeq). The plots are clearly quite different which makes one wonder how similar they are, and thus whether the results obtained are representative of LS generally or unique to these particular patients. Differences are also seen between the LS cells in Fig EV3 G and H (UMAP derived from ATACSeq). If there is such variability between the 2 patients, then I wonder about the applicability of the findings more generally to LS.

It has to be said that the multi-omics data shown in Fig 4 and EV3 is very hard for a non-expert to follow. I think the text and data could be presented in a clearer way for the non expert, and the clear differences between patients need to be better accounted for.

Referee #2 (Comments on Novelty/Model System for Author):

the authors have addressed the concerns presented by the reviewees

Referee #2 (Remarks for Author):

no additional comments

1) Please address referee #1 minor point and implement his/her suggestion regarding the multi-omics data shown in Fig 4 and EV3.

We thank the referee #1 for providing the clarification.

As addressed in the revision experiments and added to the revised manuscript, we carried out validation for our snRNA-seq analyses using RT-PCR on RNA isolated from LSP NSCs across several rounds of independent neural inductions. We observed that glial lineage markers *NFIA* and *OLIG2* were indeed elevated in LSP2, LSP3 and LSP4 NSC (Fig.EV3G,H) compared to WT1 NSCs. Thus, implying that gliogenic bias is present both at single cell and global levels, and can be captured across several rounds of inductions.

Regarding generalization of our finding, we would like to point out that the our gliogenic bias findings have also been recapitulated not only in the 2D NSC cultures but also in an independent three dimensional organoid model system.

Finally, we have addressed reviewer's concern by including the following lines in the discussion section (line 503-512):

"We observed variability within the LSP2 and LSP3 NSCs in their snRNA seq clustering profiles (Fig.4B,C). This variability could potentially reflect the technical variability well reported in the field (Stegle et al. 2015; Baysoy et al. 2023), or biological diversity where cells transition along a continuum rather than distinct cell types (Heumos et al. 2023). Notably, through RT-PCR analysis of all LSP NSCs generated across multiple neural inductions, we observed that loss of OCRL led to an elevation in transcript levels of glial lineage markers *NFIA* and *OLIG2* (Fig.EV3 G,H) irrespective of the LSP lines. This highlights that the gliogenic bias is present not only at single cell level but also at a global level. Furthermore, our 3D brain organoid culture system also recapitulated the gliogenic phenotypes (Fig.EV5G-I), thus supporting the robustness and generalizability of this phenotypes".

***** Reviewer's comments *****

Referee #1 (Remarks for Author):

The authors have largely addressed my points in this revised version of the manuscript. However, for the my first major point, I think the authors have misunderstood what I was asking. My point related to the difference in UMAP profiles between the two LS cultures i.e. LSP2 compared to LSP3, as shown in Fig 4A-C (derived from RNASeq). The plots are clearly quite different which makes one wonder how similar they are, and thus whether the results obtained are representative of LS generally or unique to these particular patients. Differences are also seen between the LS cells in Fig EV3 G and H (UMAP derived from ATACSeq). If there is such variability between the 2 patients, then I wonder about the applicability of the findings more generally to LS.

It has to be said that the multi-omics data shown in Fig 4 and EV3 is very hard for a non-expert to follow. I think the text and data could be presented in a clearer way for the non expert, and the clear differences between patients need to be better accounted for.

We thank the reviewer for raising this comment.

We wish to clarify that the UMAP and cell state diversity observed between LSP2 and LSP3 (monozygotic twins) is expected and reflects both intrinsic biological heterogeneity commonly encountered in single cell studies. Based upon the reviewer's comments, we have performed RTPCR for glial lineage markers *NFIA* and *OLIG2* and we saw an increase in LSP NSCs across multiple rounds of independent neural inductions (FigEV3G,H). To assist non-expert readers, we have added a text in the Discussion sections to clearly explain the sources of variability and implications for generalizability.

Referee #2 (Comments on Novelty/Model System for Author):

the authors have addressed the concerns presented by the reviewees

Referee #2 (Remarks for Author):

no additional comments

--

7th Oct 2025

Dear Prof. Raghu,

We are pleased to inform you that your manuscript is accepted for publication and is now being sent to our publisher to be included in the next available issue of EMBO Molecular Medicine.

Zeljko Durdevic
Senior Editor
EMBO Molecular Medicine

Rev_Com_number: RC-2024-02825

New_manu_number: EMM-2025-21962-V3

Corr_author: Raghu

Title: Enhanced gliogenesis and delayed maturation underpin the neurodevelopmental defects in Lowe syndrome